# Meta-Reinforcement Learning with Universal Policy Adaptation: Provable Near-Optimality under All-task Optimum Comparator

**Siyuan Xu & Minghui Zhu**
School of Electrical Engineering and Computer Science
The Pennsylvania State University
University Park, PA 16801
{spx5032, muz16}@psu.edu

## Abstract

Meta-reinforcement learning (Meta-RL) has attracted attention due to its capability to enhance reinforcement learning (RL) algorithms, in terms of data efficiency and generalizability. In this paper, we develop a bilevel optimization framework for meta-RL (BO-MRL) to learn the meta-prior for task-specific policy adaptation, which implements multiple-step policy optimization on one-time data collection. Beyond existing meta-RL analyses, we provide upper bounds of the expected optimality gap over the task distribution. This metric measures the distance of the policy adaptation from the learned meta-prior to the task-specific optimum, and quantifies the model's generalizability to the task distribution. We empirically validate the correctness of the derived upper bounds and demonstrate the superior effectiveness of the proposed algorithm over benchmarks.

## 1 Introduction

Meta-learning [58, 15, 25] aims to extract the shared prior knowledge, known as meta-prior, from the similarities and interdependencies of multiple existing learning tasks, in order to accelerate the learning process, increase the efficiency of data usage, and improve the overall learning performance in new tasks. Meta-learning has been extended to solve RL problems, known as meta-RL [15, 5], and shows its promise to overcome the challenges of traditional RL algorithms, including scarce real-world data [3, 44, 65], limited computing resources, and slow learning speed [54, 63].

Meta-learning methods can be generally categorized into optimization-based, model-based (black box methods), and metric-based methods [27, 5]. The optimization-based meta-learning approach [25] is compatible with any model trained by an optimization algorithm, such as gradient descent, and thus is applicable to a vast range of learning problems, including RL problems. Specifically, it formulates meta-learning as a bilevel optimization problem. At the lower-level optimization, the task-specific model is adapted from a shared meta-parameter by an optimization algorithm. At the upper-level optimization, the meta-parameter is to maximize the meta-objective, i.e., the performance of the model adapted from the meta-parameter over training tasks. The existing methods, including MAML and its variants [15, 38, 12], take a one-step gradient ascent as the lower-level policy optimization algorithm, which limits its data inefficiency and leads to sub-optimality.

During the meta-test, MAML conducts one-time data collection, i.e., collecting data using one policy (the meta-policy), and adapts the policy by one step of policy gradient to the new task. However, the collected data is only used in one policy gradient step, which may not sufficiently leverage the data and potentially fail to achieve a good performance. To mitigate the issue, a typical practice is to implement the data collection and the policy gradient alternately multiple times [15]. However, the

38th Conference on Neural Information Processing Systems (NeurIPS 2024).

Table 1: Solved theoretical challenges of meta-RL

| | Convergence of meta-objective | Optimality of meta-objective | Near-optimality under all-task optimum |
|---|---|---|---|
| [12, 57] | ✓ | × | × |
| [60] | × | ✓ When assuming convergence | × |
| [42] | × | × | ✓ Under optimal expert policy supervision |
| This paper | ✓ | Immediate result from [60] | ✓ |

environment exploration is usually costly and time-consuming during the meta-test in applications of meta-RL [44, 6, 36]. As a result, the low data efficiency limits the optimality of task-specific policies. In contrast, in this paper, we collect data by meta-policy for one time and utilize multiple policy optimization steps to improve the data efficiency.

The optimality analysis of MAML is studied in [12, 60] with a metric of **optimality on the meta-objective**, where the error of the meta-objective is defined by the expectation of the optimality gap between the task-specific policy adapted from the learned meta-parameter and the policy adapted from the best meta-parameter [60, 14, 26]. However, the best meta-parameter is shared for all tasks. Even if the meta-objective error is close to zero, i.e., the learned meta-parameter is close to the best one, the model adapted from the learned meta-parameter might be far from task-specific optimum for some tasks. In contrast, we aim to design a meta-RL algorithm that can fit a stronger optimality metric, called **near-optimality under all-task optimum**, where the comparator, i.e., the policy adapted from the best meta-parameter, is replaced by the task-specific optimal policy for each task. This metric offers a more strict comparator for the model adapted from the learned meta-parameter, i.e., when the metric achieves zero, the policy adaptation produces the optimal policy for every task. A similar metric is studied by [42]. It assumes that the task-specific optimal expert policy for each task is accessible and serves the supervision for policy adaptation during meta-training, which alleviates the analysis difficulty caused by the optimal policy comparator. However, the expert policy supervision is not accessible in a standard meta-RL problem. The metric under all-task optimum is also studied by [9, 10, 65] in the context of supervised meta-learning.

**Main contribution.** We develop a bilevel optimization framework for meta-RL, which implements multiple-step policy optimization on one-time data collection during task-specific policy adaptation. The overall contributions are summarized as follows. (i) We develop a universal policy optimization algorithm, which performs multiple optimization steps to maximize a surrogate of the accumulated reward function. The surrogate is developed only using one-time data collection. It includes various widely used policy optimization algorithms, including the policy gradient, the natural policy gradient (NPG) [30], and the proximal policy optimization (PPO) [52] as the special cases. Then, to learn the mete-prior, we formulate the meta-RL problem as a bilevel optimization problem, where the lower-level optimization is the universal policy optimization algorithm from the meta-policy and the upper-level optimization is to maximize the meta-objective function, i.e., the total reward of the models adapted from the meta-policy. (ii) We derive the implicit differentiation for both unconstrained and constrained lower-level optimization problems to compute the hypergradient, i.e., the gradient of the meta-objective, and propose the meta-training algorithm. In contrast to [60], we do not require to know the closed-form solution of the lower-level optimization. (iii) We derive upper bounds that quantify (a) the optimality gap between the adapted policy and the optimal task-specific policy for any task, and (b) the expected optimality gap over the task distribution. Since the proposed framework incorporates several existing meta-RL methods, such as MAML, as a special case, the analysis also provides the theoretical motivation for them. (iv) We conduct experiments to validate the theoretical bounds and verify the efficacy of the proposed algorithm on meta-RL benchmarks.

Table 1 compares the solved theoretical challenges of meta-RL between this paper and previous works [12, 57, 60, 42]. Specifically, paper [60] derives the optimality on the meta-objective under the assumption of bounded hypergradient. Papers [12, 57] consider the convergence of the meta-objective. The near-optimality under all-task optimum is considered in [42]. However, it assumes the optimal expert policies of the training tasks are available in meta-training, such that it can learn to approach the expert policies, while the other methods do not require the expert policies and learn from the explorations of the environments. In this paper, we show the convergence and optimality guarantee on the meta-objective, and, more importantly, the optimality guarantee under the all-task optimum comparator. It is noted that the optimality on the meta-objective is an immediate result from [60].

## 2 Related works.

**Categorization of meta-RL.** Meta-RL methods can be generally categorized into (i) optimization-based meta-RL, (ii) black-box (also called context-based) meta-RL. Optimization-based meta-RL approaches, such as MAML [15] and its variants [55, 38], usually include a policy adaptation algorithm and a meta-algorithm. During the meta-training, the meta-algorithm aims to learn a meta-policy, such that the policy adaptation algorithm can achieve good performance starting from the meta-policy. The learned meta-policy parameter is adapted to the new task using the policy adaptation algorithm during the meta-test. Black-box meta-RL [11, 59, 49, 47, 68] aims to learn an end-to-end neural network model. The model has fixed parameters for the policy adaptation during the meta-test, and generates the task-specific policy using the trajectories of the new task takes. In optimization-based meta-RL, the task-specific policy is adapted from a shared meta-policy over the task distribution. The learned meta-knowledge is not specialized for each task, and its meta-test performance on a task depends on a general policy optimization algorithm applied to new data from that task. In contrast, the end-to-end model in black-box meta-RL typically includes specialized knowledge for any task within the task distribution, and uses the new data merely as an indicator to identify the task within the distribution. As a result, the optimality of optimization-based methods is usually worse than black-box methods, especially when the task distribution is heterogeneous and the data scale for adaptation is extremely small. On the other hand, the policy adaptation algorithms in the meta-test of optimization-based methods can generally improve the policy starting from any initial policy, not only the learned meta-policy. Therefore, it is robust to sub-optimal meta-policy and can deal with tasks that are out of the training task distribution [16, 62]. In contrast, due to the specialization of the learned model, black-box methods cannot be generalized outside of the training task distribution. In this paper, we focus on the category of optimization-based meta-RL and compare the proposed algorithm with the existing optimization-based meta-RL approaches in terms of both experimental results and theory.

**Bilevel optimization in meta-RL.** Bilevel optimization has been widely studied empirically [45, 21, 17, 18, 53, 29] and theoretically [20, 22, 29]. It has been applied to many machine learning problems, including meta-learning [35, 48], hyperparameter optimization [45, 17, 18], RL [24, 34], and inverse RL [39, 40, 41]. Since the overall objective function in bilevel optimization is generally non-convex, theoretical analyses of bilevel optimization mainly focus on the algorithm convergence [20, 29, 64], rarely on the optimality. This paper formulates meta-RL as a bilevel optimization problem. The key theoretical contribution of this paper is to derive upper bounds on the near-optimality under all-task optimum, i.e., the expected optimality of the solutions of the lower-level optimization compared with that of the task-specific optimal policies. The near-optimality under all-task optimum is unique to meta-learning and has not been studied in the literature on bilevel optimization.

## 3 Problem statement

**MDP.** A Markov decision process (MDP) $\mathcal{M} \triangleq \{\mathcal{S}, \mathcal{A}, \gamma, \rho, P, r\}$ is defined by the bounded state space $\mathcal{S}$, the discrete or bounded continuous action space $\mathcal{A}$, the discount factor $\gamma$, the initial state distribution $\rho$ over $\mathcal{S}$, the transition probability $P(s'|s,a) : \mathcal{S} \times \mathcal{A} \times \mathcal{S} \to [0,1]$, and the reward function $r : \mathcal{S} \times \mathcal{A} \times \mathcal{S} \to [0, r_{max}]$.

**Policy and value function.** A stochastic policy $\pi : \mathcal{S} \to \mathbb{P}(\mathcal{A})$ is a map from states to probability distributions over actions, and $\pi(a|s)$ denotes the probability of selecting action $a$ in state $s$. For a policy $\pi$, the value function is defined as $V^\pi(s) \triangleq \mathbb{E}\left[\sum_{t=0}^\infty \gamma^t r(s_t, a_t, s_{t+1}) | s_0 = s, \pi\right]$. The action-value function is defined as $Q^\pi(s,a) \triangleq \mathbb{E}\left[\sum_{t=0}^\infty \gamma^t r(s_t, a_t, s_{t+1}) | s_0 = s, a_0 = a, \pi\right]$. The advantage function is defined as $A^\pi(s,a) \triangleq Q^\pi(s,a) - V^\pi(s)$. The accumulated reward function is $J(\pi) \triangleq \mathbb{E}_{s\sim\rho}\left[V^\pi(s)\right]$. Define the discounted state visitation distribution of a policy $\pi$ as $\nu^\pi(s) \triangleq \mathbb{E}_{s_0\sim\rho}[(1-\gamma)\sum_{t=0}^\infty \gamma^t \mathbb{P}(s_t = s|\pi)]$. In this paper, we consider parametric policy $\pi_\theta$, parameterized by $\theta$. The optimal parameter $\theta^*$ can maximize the accumulated reward function, i.e., $\theta^* \triangleq \operatorname{argmax}_\theta J(\pi_\theta)$. If $\theta^*$ is not unique, denote the set of the optimal solutions by $\Theta^*$.

**Meta-reinforcement learning.** Meta-RL aims to solve multiple RL tasks. Consider a space of RL tasks $\Gamma$, where each task $\tau \in \Gamma$ is modeled by a MDP $\mathcal{M}_\tau \triangleq \{\mathcal{S}, \mathcal{A}, \gamma, \rho_\tau, P_\tau, r_\tau\}$. Correspondingly, the notations $V_\tau^\pi, Q_\tau^\pi, A_\tau^\pi, \nu_\tau^\pi, \theta_\tau^*, \Theta_\tau^*$ and $J_\tau$ are defined for task $\tau$. The RL tasks follow a probability distribution $\mathbb{P}(\Gamma)$. Meta-RL aims to learn a meta-policy $\pi_\phi$ parameterized by a meta parameter $\phi$,

such that it can adapt to an unseen task $\tau_{new} \sim \mathbb{P}(\Gamma)$ with a few iterations and a small number of new environment explorations. In specific, during the meta-training, several tasks can be i.i.d. sampled from $\mathbb{P}(\Gamma)$, i.e., $\{\tau_j\}_{j=1}^T \sim \mathbb{P}(\Gamma)$, and the tasks' MDPs $\{\mathcal{M}_{\tau_j}\}_{j=1}^T$ can be explored. The meta-learner applies a meta-algorithm to update the meta parameter $\phi$ by using the data collected from the sampled tasks. During the meta-test, a new task $\tau_{new}$ is given, one time of a within-task algorithm $\mathcal{A}lg$ with data collected from $\tau_{new}$ is applied, the meta-parameter $\phi$ is adapted to the task-specific parameter $\theta'_{\tau_{new}}$ and the task-specific policy $\pi_{\theta'_{\tau_{new}}}$ is tested on the task $\tau_{new}$.

**Optimality Metric.** Consider a meta-RL algorithm that produces a meta-parameter $\phi$, and the take-specific parameter $\pi_{\theta'_\tau}$ is adapted from the meta-parameter $\phi$ on a task $\tau$, denoted as $\pi_{\theta'_\tau} = \mathcal{A}lg(\pi_\phi, \tau)$. We define the task-expected optimality gap (TEOG) as the metric to evaluate the algorithm, i.e., $\mathbb{E}_{\tau \sim \mathbb{P}(\Gamma)}[J_\tau(\pi_{\theta^*_\tau}) - J_\tau(\mathcal{A}lg(\pi_\phi, \tau))]$, where $\theta^*_\tau$ is the optimal parameter for task $\tau$. First, the TEOG considers the expected error over the task distribution $\mathbb{P}(\Gamma)$, reflecting the generalizability of the produced meta-parameter. Second, the TEOG adopts the comparator of the optimal task-specific policy $\pi_{\theta^*_\tau}$ for any task $\tau$ (all-task optimum comparator), and evaluates the optimality gap $J_\tau(\pi_{\theta^*_\tau}) - J_\tau(\mathcal{A}lg(\pi_\phi, \tau))$. In contrast, [60, 14, 26] adopts the comparator of the policy adapted from the optimal meta-parameter $\pi_{\phi^*}$, and evaluates the optimality gap $J_\tau(\mathcal{A}lg(\pi_{\phi^*}, \tau)) - J_\tau(\mathcal{A}lg(\pi_\phi, \tau))$. The latter only considers the optimality on the meta-objective, i.e., how well the trained meta-objective can approach the optimal meta-objective. However, even if the error of the meta-objective is approaching zero, i.e., the learned meta-policy is close to the best candidate, the performance of the model adapted from the optimal meta-policy might still be lacking. This is because policy optimization usually requires thousands of value/policy iterations to converge; when tasks are heterogeneous, even if it starts from the best meta-policy, one time of $\mathcal{A}lg$ with one time of value estimate may not be sufficient. In contrast, if our metric is zero, the policy adapted from the meta-parameter to any task is optimal for the task.

**Policy distance and task variance.** To find the solution for a new task within a few iterations of policy optimization, it is crucial that the meta-policy $\pi_\phi$ can benefit from learning on correlated tasks. Similar to [4, 9, 31], we measure the correlation of tasks in the task distribution $\mathbb{P}(\Gamma)$ by its variance, defined by the minimal mean square of the distances among the optimal task-specific policies, i.e., $Var(\mathbb{P}(\Gamma)) \triangleq \min_\theta \min_{\theta^*_\tau \in \Theta^*_\tau} \mathbb{E}_{\tau \sim \mathbb{P}(\Gamma)}[D^2_\tau(\pi_\theta, \pi_{\theta^*_\tau})]$. Here, $D_\tau(\pi_\theta, \pi_{\theta^*_\tau})$ is the distance metric between $\pi_\theta$ and $\pi_{\theta^*_\tau}$ on the task $\tau$ and is defined by $D_\tau(\pi_\theta, \pi_{\theta'}) \triangleq \sqrt{\mathbb{E}_{s \sim \nu_\tau^{\pi_\theta}}[d^2(\pi_\theta(\cdot|s), \pi_{\theta'}(\cdot|s))]}$, where $d(\pi_\theta(\cdot|s), \pi_{\theta'}(\cdot|s))$ is the distance of the policies $\pi_\theta$ and $\pi_{\theta'}$ on the state $s$.

Note that the distance metrics $D_\tau(\cdot, \cdot)$ and $d(\cdot, \cdot, s)$ can be custom-defined, leading to multiple policy update algorithms, as shown in Section 4. Here, we introduce several examples of $d(\cdot, \cdot, s)$ and $D_\tau(\cdot, \cdot)$, which are commonly used as the distance metrics in RL literature [51, 30, 37]. For policies $\pi_\theta$ and $\pi_{\theta'}$, we apply (i) the KL-divergence of the action probability distribution, i.e., $d_1^2(\pi_\theta, \pi_{\theta'}, s) \triangleq D_{\text{KL}}(\pi_\theta(\cdot|s)\|\pi_{\theta'}(\cdot|s))$, which is similar to the definition in [31]; (ii) The KL-divergence with the other order, i.e., $d_2^2(\pi_\theta, \pi_{\theta'}, s) \triangleq D_{\text{KL}}(\pi_{\theta'}(\cdot|s)\|\pi_\theta(\cdot|s))$; (iii) the Euclidean distance of the parameters, i.e., $d_3^2(\pi_\theta, \pi_{\theta'}, s) \triangleq \|\theta - \theta'\|^2$. Correspondingly, for $i = 1, 2$, and 3, we define $D_{\tau,i}(\pi_\theta, \pi_{\theta'}) \triangleq \sqrt{\mathbb{E}_{s \sim \nu_\tau^{\pi_\theta}}[d_i^2(\pi_\theta, \pi_{\theta'}, s)]}$. Note that the distance metrics (i)(ii) are not symmetric, i.e., $D_\tau(\pi_{\theta'}, \pi_{\theta''}) \neq D_\tau(\pi_{\theta''}, \pi_{\theta'})$, and (iii) is symmetric.

In the subsequent sections, we present algorithms based on the generalized distance definitions of $D_\tau(\cdot, \cdot)$ and $d(\cdot, \cdot, s)$. Moreover, we conduct analyses for the introduced distance metrics, from $D_{\tau,1}$ to $D_{\tau,3}$, to provide comprehensive insights into their respective performances.

## 4 Meta-Reinforcement Learning Framework

In this section, we develop a meta-RL algorithm by bilevel optimization, where the lower-level optimization is the within-task algorithm that adapts the parameter from the meta-parameter and the upper-level optimization is the meta-algorithm that obtains the meta-parameter. The proposed algorithm has two distinctions compared with existing algorithms. First, it uses one time of a universal policy optimization algorithm as the lower-level within-task algorithm. Second, we derive the hypergradient by the implicit differentiation, where the closed-form solution of the lower-level optimization is not required.

**Within-task algorithm.** Consider the policy optimization from the meta policy as the within-task algorithm $\mathcal{Alg}$. Specifically, given the meta-parameter $\phi$ and a task $\tau$, the task-specific policy $\pi_{\theta'_\tau} = \mathcal{Alg}(\pi_\phi, \lambda, \tau)$ is defined by $\theta'_\tau = \operatorname{argmax}_\theta \mathbb{E}_{s \sim \nu_\tau^{\pi_\phi}, a \sim \pi_\theta(\cdot|s)}\left[Q_\tau^{\pi_\phi}(s,a)\right] - \lambda D_\tau^2(\pi_\phi, \pi_\theta)$. When the action space $\mathcal{A}$ is discretized and the policy is tabular, i.e., the probabilities of actions are independent between different states, the above problem can be solved by $\pi_{\theta'_\tau}(\cdot|s) =$

$$\mathcal{Alg}(\pi_\phi, \lambda, \tau)(\cdot|s) = \operatorname*{argmax}_{\pi_\theta(\cdot|s)} \sum_{a \in \mathcal{A}} \pi_\theta(a|s) Q_\tau^{\pi_\phi}(s,a) - \lambda d^2(\pi_\phi(\cdot|s), \pi_\theta(\cdot|s)), \tag{1}$$

for all states $s \in \mathcal{S}$. When the policy is parameterized by an approximation function, in both continuous and discrete action space $\mathcal{A}$, $\pi_{\theta'_\tau} = \mathcal{Alg}(\pi_\phi, \lambda, \tau)$ is computed by $\theta'_\tau =$

$$\operatorname*{argmax}_\theta \mathbb{E}_{s \sim \nu_\tau^{\pi_\phi}, a \sim \pi_\phi(\cdot|s)}\left[\frac{\pi_\theta(a|s)}{\pi_\phi(a|s)} Q_\tau^{\pi_\phi}(s,a)\right] - \lambda D_\tau^2(\pi_\phi, \pi_\theta). \tag{2}$$

In (1) and (2), $\lambda > 0$ is a tuning hyperparameter and the distance metric $D_\tau$ can be arbitrarily chosen. Considering the explorations for the task $\tau$ are limited, $\mathcal{Alg}$ only needs to evaluate the $Q_\tau^{\pi_\phi}$ by Monte-Carlo sampling on a single policy $\pi_\phi$, where the data sampling complexity is exactly the same as the one-step gradient descent in MAML [15]. Therefore, we denote $\mathcal{Alg}$, i.e., collecting data on the meta-policy and solving the optimal solution of (1) and (2) as the one-time policy adaptation. More details about the data sample complexity and the computational complexity of (1) and (2) are clarified in Appendix F. On the other hand, one gradient step is usually not sufficient to identify a good policy. Therefore, $\mathcal{Alg}$ is to solve the optimal solution of (1) or (2). As shown in Section 5.4, the objective function of (1) or (2) is an approximation of the true objective function $J_\tau(\pi)$.

Note that the objective function in (1) and (2) can reduce to that of multiple widely used policy optimization approaches: (i) PPO in [51, 52] when $D_\tau = D_{\tau,2}$; (ii) a variant of the PPO [60, 37], when $D_\tau = D_{\tau,1}$; (iii) the proximally regularized policy update, i.e., the policy optimization regularized by Euclidean distance of the policy parameter [51], when $D_\tau = D_{\tau,3}$. Moreover, (iv) if we approximate the expectation in (2) by its first-order approximation and also select $D_\tau = D_{\tau,3}$, the within-task algorithm (2) also can be reduced to one-step policy gradient, as shown in Appendix H; (v) if we use the first-order approximation of the expectation in (2), the second-order approximation of the term $D_\tau^2(\pi_\phi, \pi_\theta)$, and select $D_\tau = D_{\tau,2}$, the within-task algorithm (2) is reduced to the natural policy gradient (NPG).

**Meta-algorithm.** The performance of the meta-parameter $\phi$ is evaluated by the meta-objective function, which is defined as the expected accumulated reward after the parameter is adapted by the within-task algorithm, i.e., $\mathbb{E}_{\tau \sim \mathbb{P}(\Gamma)}[J_\tau(\mathcal{Alg}(\pi_\phi, \lambda, \tau))]$. In the meta-algorithm, we maximize the meta-objective to obtain the optimal meta-parameter $\phi^*$, i.e.,

$$\phi^* = \operatorname*{argmax}_\phi \mathbb{E}_{\tau \sim \mathbb{P}(\Gamma)}[J_\tau(\mathcal{Alg}(\pi_\phi, \lambda, \tau))]. \tag{3}$$

As (1) and (2) provide multiple choices of the within-task algorithms when selecting different $D_\tau$, the meta-algorithm (3) provides the algorithms to learn the corresponding meta-priors. For example, (3) takes on the role of the meta-PPO algorithm when $D_\tau = D_{\tau,1}$ or $D_{\tau,2}$, i.e., (3) learns the meta-initialization for PPO. It is a meta-NPG algorithm with the corresponding approximation and $D_\tau$. Moreover, when $\mathcal{Alg}(\pi_\phi, \lambda, \tau)$ in (2) reduces to the one-step policy gradient shown in (iv) of the last paragraph, (3) represents a precise formulation of MAML in [15]. More details about the formulation and its relations with MAML are shown in Appendix G and H.

**Hypergradient computation.** Simlar to [29, 64], the meta-algorithm in (3) aims to solve a bilevel optimization problem. In previous works [60], they apply the policy optimizations that have known closed-form solutions as the lower-level within-task algorithms. As a result, the bilevel optimization problem is reduced to a single-level problem. In contrast, in this paper, as we consider a universal policy optimization, its closed-form solution cannot be obtained. To address the challenge, we compute $\nabla_\phi \mathcal{Alg}(\pi_\phi, \lambda, \tau)$ and the hypergradient by deriving the implicit differentiation on $\mathcal{Alg}(\pi_\phi, \lambda, \tau)$. As shown in Section 4, the optimization problem $\mathcal{Alg}(\pi_\phi, \lambda, \tau)$ is unconstrained in (2), but is constrained in (1) due to $\sum_{a \in \mathcal{A}} \pi(a|s) = 1$. Therefore, we derive the implicit differentiation for both unconstrained and constrained optimization problems. The following proposition shows the hypergradient computation for the tabular policy. Its proof is shown in Appendix J.1.

**Proposition 1** (Hypergradient for the **tabular policy**). *For the tabular policy in the discrete state-action space, consider any meta-parameter $\phi$ and the within-task algorithm (1). Let*

$\pi_{\theta'_\tau} = Alg(\pi_\phi, \lambda, \tau)$. *If* $M(s) \triangleq \lambda\nabla^2_{\pi(\cdot|s)}d^2(\pi_\phi(\cdot|s), \pi(\cdot|s))$ *is non-singular for each* $s \in$ $\mathcal{S}$, *we have* $\nabla_\phi J_\tau(\pi_{\theta'_\tau}) = \frac{1}{1-\gamma}\mathbb{E}_{s\sim\nu^{\pi_{\theta'_\tau}}_\tau}\left[\sum_{a\in\mathcal{A}}\nabla_\phi\pi_{\theta'_\tau}(a|s)Q^{\pi_{\theta'_\tau}}_\tau(s,a)\right]$, *where* $\nabla^\top_\phi\pi_{\theta'_\tau}(\cdot|s) =$ $\left(M(s)^{-1} - \frac{M(s)^{-1}\mathbf{1}\,\mathbf{1}^\top M(s)^{-1}}{\mathbf{1}^\top M(s)^{-1}\mathbf{1}}\right)\left(\nabla^\top_\phi Q^{\pi_\phi}_\tau(s,\cdot) - \lambda\nabla^\top_\phi\nabla_{\pi(\cdot|s)}d^2(\pi_\phi(\cdot|s),\pi(\cdot|s))\right)|_{\pi=\pi_{\theta'_\tau}}$.

The computation of $\nabla_\phi Q^{\pi_\phi}_\tau(s,\cdot)$ is shown in Appendix C. A sufficient condition of $M(s)$ being non-singular is that $d$ is locally strongly-convex at $\pi = \pi_{\theta'_\tau}$, shown in Appendix J.1. Moreover, when $d = d_1$ or $d = d_2$ (correspondingly, $D_\tau = D_{\tau,1}$ or $D_\tau = D_{\tau,2}$ in (1)), the matrix $M(s) = \lambda\nabla^2_{\pi(\cdot|s)}d^2(\pi_\phi(\cdot|s),\pi(\cdot|s))$ is always non-singular for any $\phi$ and $M(s)$ is always diagonal, and thus it is easy to compute $M^{-1}(s)$. The hypergradient computation $\nabla_\phi J_\tau(\pi_{\theta'_\tau})$ for $D_\tau = D_{\tau,1}$ and $D_{\tau,2}$ is shown in Appendix K.1 and L.1.

The following proposition shows the hypergradient computation for the policy with function approximation. Its proof is shown in Appendix J.2.

**Proposition 2** (Hypergradient for the **policy with function approximation**). *When a policy is represented by a function approximation, in both the discrete and continuous action spaces, for any meta-parameter* $\phi$ *and the within-task algorithm in (2). Let* $\pi_{\theta'_\tau} = Alg(\pi_\phi, \lambda, \tau)$. *If* $\nabla_\phi J_\tau(\pi_{\theta'_\tau})$ *exists,* $\nabla_\phi J_\tau(\pi_{\theta'_\tau}) = \frac{1}{1-\gamma}\nabla_\phi\theta'_\tau\mathbb{E}_{s\sim\nu^{\pi_{\theta'_\tau}}_\tau,a\sim\pi_{\theta'_\tau}(\cdot|s)}\left[\frac{\nabla_{\theta'_\tau}\pi_{\theta'_\tau}(a|s)}{\pi_{\theta'_\tau}(a|s)}Q^{\pi_{\theta'_\tau}}_\tau(s,a)\right]$, *and*

$\nabla^\top_\phi\theta'_\tau = -\mathbb{E}_{s\sim\nu^{\pi_\phi}_\tau,a\sim\pi_\phi(\cdot|s)}[\nabla^2_\theta d^2(\pi_\phi(\cdot|s),\pi_\theta(\cdot|s)) - \frac{\nabla^2_\theta\pi_\theta(a|s)}{\lambda\pi_\phi(a|s)}Q^{\pi_\phi}_\tau(s,a)]^{-1}\ \mathbb{E}_{s\sim\nu^{\pi_\phi}_\tau,a\sim\pi_\phi(\cdot|s)}$ $[\nabla^\top_\phi\nabla_\theta d^2(\pi_\phi(\cdot|s),\pi_\theta(\cdot|s)) - \frac{\nabla_\theta\pi_\theta(a|s)}{\lambda\pi_\phi(a|s)}\nabla^\top_\phi Q^{\pi_\phi}_\tau(s,a)]|_{\theta=\theta'_\tau}$.

A sufficient condition of $\nabla_\phi J_\tau(\pi_{\theta'_\tau})$ being existent is the objective function of (2) is locally strongly concave at $\theta = \theta'_\tau$, as proven in Appendix J.2. The computation of $\nabla_\phi Q^{\pi_\phi}_\tau(s,\cdot)$ is shown in Appendix C. Note that we need to compute the inverse of the Hessian when computing the hypergradient in Proposition 2. Similar to several widely used RL algorithms, such as TRPO [51] and CPO [1], we apply the conjugate gradient algorithm [23] to compute the inverse of the Hessian, which has demonstrated high efficiency across a wide range of applications of RL and meta-learning [51, 29, 15]. More clarifications about the computation efficiency of the Hessian inverse are shown in Appendix E.

---

**Algorithm 1** Meta-Training for BO-MRL

---

**Require:** Regularization weight $\lambda > 0$; Initial meta-parameter $\phi_0$; learning rate $\alpha$
1: **for** $t = 0, \cdots, T$ **do**
2:     Sample a task $\tau \sim \mathbb{P}(\Gamma)$ with the MDP $\mathcal{M}_\tau$ i.i.d.
3:     Evaluate $Q^{\pi_{\phi_t}}_\tau(\cdot,\cdot)$ for current meta-policy $\pi_{\phi_t}$ by Monte-Carlo sampling
4:     Adapt the task-specific policy $\pi_{\theta'_\tau}$ from the meta-policy $\pi_{\phi_t}$ by solving $\pi_{\theta'_\tau} = Alg(\lambda, \phi_t, \tau)$ defined in (1) or (2).
5:     Evaluate $Q^{\pi_{\theta'_\tau}}_\tau(\cdot,\cdot)$ for adapted policy $\pi_{\theta'_\tau}$ Monte-Carlo sampling
6:     Compute the hypergradient $\nabla_\phi J_\tau(\pi_{\theta'_\tau})$ in Proposition 1 or 2 by conjugate gradient method
7:     Update meta-parameter $\phi_{t+1} = \phi_t + \alpha\nabla_\phi J_\tau(\pi_{\theta'_\tau})$
8: **end for**
9: Return $\phi_T$

---

With the hypergradient computations in Proposition 1 and Proposition 2, we apply the stochastic gradient ascent (SGD) to solve the optimization problem in (3). The meta-training of the bilevel optimization framework for meta-RL (BO-MRL) is formally stated in Algorithm 1. The state-action value function in lines 3 and 5 can be estimated by many approaches, including Monte-Carlo sampling used in MAML [15] and vine in [51]. We also propose a practical algorithm of Algorithm 1, as shown in Algorithm 2 in Appendix D, which includes more implementation details of the algorithm and several mechanisms to improve Algorithm 1.

## 5  Theoretical Results

In this section, we quantify the performance of Algorithm 1, where the softmax policies and several distance metrics introduced in Section 3 are adopted. For convenience, we denote $Alg^{(1)}$ as $Alg$ in (1) and (2) when $D_\tau = D_{\tau,1}$, and denote $Alg^{(2)}$ and $Alg^{(3)}$ in an analogous way. In Section 5.1, we

introduce the softmax policy and necessary assumptions. In the following three sections, we consider two cases of Algorithm 1, including (i) Algorithm 1 with the within-task algorithm $\mathcal{A}lg^{(1)}$ and $\mathcal{A}lg^{(2)}$ for the tabular softmax policy; and (ii) Algorithm 1 with the within-task algorithm $\mathcal{A}lg^{(3)}$ for the softmax policy with function approximation. For the algorithms in (i) and (ii), we study the existence of hypergradient in Section 5.2, derive the convergence guarantees in Section 5.3, and derive the near-optimality under the all-task optimum, i.e., derive the upper bounds of TEOG, in Section 5.4.

## 5.1 Softmax policy and assumptions

We apply the softmax policies, which are commonly applied in [66, 37, 60], and use the following assumptions on the task $\tau$.

**Softmax policies.** Consider the softmax policies $\hat{\pi}_\theta$ parameterized by $\theta$ for (i) the tabular policy and (ii) the policy with function approximation. In particular, the tabular policy in a discrete state-action space is defined by $\hat{\pi}_\theta(\cdot|s) \propto \exp(\theta(s,\cdot))$, where $\theta \in \mathbb{R}^{|\mathcal{S}| \times |\mathcal{A}|}$ is a tabular map. The policy with function approximation is defined by $\hat{\pi}_\theta(\cdot|s) \propto \exp(f_\theta(s,\cdot))$, where $f_\theta$ is a function approximation model $\mathcal{S} \times \mathcal{A} \to \mathbb{R}$ with the parameter $\theta \in \mathbb{R}^n$.

**Assumption 1** (Upper bound of advantage function). *For any task $\tau \in \Gamma$ and any softmax policy $\hat{\pi}_\theta$, $|A_\tau^{\hat{\pi}_\theta}(s,a)| \leq A_{max}$ for any $a \in \mathcal{A}$ and any $s \in \mathcal{S}$.*

Since the reward $r_\tau \leq r_{max}$ is bounded, it is easy to show that $|A_\tau^{\hat{\pi}_\theta}(s,a)| \leq \frac{r_{max}}{1-\gamma}$ and Assumption 1 always holds. But we still keep Assumption 1 here, since there usually exist $A_{max}$ such that $A_{max} \ll \frac{r_{max}}{1-\gamma}$. We also have the following assumption and show its remark.

**Assumption 2** (Sufficient state visit). *For any task $\tau \in \Gamma$, there exists a constant $\epsilon > 0$, such that for all bounded parameters $\phi$, $\nu_\tau^{\hat{\pi}_\phi}(s) \geq \epsilon$ for all $s \in \mathcal{S}$.*

**Remark 1.** *Here are two sufficient conditions for Assumption 2: (i) For any task $\tau \in \Gamma$, the MDP $\mathcal{M}_\tau$ is ergodic [43, 56]; or (ii) the initial state distribution $\rho_\tau$ has $\rho_\tau(s) > 0$ for any $s \in \mathcal{S}$.*

The proof of Remark 1 is shown in Appendix O. Note that (i) of Remark 1 is a mild condition and is assumed in recent studies on RL algorithm analysis [61, 46].

For the policy with function approximation, we require the following additional assumptions on the approximate function $f_\theta$, which are standard or weaker than those in the analysis of meta-learning and meta-RL problems [9, 12, 13, 14].

**Assumption 3** (Property of the approximate function). *For any state-action pair $(s,a) \in \mathcal{S} \times \mathcal{A}$, (i) the approximate function $f_\theta(s,a)$ are cubic differentiable. (ii) $f_\theta(s,a)$ is $L_1$-Lipschitz, i.e., $\|f_{\theta_1}(s,a) - f_{\theta_2}(s,a)\| \leq L_1\|\theta_1 - \theta_2\|$ for any $\theta_1, \theta_2 \in \mathbb{R}^n$. (iii) $\nabla_\theta f_\theta(s,a)$ is $L_2$-Lipschitz, i.e., $\|\nabla_\theta f_{\theta_1}(s,a) - \nabla_\theta f_{\theta_2}(s,a)\| \leq L_2\|\theta_1 - \theta_2\|$ for any $\theta_1, \theta_2 \in \mathbb{R}^n$, (iv) $\nabla_\theta^2 f_\theta(s,a)$ is $L_3$-Lipschitz, i.e., $\|\nabla_\theta^2 f_{\theta_1}(s,a) - \nabla_\theta^2 f_{\theta_2}(s,a)\| \leq L_3\|\theta_1 - \theta_2\|$ for any $\theta_1, \theta_2 \in \mathbb{R}^n$.*

## 5.2 Existence of hypergradient.

An essential prerequisite for using Algorithm 1 is that the hypergradients in Propositions 1 and 2 exist. As shown in Section 4, for the tabular policy, when $i = 1$ or 2, the hypergradient $\nabla_\phi J_\tau(\mathcal{A}lg^{(i)}(\hat{\pi}_\phi, \lambda, \tau))$ exists for any $\phi$. For the policy with function approximation, we derive the following sufficient condition of the hypergradient being existent. Its proof is shown in Appendix M.

**Proposition 3** (Existence of hypergradient for the policy with function approximation). *In both discrete and continuous action space, consider the softmax policy with function approximation shown in Section 5.1. Suppose that Assumptions 1 and 3 hold. If $\lambda > (6L_1^2 + 2L_2)A_{max}$, $\nabla_\phi J_\tau(\mathcal{A}lg^{(3)}(\hat{\pi}_\phi, \lambda, \tau))$ always for any $\phi$.*

## 5.3 Convergence guarantee

We begin with the convergence guarantee of Algorithm 1 for the tabular policy. The following notations are used in the theorem: $B_i$, $C_i$, $G_i$, $K_i$, $M_i$ ($i = 1$ and 2), where $K_i \triangleq \frac{2(B_i + 2C_i^2)r_{max}^2}{(1-\gamma)^4}$, $M_i \triangleq \frac{(B_i + 2C_i^2)G_i r_{max}}{(1-\gamma)^4}$ for $i = 1$ and 2. $B_1 \triangleq \frac{16r_{max}}{\lambda(1-\gamma)^3} + \frac{24}{1-\gamma} + \frac{12}{\lambda}$, $C_1 \triangleq \frac{6}{1-\gamma}$, and $G_1 \triangleq \frac{4A_{max}}{(1-\gamma)^2}$. $B_2 \triangleq \frac{16r_{max}}{\lambda(1-\gamma)^3} + \frac{18}{(1-\gamma)^2}$, $C_2 \triangleq \frac{4}{1-\gamma}$, and $G_2 \triangleq \frac{2A_{max}}{(1-\gamma)^2}$.

**Theorem 1** (Convergence guarantee for **tabular softmax policy**). *Consider the tabular softmax policy in the discrete action space. Suppose that Assumptions 1 and 2 hold. Let $\{\phi_t\}_{t=1}^T$ be the sequence of meta-parameters generated by Algorithm 1 with $\lambda \geq 2A_{max}$ and the step size $\alpha = \min\left\{\left(\frac{r_{max}B_i}{(1-\gamma)^2} + \frac{2\gamma r_{max}C_i^2}{(1-\gamma)^3}\right)^{-1}, \frac{1}{G_i\sqrt{T}}\right\}$. Then, the bound:*

$$\frac{1}{T}\sum_{t=1}^T \mathbb{E}[\|\nabla_\phi \mathbb{E}_{\tau \sim \mathbb{P}(\Gamma)}[J_\tau(\mathcal{A}lg^{(i)}(\hat\pi_{\phi_t}, \lambda, \tau))]\|^2] \leq \frac{K_i}{T} + \frac{M_i}{\sqrt{T}}. \text{ holds for } i = 1 \text{ or } 2.$$

The first expectation comes from the random sampling in line 2 of Algorithm 1. The proofs of Theorem 1 are shown in Appendices K.2 and L.2.

The following theorem shows the convergence guarantee for the policy with function approximation. The notations are used in the theorem: $B_3$, $C_3$, $G_3$, $K_3$, $M_3$, where $K_3 \triangleq \frac{2(B_3+2C_3^2)r_{max}^2}{(1-\gamma)^4}$, $M_3 \triangleq \frac{(B_3+2C_3^2)G_3 r_{max}}{(1-\gamma)^4}$, $G_3 \triangleq \frac{L_1 A_{max}(\lambda+\frac{2\gamma}{1-\gamma}L_1^2 A_{max})}{(1-\gamma)(\lambda-(6L_1^2+2L_2)A_{max})}$, $C_3 \triangleq \frac{2L_1(\lambda+\frac{2\gamma}{1-\gamma}L_1^2 A_{max})}{(1-\gamma)(\lambda-(6L_1^2+2L_2)A_{max})}$, and $B_3 \triangleq \frac{(160L_1^3+56L_1 L_2+4L_3)(\lambda+\frac{2\gamma}{1-\gamma}L_1^2 A_{max})^2}{(1-\gamma)^3(\lambda-(6L_1^2+2L_2)A_{max})^2}$.

**Theorem 2** (Convergence guarantee for **softmax policy with function approximation**). *In both discrete and continuous action space, consider the softmax policy with function approximation. Suppose that Assumptions 1, 2, and 3 hold. Let $\{\phi_t\}_{t=1}^T$ be the sequence of meta-parameters generated by Algorithm 1 with $\lambda > (6L_1^2+2L_2)A_{max}$ and the step size $\alpha = \min\left\{\left(\frac{r_{max}B_3}{(1-\gamma)^2} + \frac{2\gamma r_{max}C_3^2}{(1-\gamma)^3}\right)^{-1}, \frac{1}{G_3\sqrt{T}}\right\}$. Then, the bound $\frac{1}{T}\sum_{t=1}^T \mathbb{E}\left[\|\nabla_\phi \mathbb{E}_{\tau \sim \mathbb{P}(\Gamma)}[J_\tau(\mathcal{A}lg^{(3)}(\hat\pi_{\phi_t}, \lambda, \tau))]\|^2\right] \leq \frac{K_3}{T} + \frac{M_3}{\sqrt{T}}. \text{ holds.}$*

The first expectation arises from the random sampling in line 2 of Algorithm 1. The proof of Theorem 2 is shown in Appendix M. Theorems 1 and 2 show that the convergence rate of Algorithm 1 is $\mathcal{O}(\frac{1}{\sqrt{T}})$ and the constants in the notation $\mathcal{O}$ are only related to the discount factor $\gamma$, the reward bound $r_{max}$, the bound of the advantage function $A_{max}$, and the Lipschitz constants of $f_\theta$.

## 5.4 Near-optimality under all-task optimum

Before the derivation of the optimality analysis, we first introduce two intermediate Lemmas.

**Lemma 1.** *Suppose that Assumptions 1, 2 hold. For any task $\tau$, any bounded parameters $\theta$ and $\theta'$, and $i = 1$ or 2, we have $J_\tau(\hat\pi_{\theta'}) - J_\tau(\hat\pi_\theta) \geq \mathbb{E}_{s\sim\nu_\tau^{\hat\pi}, a\sim\hat\pi'(\cdot|s)}\left[\frac{A_\tau^{\hat\pi_\theta}(s,a)}{1-\gamma}\right] - \frac{2\gamma A_{max}}{(1-\gamma)^2\epsilon}D_{\tau,i}^2(\hat\pi_\theta, \hat\pi_{\theta'}).$*

**Lemma 2.** *Consider the softmax policy with function approximation shown in Section 5.1. Suppose that Assumptions 1, 2, and 3 hold. For any task $\tau$, and any softmax policies parameterized by bounded $\theta$ and $\theta'$, we have $J_\tau(\hat\pi_{\theta'}) - J_\tau(\hat\pi_\theta) \geq \mathbb{E}_{s\sim\nu_\tau^{\hat\pi_\theta}, a\sim\hat\pi_{\theta'}(\cdot|s)}\left[\frac{A_\tau^{\hat\pi_\theta}(s,a)}{1-\gamma}\right] - \frac{4\gamma A_{max}L_1^2}{(1-\gamma)^2\epsilon}D_{\tau,3}^2(\hat\pi_\theta, \hat\pi_{\theta'}).$*

The proofs of Lemmas 1 and 2 are shown in Appendix N.1. Given Lemma 1, when $\lambda = \frac{2\gamma A_{max}}{(1-\gamma)\epsilon}$, the within-task algorithm $\mathcal{A}lg^{(1,2)}(\hat\pi, \lambda, \tau)$ in (1) is actually designed to maximize the right-hand side of the inequality, where $\hat\pi'$ is the decision variable. Similarly, Given Lemma 2, when $\lambda = \frac{4\gamma A_{max}L_1^2}{(1-\gamma)\epsilon}$, $\mathcal{A}lg^{(3)}(\hat\pi_\theta, \lambda, \tau)$ in (2) maximizes the right-hand side of the inequality, where $\hat\pi_{\theta'}$ is the decision variable. In other words, for each $i = 1, 2$, and $3$, the within-task algorithm $\mathcal{A}lg^{(i)}$ is to maximize a lower bound of $J_\tau(\hat\pi_\theta)$, denoted as $\bar{J}_\tau(\hat\pi_\theta)$. This idea, referred to as the minorization-maximization (MM) [28], is widely used in [51, 33]. The design of $\mathcal{A}lg^{(i)}$ enables us to connect the accumulated reward of the policy after the policy adaptation with that of the optimal policy $\hat\pi_{\theta_\tau^*}$ for task $\tau$, i.e., $\bar{J}_\tau(\mathcal{A}lg^{(i)}(\hat\pi_\phi, \lambda, \tau)) \geq \bar{J}_\tau(\hat\pi_{\theta_\tau^*})$, which is a key intermediate result for the optimality analysis.

The final preparatory step is that we borrow the analysis of the meta-training error from [60]. In particular, its theoretical result is encapsulated in the following assumption.

**Assumption 4.** *(Bounding error of meta-objective using gradient) Let $F^{(i)}(\phi) \triangleq \mathbb{E}_{\tau \sim \mathbb{P}(\Gamma)}[J_\tau(\mathcal{A}lg^{(i)}(\hat\pi_\phi, \lambda, \tau))]$. For both the tabular policy and the policy with functional approximation, there exists a concave positive non-decreasing function $h_i : [0, +\infty) \to [0, +\infty)$, such that $\max_{\phi'} F^{(i)}(\phi') - F^{(i)}(\phi) \leq h_i(\|\nabla_\phi F^{(i)}(\phi)\|^2).$*

Assumption 4 assumes the optimality gap of $\hat{\pi}_\phi$ on the meta-objective is upper bounded by an increasing function of its gradient. A sufficient condition of Assumption 4 is provided by [60]. Combine the Assumption 4 and the convergence analysis in Theorems 1 and 2, we can bound the error of the meta-objective, i.e., $\max_\phi F^{(i)}(\phi) - F^{(i)}(\phi_t)$. This result is referred to as the optimality of the meta-objective shown in Table 1. Finally, we derive the upper bounds of the TEOG for both the tabular policy and the policy with function approximation.

**Theorem 3** (Optimality guarantee for **softmax tabular policy**). *Consider the tabular soft-max policy for the discrete state-action space. Suppose that Assumptions 1,2 and 4 hold. Let $\{\phi_t\}_{t=1}^T$ be the sequence of meta-parameters generated by Algorithm 1 with $\lambda = \frac{2A_{max}}{(1-\gamma)\epsilon}$ and the step size $\alpha$ shown in Theorem 1. Then, the following holds for $i = 1$ or 2: $\frac{1}{T}\sum_{t=1}^T \mathbb{E}_t\left[\mathbb{E}_{\tau\sim\mathbb{P}(\Gamma)}[J_\tau(\hat{\pi}_{\theta_\tau^*}) - J_\tau(\mathcal{A}lg^{(i)}(\hat{\pi}_{\phi_t}, \lambda, \tau))]\right] \leq h_i\left(\frac{K_i}{T} + \frac{M_i}{\sqrt{T}}\right) + \frac{2(1+\gamma)A_{max}}{(1-\gamma)^2\epsilon}\mathcal{V}ar_i(\mathbb{P}(\Gamma))$, where $\hat{\pi}_{\theta_\tau^*}$ is the optimal softmax policy for task $\tau$ and the constants $K_i$ and $M_i$ are shown in Theorem 1.*

**Theorem 4** (Optimality guarantee for **softmax policy with function approximation**). *In both discrete and continuous action space, consider the softmax policy with function approximation. Suppose that Assumptions 1,2, 3 and 4 hold. Let $\{\phi_t\}_{t=1}^T$ be the sequence of meta-parameters generated by Algorithm 1 with $\lambda = \frac{(6L_1^2+2L_2)A_{max}}{(1-\gamma)\epsilon}$ and the step size $\alpha$ shown in Theorem 2. The following holds: $\frac{1}{T}\sum_{t=1}^T \mathbb{E}_t\left[\mathbb{E}_{\tau\sim\mathbb{P}(\Gamma)}[J_\tau(\hat{\pi}_{\theta_\tau^*}) - J_\tau(\mathcal{A}lg^{(3)}(\hat{\pi}_{\phi_t}, \lambda, \tau))]\right] \leq h_3\left(\frac{K_3}{T} + \frac{M_3}{\sqrt{T}}\right) + \frac{((6+4\gamma)L_1^2+2L_2)A_{max}}{(1-\gamma)^2\epsilon}\mathcal{V}ar_3(\mathbb{P}(\Gamma))$, where $\hat{\pi}_{\theta_\tau^*}$ is the optimal softmax policy for task $\tau$ and the constants $K_3$ and $M_3$ are the same as Theorem 2.*

The proofs of Theorems 3 and 4, as well as the selection of the hyperparameter $\lambda$ in these two theorems, are shown in Appendix N.2. The theorems derive the upper bounds of the TEOGs between the parameter adapted by one-time policy adaptation from the produced meta-parameter $\phi_t$ and the task-specific optimal parameter $\theta_\tau^*$. It is shown that, with at most $T$ iterations, we can achieve the upper bounds in the order of $\mathcal{O}(h_i(\frac{1}{\sqrt{T}}) + \mathcal{V}ar(\mathbb{P}(\Gamma)))$. In other words, there exists a $t \leq T$ with $\mathbb{E}_{\tau\sim\mathbb{P}(\Gamma)}[J_\tau(\mathcal{A}lg(\hat{\pi}_{\phi_t}, \lambda, \tau))] \geq \mathbb{E}_{\tau\sim\mathbb{P}(\Gamma)}[J_\tau(\hat{\pi}_{\theta_\tau^*})] - \mathcal{O}(h_i(\frac{1}{\sqrt{T}}) + \mathcal{V}ar(\mathbb{P}(\Gamma)))$. As the number of iterations $T$ increases, or the variance of the task distribution $\mathcal{V}ar(\mathbb{P}(\Gamma))$ reduces, the optimality of the meat-parameter $\phi_t$ improves. The second term $\mathcal{V}ar(\mathbb{P}(\Gamma))$ in the upper bounds of Theorems 3 and 4 corresponds the intuition of meta-learning, which is that, if the variance of a task distribution is smaller, the meta-policy learned from the task distribution is more helpful for new tasks in the task distribution, then the performance is better. Moreover, this term shows that the learned meta-policy achieves a better performance than the meta-policy $\phi^{center}$ defined by $\arg\min_\phi \mathbb{E}_{\tau\sim\mathbb{P}(\Gamma)}[D_{\tau,i}^2(\pi_\phi, \pi_{\theta_\tau^*})]$, which is the center of all the task-specific optimal policies $\pi_{\theta_\tau^*}$. The order of our upper bounds are comparable to $\mathcal{O}(T^{-\frac{1}{4}} + \mathcal{V}ar(\mathbb{P}(\Gamma)))$ that is shown in [31]. On the other hand, compared with [31], in this paper, the constants in the notation $\mathcal{O}$ only consist of $\gamma$, $r_{max}$, $A_{max}$, and the Lipschitz constants of $f$, and do not rely on $|\mathcal{A}|$ and $|\mathcal{S}|$. As a result, our upper bounds are tighter when handling high-dimensional problems or continuous spaces.

**Monotonic improvement of the within-task algorithm.** Another benefit from Lemmas 1 and 2 and the idea of MM used by the within-task algorithm is that, the policy update by the within-task algorithm monotonically improves, i.e., $J_\tau(\mathcal{A}lg^{(i)}(\hat{\pi}_\theta, \lambda, \tau)) \geq J_\tau(\hat{\pi}_\theta)$ for $i = 1, 2$ and 3 and any $\theta$ and any task $\tau$. Therefore, multiple times of $\mathcal{A}lg$ always perform better than one-time $\mathcal{A}lg$.

# 6 Experiments

## 6.1 Verification of theoretical results

We conduct an experiment to verify the optimality bounds of Algorithm 1 shown in Theorems 3 and 4. We consider two scenarios of the Frozen Lake environment in Gym: two task distributions with a high task variance and a low task variance. More details of the setting and the hyperparameter selection are shown in Appendix A. We consider the within-task algorithm $\mathcal{A}lg^{(i)}$ for all $i = 1, 2$ and 3, where the results of $i = 2$ and 3 are shown in Appendix A.

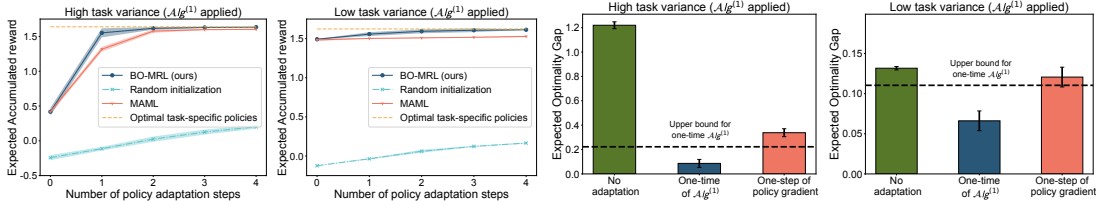

Figure 1: Results of the meta-test on Frozen Lake, where $\mathcal{A}lg^{(1)}$ is applied. **Left**: Average accumulated reward across all test tasks v.s. number of policy adaptation steps; **Right**: Comparing the expected optimality gap by the BO-MRL and baselines with the upper bound of the accumulated reward of one-time $\mathcal{A}lg^{(1)}$.

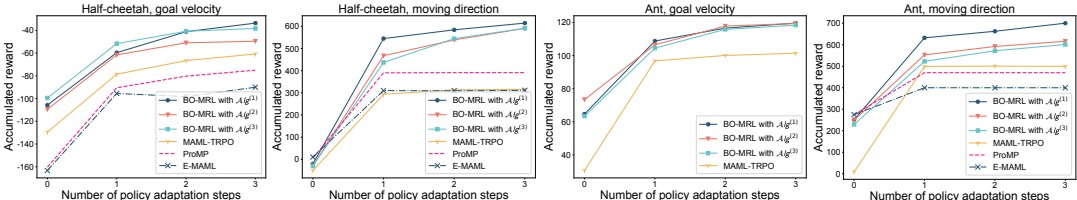

Figure 2: Average accumulated reward across all test tasks during the meta-test under the practical algorithm of BO-MRL on the locomotion tasks.

We compare our algorithm with MAML [15] and the random initialization. Figure 1 shows that, for Algorithm 1 with the within-task algorithm $\mathcal{A}lg^{(1)}$, it outperforms the baseline methods. For all scenarios, the expected optimality gap of the one-time policy adaptation is smaller than the upper bounds shown in Theorems 3 and 4, which verify our theoretical analysis. Moreover, in Figure 1, the expected optimality gap of the policy adaptation is better (smaller) but close to the upper bound, while that of the other policy adaptation approach, the policy gradient, is worse (larger) than the upper bound. It shows that the derived upper bound is tight.

## 6.2 High-dimensional Experiment

To evaluate the proposed practical algorithm, Algorithm 2 in Appendix D, we conduct experiments on high-dimensional locomotion settings in the MuJoCo simulator, including Half-Cheetah with goal directions and goal velocities, Ant with goal directions and goal velocities. We compare the proposed algorithm with several optimization-based meta-RL algorithms, including MAML, E-MAML [55], and ProMP [50]. For the fairness of the comparison, all the methods share the same data requirement and task setting. More details of the task setting, the hyperparameter selection, and the supplemental results are shown in Appendix B.

Figure 2 shows that the proposed algorithm with the within-task algorithms $\mathcal{A}lg^{(i)}$ outperforms the baseline methods in all four experimental settings. For example, we achieve about $25\%$ of performance improvement in Half-cheetah direction and Ant direction experiments. Moreover, compared with the baseline methods, the proposed algorithm achieves more policy improvement when more policy optimization steps are given. For example, our approach achieves about $10\%$ of performance improvement in the second policy optimization step, while those of baseline methods are almost $0\%$.

## 7 Conclusion

This paper develops a bilevel optimization framework for meta-RL, which implements multiple-step policy optimization on one-time data collection during task-specific policy adaptation. Beyond existing meta-RL analyses, we provide upper bounds of the expected optimality gap over the task distribution. Our experiments validate the bounds derived from our theoretical analysis and show the superior effectiveness of the proposed framework.

## Acknowledgments and Disclosure of Funding

This work is partially supported by the National Science Foundation through grants ECCS 1846706 and ECCS 2140175. We would like to thank the reviewers for their constructive and insightful suggestions.

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

# Appendix for "Meta-Reinforcement Learning with Universal Policy Adaptation: Provable Near-Optimality under All-task Optimum Comparator"

## Experimental Supplements

All experiments are executed on a computer with a 5.20 GHz Intel Core i12 CPU.

## A Experimental Supplements of Verification of Theoretical Results.

**Experimental settings.** In Section 6, we use the Frozen Lake environment in Gym [7] and consider a task distribution $\mathbb{P}(\Gamma)$ with high task variance and a task distribution $\mathbb{P}(\Gamma)$ with low task variance. In each distribution, there are 20 tasks. The tasks are characterized by the different settings of holes in the lake, where the holes are generated by random sampling. In the task distribution with high variance, the probability of the appearing hole in each grid is $0.3$; in the task distribution with low variance, its probability is $0.1$. We set $\gamma = 0.8$, the reward is 1 when reaching the goal, and the reward is $-1$ when reaching the holes. When deriving the upper bound in Theorems 3 and 4, we approximately regard $T$ be sufficiently large, and $\mathcal{O}(h_i(\frac{1}{\sqrt{T}}))$ be close to $0$. The Lipschitz of the tabular policy is 1, i.e., $L_1 = 1$; the Lipschitz of the derivative and the second-order derivative of the tabular policy are both 0, i.e., $L_2 = 0$ and $L_3 = 0$.

**Selection of hyper-parameters.** We consider the tabular softmax policy and use Monte Carlo sampling to evaluate the Q-value. For the task distribution with high task variance, we set $\lambda = 0.5$ for $\mathcal{A}lg^{(1)}$, $\lambda = 0.5$ for $\mathcal{A}lg^{(2)}$, and $\lambda = 0.04$ for $\mathcal{A}lg^{(3)}$. For the task distribution with low task variance, we set $\lambda = 0.25$ for $\mathcal{A}lg^{(1)}$, $\lambda = 0.25$ for $\mathcal{A}lg^{(2)}$, and $\lambda = 0.02$ for $\mathcal{A}lg^{(3)}$. There is a clarification about the hyper-parameter selection and the verified bound shown in Appendix N.3.

**Supplemental results.** Figures 3 and 4 show the results of the proposed algorithm with $\mathcal{A}lg^{(2)}$ and $\mathcal{A}lg^{(3)}$. It shows that, for all scenarios, the expected optimality gap of the policy adaptation $\mathcal{A}lg^{(2)}$ or $\mathcal{A}lg^{(3)}$ is smaller than the upper bound shown in Theorems 3 and 4, which verify our theoretical analysis.

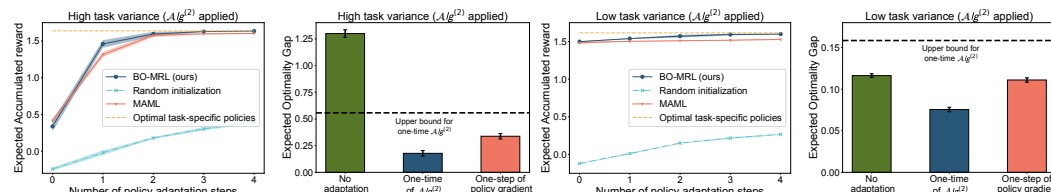

Figure 3: Results of the meta-test of BO-MRL on Frozen Lake, where $\mathcal{A}lg^{(2)}$ is applied. **Left**: Average accumulated reward across all test tasks v.s. number of policy adaptation steps; **Right**: Comparing the expected optimality gap by the BO-MRL and baselines with the upper bound of the accumulated reward of one-time $\mathcal{A}lg^{(2)}$.

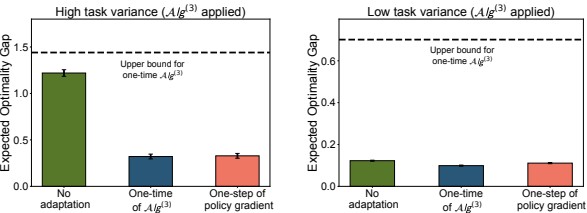

Figure 4: Results of BO-MRL on Frozen Lake, where $\mathcal{A}lg^{(3)}$ is applied. Comparing the expected optimality gap by the BO-MRL and baselines with the upper bound of the accumulated reward of one-time $\mathcal{A}lg^{(3)}$.

## B  Experimental Supplements of Locomotion.

**Experimental settings.** We consider locomotion tasks HalfCheetah with goal directions and goal velocities, Ant with goal directions and goal velocities. We follow the problem setups of [67, 15]. In the goal velocity experiments, the moving reward is the negative absolute value between the agent's current velocity and a goal velocity, which is chosen uniformly at random between $0.0$ and $2.0$ for the cheetah and between $0.0$ and $3.0$ for the ant. In the goal direction experiments, the moving reward is the magnitude of the velocity in either the forward or backward direction, chosen at random for each task $\tau$ in $\mathbb{P}$. For the Half-cheetah, the total reward = moving reward - ctrl cost. For the ant, the total reward = healthy reward + moving reward - ctrl cost - contact cost. The horizon is $H = 200$, with 20 rollouts per policy adaption step for all problems except the ant direction task, which used 40 rollouts per step.

**Selection of hyper-parameters.** We apply the proposed practical algorithm of Algorithm 1, Algorithm 2 in Appendix D. We consider the policy as a Gaussian distribution, where the neural network produces the means and variances of the actions. The neural network policy has two hidden layers of size 64, with tanh nonlinearities. We use Monte Carlo sampling to evaluate the Q-value. At the lower-level task-specific policy adaptation, the optimization number by Adam is 50. The models are trained for up to 500 meta-iterations. For the TRPO in meta-parameter optimization, we use the KL-divergence constraint as $\delta = 1e - 3$.

For the experiment of Half-Cheetah with goal velocities, we set $\lambda = 0.5$ for $\mathcal{A}lg^{(1)}$, $\lambda = 0.4$ for $\mathcal{A}lg^{(2)}$. For the experiment of Half-Cheetah with goal directions, we set $\lambda = 0.5$ for $\mathcal{A}lg^{(1)}$, $\lambda = 0.5$ for $\mathcal{A}lg^{(2)}$. For the experiment of Ant with goal velocities, we set $\lambda = 0.5$ for $\mathcal{A}lg^{(1)}$, $\lambda = 0.5$ for $\mathcal{A}lg^{(2)}$. For the experiment of Ant with goal directions, we set $\lambda = 0.5$ for $\mathcal{A}lg^{(1)}$, $\lambda = 0.5$ for $\mathcal{A}lg^{(2)}$.

**Comparison setting.** We compare the proposed algorithm with several optimization-based meta-RL algorithms, including MAML, E-MAML [55], and ProMP [50]. The experiment results of E-MAML, ProMP, and MAML-TRPO come from [67, 15]. We do not compare the proposed algorithm with black-box meta-RL algorithms, as they are based on the task context and even can achieve good performance without adaptation.

**Supplemental results.** Figure 5 shows that the proposed algorithm with both within-task algorithms $\mathcal{A}lg^{(i)}$ outperform the baseline methods in four experimental settings. The accumulated rewards of proposed algorithms increase fast and stop at points with better performance than the baseline methods.

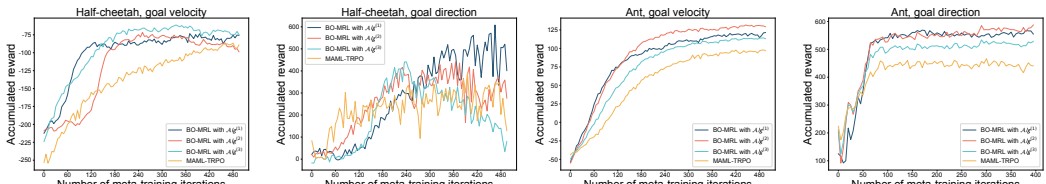

Figure 5: Accumulated rewards during the meta-training under the practical algorithm of BO-MRL on the locomotion tasks.

# Algorithm supplement

## C  Computation of $\nabla_\phi Q_\tau^{\pi_\phi}(s, a)$

In the computation of meta-objective shown in Propositions 1 and 2, we need to compute $\nabla_\phi Q_\tau^{\pi_\phi}(s, a)$

$$\nabla_\phi Q_\tau^{\pi_\phi}(s, a) = \frac{\gamma}{1 - \gamma} \cdot \mathbb{E}_{(s', a') \sim \sigma_{\tau, \pi_\phi}^{(s, a)}} \left[ \nabla_\phi \ln \pi_\phi(a'|s') Q_\tau^{\pi_\phi}(s', a') \right].$$

where the state-action visitation probability $\sigma_{\tau, \pi_\theta}^{(s, a)}$ initialized at $(s, a) \in \mathcal{S} \times \mathcal{A}$ is defined by

$$\sigma_{\tau, \pi_\phi}^{(s, a)}(s', a') = (1 - \gamma) \sum_{t=0}^{\infty} \gamma^t \mathbb{P}(s_t = s', a_t = a'|\pi_\phi, s_0 \sim P_\tau(\cdot|s, a)).$$

For the tabular softmax policy in discrete state-action space shown in Section 5.1,

$$\nabla_{\phi(s',\cdot)} Q_\tau^{\hat{\pi}_\phi}(s,a) = \frac{\gamma}{1-\gamma} \cdot \sigma_{\tau,\hat{\pi}_\phi}^{(s,a)}(s') \cdot \hat{\pi}_\phi(\cdot|s') \odot A_\tau^{\hat{\pi}_\phi}(s',\cdot), \quad (4)$$

where $\odot$ is the element-wise product, $\phi(s',\cdot)$ is the vector which includes $\phi(s',a')$ for all $a' \in \mathcal{A}$ as the elements, and $A_\tau^{\hat{\pi}_\phi}(s,\cdot)$ is the vector which includes $A_\tau^{\hat{\pi}_\phi}(s,a)$ for all $a \in \mathcal{A}$ as the elements. Equivalently,

$$\nabla_{\phi(s',a')} Q_\tau^{\hat{\pi}_\phi}(s,a) = \frac{\gamma}{1-\gamma} \cdot \sigma_{\tau,\hat{\pi}_\phi}^{(s,a)}(s') \hat{\pi}_\phi(a'|s') A_\tau^{\hat{\pi}_\phi}(s',a'). \quad (5)$$

For the softmax policy with the function approximation,

$$
\begin{aligned}
\nabla_\phi Q_\tau^{\hat{\pi}_\phi}(s,a) &= \frac{\gamma}{1-\gamma} \cdot \mathbb{E}_{(s',a') \sim \sigma_{\tau,\hat{\pi}_\phi}^{(s,a)}} \left[ \frac{\nabla_\phi \hat{\pi}_\phi(a'|s')}{\hat{\pi}_\phi(a'|s')} Q_\tau^{\hat{\pi}_\phi}(s',a') \right] \\
&= \frac{\gamma}{1-\gamma} \cdot \mathbb{E}_{(s',a') \sim \sigma_{\tau,\hat{\pi}_\phi}^{(s,a)}} \left[ \nabla_\phi f_\phi(s',a') Q_\tau^{\hat{\pi}_\phi}(s',a') \right] \\
&= \frac{\gamma}{1-\gamma} \cdot \mathbb{E}_{(s',a') \sim \sigma_{\tau,\hat{\pi}_\phi}^{(s,a)}} \left[ \nabla_\phi f_\phi(s',a') A_\tau^{\hat{\pi}_\phi}(s',a') \right]
\end{aligned}
\quad (6)
$$

*Proof.* As shown in [60],

$$
\begin{aligned}
\nabla_\phi Q_\tau^{\pi_\phi}(s,a) &= \nabla_\phi \left( (1-\gamma) \cdot r_\tau(s,a) + \gamma \cdot \mathbb{E}_{s' \sim P_\tau(\cdot|s,a)} \left[ V_\tau^{\pi_\phi}(s') \right] \right) \\
&= \frac{\gamma}{1-\gamma} \cdot \mathbb{E}_{(s',a') \sim \sigma_{\tau,\pi_\phi}^{(s,a)}} \left[ \nabla_\phi \ln \pi_\phi(a'|s') \cdot Q_\tau^{\pi_\phi}(s',a') \right] \\
&= \frac{\gamma}{1-\gamma} \cdot \mathbb{E}_{(s',a') \sim \sigma_{\tau,\pi_\phi}^{(s,a)}} \left[ \nabla_\phi \ln \pi_\phi(a'|s') \cdot A_\tau^{\pi_\phi}(s',a') \right]. \\
&= \frac{\gamma}{1-\gamma} \cdot \mathbb{E}_{(s',a') \sim \sigma_{\tau,\pi_\phi}^{(s,a)}} \left[ \frac{\nabla_\phi \pi_\phi(a'|s')}{\pi_\phi(a'|s')} \cdot A_\tau^{\pi_\phi}(s',a') \right].
\end{aligned}
$$

By Lemma 4, from (12), we can obtain (4); from (14), we can obtain (6). □

# D   Practical algorithm

In Sections 4 and 5, we develop a theoretically guaranteed algorithm with Assumptions 1, 2, and 3. In this section, we develop a practical instantiation of Algorithm 1 and evaluate its performance in high-dimensional experiments in Section 6.

Algorithm 2 states the practical algorithm of Algorithm 1. Compared with Algorithm 1, Algorithm 2 considers and overcomes the following limitations of Algorithm 1: (a) evaluating the exact expectation in (1) and (2) is costly and the approximation error could influence the task-specific policy adaptation if using sampling, especially in the meta-RL problem where the sampling data is limited; (b) the optimization problems in (1) and (2) have no closed-form solution; (c) the computation of the gradients of the meta-objectives shown in Propositions 1 and 2 is time-consuming; (d) the gradient-based approach to optimize the meta-objective is not stable in RL problems.

In the beginning of Algorithm 2, we first sample a batch of tasks $\{\tau_i\}_{i=1}^N \sim \mathbb{P}(\Gamma)$. On each task $\tau_i$, we sample the trajectories of the meta-policy $\pi_{\phi_t}$ as $B_{\tau_i}$, and evaluate state-action value function $Q_{\tau_i}^{\pi_{\phi_t}}(\cdot,\cdot)$ for each $\tau_i$. Next, since the number of the sampling state-action pairs in $B_{\tau_i}$ is limited, if we directly use the sampling average to approximate the expectation in (2), the approximation error will be very large when $\pi_\phi(a|s)$ is small. Therefore, we solve the following optimization problem as the within-task algorithm instead of (2):

$$\pi_{\theta'_\tau} = \mathcal{A}lg(\lambda, \phi_t, \tau) = \arg\min_\theta \frac{1}{|B_\tau|} \sum_{(a,s) \in B_\tau} h\left( \frac{\pi_\theta(a|s)}{\pi_\phi(a|s)} \right) Q_\tau^{\pi_\phi}(s,a) - \lambda D_\tau^2(\pi_\phi, \pi_\theta), \quad (7)$$

where $h(x) = \frac{2}{1+e^{-2(x-1)}}$. The function $h$ avoids the term $\frac{\pi_\theta(a|s)}{\pi_\phi(a|s)}$ is optimized to very large. We use Adam [32] to solve the problem in (7). Next, the computation of the gradients of the meta-objectives shown in Proposition 2 is time-consuming, since the computation complexity of the term

---

**Algorithm 2** Practical Algorithm of BO-MRL

---

**Require:** Regularization weight $\lambda > 0$; initial meta-parameter $\phi_0$; learning rate $\alpha$.

1: **for** $t = 1, \cdots, T$ **do**
2:     Sample a batch of tasks $\{\tau_i\}_{i=1}^N \sim \mathbb{P}(\Gamma)$ with the MDP $\mathcal{M}_{\tau_i}$ i.i.d.
3:     On each task $\tau_i$, sample the trajectories of the meta-policy $\pi_{\phi_t}$ as $B_{\tau_i}$.
4:     Evaluate the state-action value function $Q_{\tau_i}^{\pi_{\phi_t}}(\cdot, \cdot)$ for each $\tau_i$.
5:     For each task $\tau_i$, compute the task-specific policy $\pi_{\theta'_{\tau_i}}$ by solving $\mathcal{A}lg(\lambda, \phi_t, \tau_i)$ defined in (7) by Adam.
6:     Compute $\nabla_\phi J_{\tau_i}(\pi_{\theta'_{\tau_i}})$ in (8) by conjugate gradient method
7:     Update meta-parameter by the TRPO with the gradient $\frac{1}{N}\sum_i \nabla_\phi J_{\tau_i}(\pi_{\theta'_{\tau_i}})$ and the sampling trajectories $\{B_{\tau_i}\}_{i=1}^N$.
8: **end for**
9: Return $\phi_T$

---

$-\frac{\nabla_\theta \pi_\theta(a|s)}{\lambda \pi_\phi(a|s)}\nabla_\phi^\top Q_\tau^{\pi_\phi}(s,a)$ is very high. So, we omit the term, and compute $\nabla_\phi J_\tau(\pi_{\theta'_\tau})$ as

$$\frac{1}{1-\gamma}\nabla_\phi \theta'_\tau \cdot \mathbb{E}_{\substack{s\sim\nu_\tau^{\pi_{\theta'_\tau}} \\ a\sim\pi_{\theta'_\tau}(\cdot|s)}}\left[\frac{\nabla_{\theta'_\tau}\pi_{\theta'_\tau}(a|s)}{\pi_{\theta'_\tau}(a|s)}Q_\tau^{\pi_{\theta'_\tau}}(s,a)\right], \tag{8}$$

where

$$\nabla_\phi^\top \theta'_\tau \approx - \mathbb{E}_{\substack{s\sim\nu_\tau^{\pi_\phi} \\ a\sim\pi_\phi(\cdot|s)}}\left[\nabla_\theta^2 d^2(\pi_\phi(\cdot|s),\pi_\theta(\cdot|s)) - \frac{\nabla_\theta^2 \pi_\theta(a|s)}{\lambda\pi_\phi(a|s)}Q_\tau^{\pi_\phi}(s,a)\right]^{-1}$$

$$\mathbb{E}_{\substack{s\sim\nu_\tau^{\pi_\phi} \\ a\sim\pi_\phi(\cdot|s)}}\left[\nabla_\phi^\top\nabla_\theta d^2(\pi_\phi(\cdot|s),\pi_\theta(\cdot|s))\right]|_{\theta=\theta'_\tau}.$$

Finally, since the gradient-based approach is not stable in RL problems, we optimize meta-parameter by the TRPO with the gradient $\frac{1}{N}\sum_i \nabla_\phi J_{\tau_i}(\pi_{\theta'_{\tau_i}})$ and the sampling trajectories $\{B_{\tau_i}\}_{i=1}^N$, similar to [15].

## E   Discussion about computational complexity of hyper-gradient

In Algorithms 1 and 2, we compute the inverse of the Hessian matrix when computing the hyper-gradient by Proposition 2 and (8). The computation of the inverse of the Hessian matrix is not time-consuming and does not increase the processing time much. Here are the two reasons.

First, we apply the conjugate gradient algorithm to compute the inverse of the Hessian and its computation complexity is not high. According to our experiment of Half-cheetah, the computation time of the hyper-gradient with the inverse of Hessian for a three-layer neural network is about $0.3$ second in each meta-parameter update, where we use only the CPU to compute the hyper-gradient. This approach has demonstrated high efficiency across a wide range of applications, including several widely used RL algorithms, such as TRPO [51] and CPO [1], which compute the inverse of the Hessian in each policy update iteration. The detail is shown in Appendix C of [51]. They usually compute thousands times of the Hessian inverse for a single RL task. In the simplest meta-RL method, MAML [15], the authors use the TRPO to update the meta-parameter, as shown in Section 5.3 of [15], the inverse of the Hessian is also computed. Therefore, the computational complexity of the hyper-gradient in our proposed method is comparable to many existing RL and meta-RL approaches, which are shown efficient.

Second, the biggest computational bottleneck in the meta-RL framework is not the hyper-gradient computation. According to our experiment, the percentage of the computation time in the meta-parameter update, including the computation time of the hyper-gradient computation, is less than $5\%$, where we use only the CPU to compute the hyper-gradient. The percentage of computation time in the data collection and the Q value computation by Monte-Carlo sampling is more than $70\%$, although the state-action data points are collected in the MDP simulator Gym and the data collection is very fast. In real-world applications, the state-action data points are even harder to collect and

data collection consumes a longer time. Therefore, the computational time of the hyper-gradient computation has a relatively small impact on the mete-RL framework.

## F Data sampling complexity and computational complexity of one-time policy adaptation

The one-time policy adaptation in our algorithm is defined as solving the optimal solution of the optimization problem in (1) or (2) by multiple optimization iterations. The definition of the one-time policy adaptation follows many widely used RL algorithms, such as TRPO [51] and CPO [1], which evaluate the Q-values for the current policy and solve the optimal solution for an optimization problem to obtain the next policy in each policy optimization iteration. For example, TRPO solves the optimization problem in (14) of [51] in each iteration.

In the one-time policy adaptation, we only need to evaluate the Q-function for one policy $\pi_\phi$ by Monte-Carlo sampling, which requires the agent to explore the MDP using one policy $\pi_\phi$, then solve the optimization problem in (1) or (2) by multiple optimization iterations with the fixed Q-function. The data sampling complexity is exactly the same as the one-step gradient descent in MAML, which uses Monte-Carlo sampling to evaluate the Q-function and compute the policy gradient based on the Q-function.

The multiple optimization steps in the one-time policy adaptation are different from the multi-step policy gradient update in MAML. In our algorithm, the multiple optimization steps in a one-time policy adaptation only need to evaluate the Q-function for one policy $\pi_\phi$, which requires the agent to explore the MDP using only $\pi_\phi$. In MAML, the Q-function for a new policy needs to be evaluated in each policy gradient update, and then multiple Q-functions are evaluated for multiple policies, which requires the agent to explore the MDP using multiple policies. Instead, the one-time policy adaptation in our algorithm corresponds to a one-step policy gradient update in MAML, as they use the same number of data points.

Moreover, we would like to claim that the computation complexity for the one-time policy adaptation in our algorithm and that of the one-step policy gradient update in MAML is comparable, although our algorithm requires multiple optimization iterations. As mentioned in Appendix E, the computation time in the data collection and the Q value computation takes more than $70\%$ of total computation time, which is much longer than other parts of the algorithm, including the multiple optimization iterations in police adaptation ($15\%$ of total computation time). This happens although the state-action data points are collected in the MDP simulator Gym and the data collection is very fast. In real-world applications, the state-action data points are even harder to collect and the consuming time of data collection is much longer. Therefore, the computational time of the multiple optimization iterations has a relatively small impact on the mete-RL framework. Therefore, the computation time of our algorithm and that of MAML is comparable.

From the statement in the above paragraphs, both the data sampling complexity and computational complexity of the one-time policy adaptation in our algorithm and the one-step policy gradient update in MAML are similar. Thus, we define solving the optimal solution of the optimization problem in (1) or (2) as a single policy adaptation step.

## G Algorithm details with the first-order approximation

As we mentioned in Section 4, we can approximate the first term $\mathbb{E}_{s\sim\nu_\tau^{\pi_\phi},a\sim\pi_\phi(\cdot|s)}[\frac{\pi_\theta(a|s)}{\pi_\phi(a|s)}Q_\tau^{\pi_\phi}(s,a)]$ in (2) by its first-order approximation as the within-task algorithm, similar to the implementations in TRPO [51] and PPO [52]. In particular, the within-task algorithm is reduced to the following formulation,

$$\pi_{\theta_\tau'} = \mathcal{A}lg(\pi_\phi, \lambda, \tau) \triangleq \underset{\pi_\theta}{\operatorname{argmin}} -\frac{1}{\lambda}G(\phi)^\top\theta + D_\tau^2(\pi_\phi, \pi_\theta). \tag{9}$$

Here, we use the first-order approximation to replace the first term of (2). In particular, $G(\phi)^\top(\theta - \phi)$ is the first order approximation of $\mathbb{E}_{s\sim\nu_\tau^{\pi_\phi},a\sim\pi_\phi(\cdot|s)}[\frac{\pi_\theta(a|s)}{\pi_\phi(a|s)}Q_\tau^{\pi_\phi}(s,a)]$, where $G(\phi) = \nabla_\theta\mathbb{E}_{s\sim\nu_\tau^{\pi_\phi},a\sim\pi_\phi(\cdot|s)}[\frac{\nabla_\theta\pi_\theta(a|s)}{\pi_\phi(a|s)}Q_\tau^{\pi_\phi}(s,a)]|_{\theta=\phi} = \mathbb{E}_{s\sim\nu_\tau^{\pi_\phi},a\sim\pi_\phi(\cdot|s)}[\frac{\nabla_\phi\pi_\phi(a|s)}{\pi_\phi(a|s)}Q_\tau^{\pi_\phi}(s,a)]$. Under the simplified within-task algorithm $\mathcal{A}lg$, the hypergradient of the meta-objective function

$\nabla_\phi J_\tau(\pi_{\theta'_\tau})$ can be computed by

$$\nabla_\phi J_\tau(\pi_{\theta'_\tau}) = \frac{1}{1-\gamma} \nabla_\phi \theta'_\tau \cdot \mathop{\mathbb{E}}_{\substack{s \sim \nu_\tau^{\pi_{\theta'_\tau}} \\ a \sim \pi_{\theta'_\tau}(\cdot|s)}} \left[ \frac{\nabla_{\theta'_\tau} \pi_{\theta'_\tau}(a|s)}{\pi_{\theta'_\tau}(a|s)} Q_\tau^{\pi_{\theta'_\tau}}(s,a) \right],$$

where

$$\nabla_\phi^\top \theta'_\tau = \nabla_{\theta'_\tau}^2 D_\tau^2(\pi_\phi, \pi_{\theta'_\tau})^{-1} \Big( \mathop{\mathbb{E}}_{\substack{s \sim \nu_\tau^{\pi_\phi} \\ a \sim \pi_\phi(\cdot|s)}} \Big[ \frac{1}{\lambda} \frac{\nabla_\phi \pi_\phi(a|s)}{\pi_\phi(a|s)} \nabla_\phi^\top Q_\tau^{\pi_\phi}(s,a) +$$
$$\frac{1}{\lambda} \frac{\nabla_\phi^2 \pi_\phi(a|s)}{\pi_\phi(a|s)} Q_\tau^{\pi_\phi}(s,a) \Big] - \nabla_\phi^\top \nabla_{\theta'_\tau} D_\tau^2(\pi_\phi, \pi_{\theta'_\tau}) \Big). \tag{10}$$

The computation of $\nabla_\phi^\top \theta'_\tau$ is derived in Section J.3.

## H    Connection between the proposed algorithm and MAML

As we claim in Section 4, when we approximate the first term $\mathbb{E}_{s \sim \nu_\tau^{\pi_\phi}, a \sim \pi_\phi(\cdot|s)}[\frac{\pi_\theta(a|s)}{\pi_\phi(a|s)} Q_\tau^{\pi_\phi}(s,a)]$ in (2) by its first-order approximation and also select $D_\tau = D_{\tau,3}$, the within-task algorithm (2) is reduced to the policy gradient ascent. In particular, the term $\mathbb{E}_{s \sim \nu_\tau^{\pi_\phi}, a \sim \pi_\phi(\cdot|s)}[\frac{\pi_\theta(a|s)}{\pi_\phi(a|s)} Q_\tau^{\pi_\phi}(s,a)]$ is approximated by $(\theta - \phi)^\top \mathbb{E}_{s \sim \nu_\tau^{\pi_\phi}, a \sim \pi_\phi(\cdot|s)}[\frac{\nabla \pi_\phi(a|s)}{\pi_\phi(a|s)} Q_\tau^{\pi_\phi}(s,a)]$, then the within-task algorithm $Alg(\pi_\phi, \lambda, \tau)$ becomes to

$$\theta'_\tau = Alg(\pi_\phi, \lambda, \tau) \triangleq \mathop{\arg\max}_\theta -\lambda \|\theta - \phi\|^2 + \theta^\top \cdot \mathop{\mathbb{E}}_{\substack{s \sim \nu_\tau^{\pi_\phi} \\ a \sim \pi_\phi(\cdot|s)}} \left[ \frac{\nabla_\phi \pi_\phi(a|s)}{\pi_\phi(a|s)} Q_\tau^{\pi_\phi}(s,a) \right]. \tag{11}$$

Solve the optimization problem, we have

$$\theta'_\tau = \phi + \frac{1}{\lambda} \mathop{\mathbb{E}}_{\substack{s \sim \nu_\tau^{\pi_\phi} \\ a \sim \pi_\phi(\cdot|s)}} \left[ \frac{\nabla_\phi \pi_\phi(a|s)}{\pi_\phi(a|s)} Q_\tau^{\pi_\phi}(s,a) \right] = \phi + \frac{1-\gamma}{\lambda} \nabla_\phi J_\tau(\phi),$$

which is policy gradient ascent. Thus, when we select (11) as the within-task algorithm, the meta-algorithm (3) is reduced to the algorithm that can learn the initialization parameter for the policy gradient ascent.

As shown in [15], MAML also learns the initialization parameter $\phi$ for the policy gradient ascent. However, MAML ignores that the sampled trajectories with policy $\pi_\phi$ also depend on $\phi$. Specifically, MAML first uses the sampled trajectories to approximate $Q_\tau^{\pi_\phi}(s,a)$ by (Monte Carlo sampling on the REINFORCE algorithm), then computes the policy gradient and does one step of gradient ascent for the task-specific adaptation. Next, it computes $\nabla_\phi J_\tau(\theta'_\tau)$ to update the meta-parameter $\phi$. When it computes $\nabla_\phi \theta'_\tau$, it treats $Q_\tau^{\pi_\phi}(s,a)$ as a given data point that is independent with $\phi$, and then ignore the $\nabla_\phi Q_\tau^{\pi_\phi}(s,a)$. In contrast, our reduced meta-algorithm takes it into account and provides a precise formulation to learn the meta-initialization for the policy gradient algorithm.

Since the proposed meta-RL framework can include MAML as a special case, our analysis in Section 5 also provides the theoretical motivation for MAML.

## Analysis and Proof

## I    Auxiliary Results

**Lemma 3** (Policy gradient [56, 2]). *Let $\pi_\theta$ be the parameterized policy with the parameter $\theta$. It holds that*

$$\nabla_\theta J_\tau(\pi_\theta) = \frac{1}{1-\gamma} \mathbb{E}_{s \sim \nu_\tau^{\pi_\theta}, a \sim \pi_\theta(\cdot|s)} \left[ \nabla_\theta \ln \pi_\theta(a|s) Q_\tau^{\pi_\theta}(s,a) \right]$$

$$= \frac{1}{1-\gamma} \mathbb{E}_{s \sim \nu_\tau^{\pi_\theta}, a \sim \pi_\theta(\cdot|s)} \left[ \nabla_\theta \ln \pi_\theta(a|s) A_\tau^{\pi_\theta}(s,a) \right].$$

**Lemma 4** (Policy gradient of the softmax policy). *Consider the softmax policy $\hat{\pi}_\theta$ parameterized by $\theta$. For a discrete state-action space and the tabular policy, $\hat{\pi}_\theta(a|s) = \frac{\exp(\theta(s,a))}{\sum_{a'\in\mathcal{A}}\exp(\theta(s,a'))}$, $\forall(s,a) \in \mathcal{S}\times\mathcal{A}$. It holds that*

$$\nabla_{\theta(s,\cdot)} J_\tau(\hat{\pi}_\theta) = \frac{1}{1-\gamma}\nu_\tau^{\hat{\pi}_\theta}(s)\cdot\hat{\pi}_\theta(\cdot|s)\odot A_\tau^{\hat{\pi}_\theta}(s,\cdot), \tag{12}$$

*where $\odot$ is the element-wise product, $\theta(s,\cdot)$ is the vector which includes $\theta(s,a)$ for all $a\in\mathcal{A}$ as the elements, $A_\tau^{\hat{\pi}_\theta}(s,\cdot)$ is the vector which includes $A_\tau^{\hat{\pi}_\theta}(s,a)$ for all $a\in\mathcal{A}$ as the elements. Equivalently,*

$$\nabla_{\theta(s,a)} J_\tau(\hat{\pi}_\theta) = \frac{1}{1-\gamma}\nu_\tau^{\hat{\pi}_\theta}(s)\hat{\pi}_\theta(a|s)A_\tau^{\hat{\pi}_\theta}(s,a), \tag{13}$$

*For the softmax policy with function approximation, the policy $\pi_\theta$ is defined by $\pi_\theta(a|s) = \frac{\exp(f_\theta(s,a))}{\int_\mathcal{A}\exp(f_\theta(s,a'))da'}$, $\forall(s,a)\in\mathcal{S}\times\mathcal{A}$. It holds that*

$$\nabla_\theta J_\tau(\hat{\pi}_\theta) = \frac{1}{1-\gamma}\mathbb{E}_{s\sim\nu_\tau^{\hat{\pi}_\theta}, a\sim\hat{\pi}_\theta(\cdot|s)}\left[\nabla_\theta f_\theta(s,a)A_\tau^{\hat{\pi}_\theta}(s,a)\right]. \tag{14}$$

*Proof.* For the discrete state-action space and the tabular policy, (12) is shown in Lemma C.1 of [2]. For the softmax policy with function approximation, from Lemma 3, we have

$$\nabla_\theta J_\tau(\hat{\pi}_\theta) = \frac{1}{1-\gamma}\mathbb{E}_{s\sim\nu_\tau^{\hat{\pi}_\theta}, a\sim\hat{\pi}_\theta(\cdot|s)}\left[\nabla_\theta\ln\hat{\pi}_\theta(a|s)A_\tau^{\hat{\pi}_\theta}(s,a)\right]$$

$$= \frac{1}{1-\gamma}\mathbb{E}_{s\sim\nu_\tau^{\hat{\pi}_\theta}, a\sim\hat{\pi}_\theta(\cdot|s)}\left[\nabla_\theta\ln\left(\frac{\exp(f_\theta(s,a))}{\int_\mathcal{A}\exp(f_\theta(s,a'))da'}\right)A_\tau^{\hat{\pi}_\theta}(s,a)\right]$$

$$= \frac{1}{1-\gamma}\mathbb{E}_{s\sim\nu_\tau^{\hat{\pi}_\theta}, a\sim\hat{\pi}_\theta(\cdot|s)}\left[\nabla_\theta f_\theta(s,a) - \nabla_\theta\ln\left(\int_\mathcal{A}\exp(f_\theta(s,a'))da'\right)A_\tau^{\hat{\pi}_\theta}(s,a)\right]$$

Here, $\nabla_\theta\ln\left(\int_\mathcal{A}\exp(f_\theta(s,a'))da'\right)$ is independent with $a$, then $\nabla_\theta J_\tau(\hat{\pi}_\theta)$

$$= \frac{1}{1-\gamma}\mathbb{E}_{s\sim\nu_\tau^{\hat{\pi}_\theta}, a\sim\hat{\pi}_\theta(\cdot|s)}\left[\nabla_\theta f_\theta(s,a) - \nabla_\theta\ln\left(\int_\mathcal{A}\exp(f_\theta(s,a'))da'\right)A_\tau^{\hat{\pi}_\theta}(s,a)\right]$$

$$= \frac{1}{1-\gamma}\mathbb{E}_{s\sim\nu_\tau^{\hat{\pi}_\theta}, a\sim\hat{\pi}_\theta(\cdot|s)}\left[\nabla_\theta f_\theta(s,a)A_\tau^{\hat{\pi}_\theta}(s,a)\right] -$$

$$\frac{1}{1-\gamma}\mathbb{E}_{s\sim\nu_\tau^{\hat{\pi}_\theta}}\left[\nabla_\theta\ln\left(\int_\mathcal{A}\exp(f_\theta(s,a'))da'\right)\mathbb{E}_{a\sim\hat{\pi}_\theta(\cdot|s)}A_\tau^{\hat{\pi}_\theta}(s,a)\right].$$

Since $\mathbb{E}_{a\sim\hat{\pi}_\theta(\cdot|s)}A_\tau^{\hat{\pi}_\theta}(s,a) = \mathbb{E}_{a\sim\hat{\pi}_\theta(\cdot|s)}[Q_\tau^{\hat{\pi}_\theta}(s,a)] - V_\tau^{\hat{\pi}_\theta}(s) = 0$. Then,

$$\nabla_\theta J_\tau(\hat{\pi}_\theta) = \frac{1}{1-\gamma}\mathbb{E}_{s\sim\nu_\tau^{\hat{\pi}_\theta}, a\sim\hat{\pi}_\theta(\cdot|s)}\left[\nabla_\theta f_\theta(s,a)A_\tau^{\hat{\pi}_\theta}(s,a)\right].$$

$\square$

# J  Proofs of the computation of hypergradient

## J.1  Proofs of Propositions 1

*Proofs of Propositions 1.* Consider the within-task algorithm in discrete space:

$$\mathcal{A}lg(\pi_\phi,\lambda,\tau) = \underset{\pi}{\arg\max}\,\mathbb{E}_{s\sim\nu_\tau^{\pi_\phi}}\left[\sum_{a\in\mathcal{A}}\pi(a|s)Q_\tau^{\pi_\phi}(s,a)\right] - \lambda D_\tau^2(\pi_\phi,\pi)$$

$$= \underset{\pi}{\arg\max}\,\mathbb{E}_{s\sim\nu_\tau^{\pi_\phi}}\left[\sum_{a\in\mathcal{A}}\pi(a|s)Q_\tau^{\pi_\phi}(s,a) - \lambda d^2(\pi_\phi(\cdot|s),\pi(\cdot|s))\right].$$

Here, $d$ can be selected from $d_1$ to $d_3$ defined in Section 3, corresponding to the selection of $D_\tau$ from $D_{\tau,1}$ to $D_{\tau,3}$.

The above optimization problem is formally defined by the following problem,

$$\mathcal{A}lg(\pi_\phi, \lambda, \tau) = \arg\max_{\pi} \mathbb{E}_{s \sim \nu_\tau^{\pi_\phi}} \left[ \sum_{a \in \mathcal{A}} \pi(a|s) Q_\tau^{\pi_\phi}(s, a) - \lambda d^2(\pi_\phi(\cdot|s), \pi(\cdot|s), s) \right],$$

$$\text{subject to } \sum_{a \in \mathcal{A}} \pi(a|s) = 1, \text{ for any } s \in \mathcal{S}. \tag{15}$$

With Assumption 2, the problem is equivalent to that, for any $s \in \mathcal{S}$,

$$\mathcal{A}lg(\pi_\phi, \lambda, \tau)(\cdot|s) = \arg\max_{\pi(\cdot|s)} \sum_{a \in \mathcal{A}} \pi(a|s) Q_\tau^{\pi_\phi}(s, a) - \lambda d^2(\pi_\phi(\cdot|s), \pi(\cdot|s)),$$

$$\text{subject to } \sum_{a \in \mathcal{A}} \pi(a|s) = 1. \tag{16}$$

Consider a $s \in \mathcal{S}$, the Lagrangian of the above maximization problem is

$$-\sum_{a \in \mathcal{A}} \pi(a|s) Q_\tau^{\pi_\phi}(s, a) + \lambda d^2(\pi_\phi(\cdot|s), \pi(\cdot|s)) + \mu(\sum_{a \in \mathcal{A}} \pi(a|s) - 1),$$

where $\mu$ is the Lagrangian multiplier. The optimality condition of $\pi(\cdot|s)$ is that,

$$-Q_\tau^{\pi_\phi}(s, \cdot) + \lambda \nabla_{\pi(\cdot|s)} d^2(\pi_\phi(\cdot|s), \pi(\cdot|s)) + \mu[1, \cdots, 1]^\top = 0.$$

Here, $Q_\tau^{\pi_\phi}(s, \cdot)$ denotes a vector include $Q_\tau^{\pi_\phi}(s, a)$ for each $a \in \mathcal{A}$, and $\pi(\cdot|s)$ denotes a vector include $\pi(a|s)$ for each $a \in \mathcal{A}$.

Then, we have

$$-Q_\tau^{\pi_\phi}(s, \cdot) + \lambda \nabla_{\pi(\cdot|s)} d^2(\pi_\phi(\cdot|s), \pi(\cdot|s))|_{\pi = \mathcal{A}lg(\pi_\phi, \lambda, \tau)} + \mu[1, \cdots, 1]^\top = 0. \tag{17}$$

Note that the optimization problem (15) depends on $\phi$, and $\pi = \mathcal{A}lg(\pi_\phi, \lambda, \tau)$ is a function of $\phi$, we have

$$-Q_\tau^{\pi_\phi}(s, \cdot) + \lambda \nabla_{\pi(\cdot|s)} d^2(\pi_\phi(\cdot|s), \pi(\cdot|s))|_{\pi = \mathcal{A}lg(\pi_\phi, \lambda, \tau)} + \mu(\phi)[1, \cdots, 1]^\top = 0,$$

i.e., $\mu$ is a function of $\phi$.

Also, we have

$$\mu(\phi)(\sum_{a \in \mathcal{A}} \mathcal{A}lg(\pi_\phi, \lambda, \tau)(a|s) - 1) = 0. \tag{18}$$

With (17) and (18), we can compute $\nabla_\phi \mathcal{A}lg(\pi_\phi, \lambda, \tau)$, where $\mathcal{A}lg(\pi_\phi, \lambda, \tau)$ is continuously differentiable as shown in [64]. We do derivative of (17) and (18) with respect to $\phi$, we have $[\nabla_\phi \mathcal{A}lg(\pi_\phi, \lambda, \tau), \nabla_\phi \mu(\phi)]^\top =$

$$-\begin{bmatrix} \lambda \nabla_{\pi(\cdot|s)}^2 d^2(\pi_\phi(\cdot|s), \pi(\cdot|s)) & \mathbf{1} \\ \mathbf{1}^\top & 0 \end{bmatrix}^{-1} \begin{bmatrix} -\nabla_\phi^\top Q_\tau^{\pi_\phi}(s, \cdot) + \lambda \nabla_\phi^\top \nabla_{\pi(\cdot|s)} d^2(\pi_\phi(\cdot|s), \pi(\cdot|s)) \\ 0 \end{bmatrix}$$

where $\pi = \mathcal{A}lg(\pi_\phi, \lambda, \tau)$.

Solve the equation, we have

$$\nabla_\phi^\top \mathcal{A}lg(\pi_\phi, \lambda, \tau)(\cdot|s) = \left( M(s)^{-1} - \frac{M(s)^{-1} \mathbf{1} \, \mathbf{1}^\top M(s)^{-1}}{\mathbf{1}^\top M(s)^{-1} \mathbf{1}} \right)$$

$$\left( \nabla_\phi^\top Q_\tau^{\pi_\phi}(s, \cdot) - \lambda \nabla_\phi^\top \nabla_{\pi(\cdot|s)} d^2(\pi_\phi(\cdot|s), \pi(\cdot|s)) \right), \tag{19}$$

where $M(s) = \lambda \nabla_{\pi(\cdot|s)}^2 d^2(\pi_\phi(\cdot|s), \pi(\cdot|s))$. It is easy to show that $\nabla_{\pi(\cdot|s)}^2 d^2(\pi_\phi(\cdot|s), \pi(\cdot|s))$ is non-singular for any $\phi$ for any selected $d = d_1$, $d = d_2$, or $d = d_3$.

From the policy gradient theorem in Lemma 3,

$$\nabla_\phi J_\tau(\pi_{\theta'_\tau}) = \frac{1}{1-\gamma}\mathbb{E}_{s\sim\nu_\tau^{\pi_{\theta'_\tau}},a\sim\pi_{\theta'_\tau}(\cdot|s)}[\nabla_\phi \ln \pi_{\theta'_\tau}(a|s)A_\tau^{\pi_{\theta'_\tau}}(s,a)]|_{\pi_{\theta'_\tau}=\mathcal{A}lg(\pi_\phi,\lambda,\tau)}$$

$$= \frac{1}{1-\gamma}\mathbb{E}_{s\sim\nu_\tau^{\pi_{\theta'_\tau}},a\sim\pi_{\theta'_\tau}(\cdot|s)}\left[\frac{\nabla_\phi\pi_{\theta'_\tau}(a|s)}{\pi_{\theta'_\tau}(a|s)}A_\tau^{\pi_{\theta'_\tau}}(s,a)\right]|_{\pi_{\theta'_\tau}=\mathcal{A}lg(\pi_\phi,\lambda,\tau)},$$

$$= \frac{1}{1-\gamma}\mathbb{E}_{s\sim\nu_\tau^{\pi_{\theta'_\tau}}}\left[\sum_{a\in\mathcal{A}}\nabla_\phi\pi_{\theta'_\tau}(a|s)A_\tau^{\pi_{\theta'_\tau}}(s,a)\right]|_{\pi_{\theta'_\tau}=\mathcal{A}lg(\pi_\phi,\lambda,\tau)}.$$

where $\nabla_\phi\pi_{\theta'_\tau}(\cdot|s) = \nabla_\phi\mathcal{A}lg(\pi_\phi,\lambda,\tau)(\cdot|s)$ is shown in (19).

$\square$

## J.2 Proofs of Propositions 2

*Proofs of Propositions 2.* First, we have

$$\nabla_\phi J_\tau(\pi_{\theta'_\tau}) = \nabla_\phi\theta'_\tau\nabla_{\theta'_\tau}J_\tau(\pi_{\theta'_\tau})$$

From the policy gradient theorem in Lemma 3,

$$\nabla_\phi J_\tau(\pi_{\theta'_\tau}) = \frac{1}{1-\gamma}\nabla_\phi\theta'_\tau{}^\top\mathbb{E}_{s\sim\nu_\tau^{\pi_{\theta'_\tau}},a\sim\pi_{\theta'_\tau}(\cdot|s)}\left[\nabla_{\theta'_\tau}\ln\pi_{\theta'_\tau}(a|s)A_\tau^{\pi_{\theta'_\tau}}(s,a)\right]|_{\theta'_\tau=\mathcal{A}lg(\pi_\phi,\lambda,\tau)}.$$

We have

$$\nabla_\phi J_\tau(\pi_{\theta'_\tau}) = \frac{1}{1-\gamma}\nabla_\phi\theta'_\tau{}^\top\mathbb{E}_{s\sim\nu_\tau^{\pi_{\theta'_\tau}},a\sim\pi_{\theta'_\tau}(\cdot|s)}\left[\frac{\nabla_{\theta'_\tau}\pi_{\theta'_\tau}(a|s)}{\pi_{\theta'_\tau}(a|s)}A_\tau^{\pi_{\theta'_\tau}}(s,a)\right]|_{\theta'_\tau=\mathcal{A}lg(\pi_\phi,\lambda,\tau)}.$$

Next, we compute $\nabla_\phi\theta'_\tau$, where

$$\theta'_\tau = \mathcal{A}lg(\pi_\phi,\lambda,\tau) \triangleq \underset{\theta}{\arg\max}\, \mathbb{E}_{s\sim\nu_\tau^{\pi_\phi},a\sim\pi_\phi(\cdot|s)}\left[\frac{\pi_\theta(a|s)}{\pi_\phi(a|s)}Q_\tau^{\pi_\phi}(s,a)\right] - \lambda D_\tau^2(\pi_\phi,\pi_\theta).$$

The optimization problem is equivalent to

$$\theta'_\tau \triangleq \underset{\theta}{\arg\max}\, \mathbb{E}_{s\sim\nu_\tau^{\pi_\phi}}\left[\int_{\mathcal{A}}\pi_\theta(a|s)Q_\tau^{\pi_\phi}(s,a)da - \lambda d^2(\pi_\phi(\cdot|s),\pi_\theta(\cdot|s))\right]$$

$$= \underset{\theta}{\arg\min}\, \mathbb{E}_{s\sim\nu_\tau^{\pi_\phi}}\left[-\int_{\mathcal{A}}\pi_\theta(a|s)Q_\tau^{\pi_\phi}(s,a)da + \lambda d^2(\pi_\phi(\cdot|s),\pi_\theta(\cdot|s))\right]$$

$$= \underset{\theta}{\arg\min}\, \sum_{s\in\mathcal{S}}\nu_\tau^{\pi_\phi}(s)\left(-\int_{\mathcal{A}}\pi_\theta(a|s)Q_\tau^{\pi_\phi}(s,a)da + \lambda d^2(\pi_\phi(\cdot|s),\pi_\theta(\cdot|s))\right).$$

Similar to the derivation from (15) to (16) with Assumption 2, we have that, when $\theta = \mathcal{A}lg(\pi_\phi,\lambda,\tau)$,

$$\nabla_\theta\left(-\int_{\mathcal{A}}\pi_\theta(a|s)Q_\tau^{\pi_\phi}(s,a)da + \lambda d^2(\pi_\phi(\cdot|s),\pi_\theta(\cdot|s))\right) = 0.$$

Then, we have

$$\sum_{s\in\mathcal{S}}\nabla_\phi\nu_\tau^{\pi_\phi}(s)\nabla_\theta\left(-\int_{\mathcal{A}}\pi_\theta(a|s)Q_\tau^{\pi_\phi}(s,a)da + \lambda d^2(\pi_\phi(\cdot|s),\pi_\theta(\cdot|s))\right) = 0.$$

By using implicit differentiation, if the matrix $\mathbb{E}_{s\sim\nu_\tau^{\pi_\phi}}\left[-\int_{\mathcal{A}}\nabla_\theta^2\pi_\theta(a|s)Q_\tau^{\pi_\phi}(s,a)da + \lambda\nabla_\theta^2 d^2(\pi_\phi(\cdot|s),\pi_\theta(\cdot|s))\right]$
is invertible, i.e., $\mathbb{E}_{s\sim\nu_\tau^{\pi_\phi}}\left[-\int_{\mathcal{A}}\pi_\theta(a|s)Q_\tau^{\pi_\phi}(s,a)da + \lambda d^2(\pi_\phi(\cdot|s),\pi_\theta(\cdot|s))\right]$ is strongly convex at
$\theta = \theta'_\tau$, we have

$$\nabla_\phi^\top\theta'_\tau = -\left(\mathbb{E}_{s\sim\nu_\tau^{\pi_\phi}}\left[-\int_{\mathcal{A}}\nabla_\theta^2\pi_\theta(a|s)Q_\tau^{\pi_\phi}(s,a)da + \lambda\nabla_\theta^2 d^2(\pi_\phi(\cdot|s),\pi_\theta(\cdot|s))\right]\right)^{-1}$$

$$\left(-\mathbb{E}_{s\sim\nu_\tau^{\pi_\phi}}\left[\int_{\mathcal{A}}\nabla_\theta\pi_\theta(a|s)\nabla_\phi^\top Q_\tau^{\pi_\phi}(s,a)da\right] + \mathbb{E}_{s\sim\nu_\tau^{\pi_\phi}}\left[\lambda\nabla_\phi^\top\nabla_\theta d^2(\pi_\phi(\cdot|s),\pi_\theta(\cdot|s))\right] + \right.$$

$$\left.\sum_{s\in\mathcal{S}}\nabla_\phi\nu_\tau^{\pi_\phi}(s)\nabla_\theta\left(-\int_{\mathcal{A}}\pi_\theta(a|s)Q_\tau^{\pi_\phi}(s,a)da + \lambda d^2(\pi_\phi(\cdot|s),\pi_\theta(\cdot|s))\right)\right)|_{\theta=\theta'_\tau},$$

This is equivalent to

$$\nabla_\phi^\top \theta_\tau' = -\left(\mathbb{E}_{s\sim\nu_\tau^{\pi_\phi},a\sim\pi_\phi(\cdot|s)}\left[-\frac{\nabla_\theta^2 \pi_\theta(a|s)}{\pi_\phi(a|s)}Q_\tau^{\pi_\phi}(s,a) + \lambda\nabla_\theta^2 d^2(\pi_\phi(\cdot|s),\pi_\theta(\cdot|s))\right]\right)^{-1}$$

$$\mathbb{E}_{s\sim\nu_\tau^{\pi_\phi},a\sim\pi_\phi(\cdot|s)}\left[-\frac{\nabla_\theta\pi_\theta(a|s)}{\pi_\phi(a|s)}\nabla_\phi^\top Q_\tau^{\pi_\phi}(s,a) + \lambda\nabla_\phi^\top\nabla_\theta d^2(\pi_\phi(\cdot|s),\pi_\theta(\cdot|s))\right]\big|_{\theta=\theta_\tau'}.$$

□

### J.3 Proofs of hypergradient of the algorithm in Section G

*Deviation of (10).* As $\theta_\tau' = \underset{\theta}{\operatorname{argmin}} -\frac{1}{\lambda}\theta^\top\mathbb{E}_{s\sim\nu_\tau^{\pi_\phi},a\sim\pi_\phi(\cdot|s)}[\frac{\nabla_\phi\pi_\phi(a|s)}{\pi_\phi(a|s)}A_\tau^{\pi_\phi}(s,a)] + D_\tau^2(\pi_\phi,\pi_\theta)$, by the implicit differentiation theorem in bilevel optimization analysis,

$$\nabla_\phi^\top\theta_\tau' = -\nabla_\theta^2\left[-\frac{1}{\lambda}\theta^\top\mathbb{E}_{s\sim\nu_\tau^{\pi_\phi},a\sim\pi_\phi(\cdot|s)}[\frac{\nabla_\phi\pi_\phi(a|s)}{\pi_\phi(a|s)}Q_\tau^{\pi_\phi}(s,a)] + D_\tau^2(\pi_\phi,\pi_\theta)\right]^{-1}$$

$$\nabla_\phi\nabla_\theta\left[-\frac{1}{\lambda}\theta^\top\mathbb{E}_{s\sim\nu_\tau^{\pi_\phi},a\sim\pi_\phi(\cdot|s)}[\frac{\nabla_\phi\pi_\phi(a|s)}{\pi_\phi(a|s)}Q_\tau^{\pi_\phi}(s,a)] + D_\tau^2(\pi_\phi,\pi_\theta)\right]\big|_{\theta=\theta_\tau'}$$

Also, we have

$$\nabla_\theta^2(\frac{1}{\lambda}\theta^\top\mathbb{E}_{s\sim\nu_\tau^{\pi_\phi},a\sim\pi_\phi(\cdot|s)}[\frac{\nabla_\phi\pi_\phi(a|s)}{\pi_\phi(a|s)}Q_\tau^{\pi_\phi}(s,a)]) = 0,$$

and

$$\nabla_\phi\nabla_\theta(\frac{1}{\lambda}\theta^\top\mathbb{E}_{s\sim\nu_\tau^{\pi_\phi},a\sim\pi_\phi(\cdot|s)}[\frac{\nabla_\phi\pi_\phi(a|s)}{\pi_\phi(a|s)}Q_\tau^{\pi_\phi}(s,a)])$$

$$= \mathbb{E}_{s\sim\nu_\tau^{\pi_\phi}\atop a\sim\pi_\phi(\cdot|s)}[\frac{1}{\lambda}\frac{\nabla_\phi\pi_\phi(a|s)}{\pi_\phi(a|s)}\nabla_\phi^\top Q_\tau^{\pi_\phi}(s,a) + \frac{1}{\lambda}\frac{\nabla_\phi^2\pi_\phi(a|s)}{\pi_\phi(a|s)}Q_\tau^{\pi_\phi}(s,a)].$$

Then, we can get $\nabla_\phi^\top\theta_\tau'$. □

## K  Proofs of convergence when $D_\tau = D_{\tau,1}$

### K.1  Gradients of $\nabla_\phi J_\tau(\pi_{\theta_\tau'})$ when $D_\tau = D_{\tau,1}$

From Proposition 1,

$$\nabla_\phi J_\tau(\pi_{\theta_\tau'}) = \frac{1}{1-\gamma}\mathbb{E}_{s\sim\nu_\tau^{\pi_{\theta_\tau'}}}\left[\sum_{a\in\mathcal{A}}\nabla_\phi\pi_{\theta_\tau'}(a|s)A_\tau^{\pi_{\theta_\tau'}}(s,a)\right] \qquad (20)$$

$$= \frac{1}{1-\gamma}\mathbb{E}_{s\sim\nu_\tau^{\pi_{\theta_\tau'}}}\left[\nabla_\phi\pi_{\theta_\tau'}(\cdot|s)\cdot A_\tau^{\pi_{\theta_\tau'}}(s,\cdot)\right],$$

where

$$\nabla_\phi^\top\pi_{\theta_\tau'}(\cdot|s) = \left(M(s)^{-1} - \frac{M(s)^{-1}\mathbf{1}\,\mathbf{1}^\top M(s)^{-1}}{\mathbf{1}^\top M(s)^{-1}\mathbf{1}}\right)\left(\nabla_\phi^\top Q_\tau^{\pi_\phi}(s,\cdot) - \lambda\nabla_\phi^\top\nabla_{\pi(\cdot|s)}d_1^2(\pi_\phi,\pi,s)\right)\big|_{\pi=\pi_{\theta_\tau'}},$$

where

$$M(s) = \lambda\nabla_{\pi(\cdot|s)}^2 d_1^2(\pi_\phi,\pi,s) = \lambda\begin{bmatrix}\frac{\pi_\phi(a_1|s)}{\pi_{\theta_\tau'}(a_1|s)^2} & & \\ & \ddots & \\ & & \frac{\pi_\phi(a_n|s)}{\pi_{\theta_\tau'}(a_n|s)^2}\end{bmatrix}.$$

Then,

$$M(s)^{-1} = \frac{1}{\lambda}\begin{bmatrix}\frac{\pi_{\theta_\tau'}(a_1|s)^2}{\pi_\phi(a_1|s)} & & \\ & \ddots & \\ & & \frac{\pi_{\theta_\tau'}(a_n|s)^2}{\pi_\phi(a_n|s)}\end{bmatrix}, \qquad (21)$$

and

$$\frac{M(s)^{-1}\mathbf{1}\,\mathbf{1}^\top M(s)^{-1}}{\mathbf{1}^\top M(s)^{-1}\mathbf{1}} = \frac{1}{\lambda \sum_{a\in\mathcal{A}} \frac{\pi_{\theta'_\tau}(a|s)^2}{\pi_\phi(a|s)}} \begin{bmatrix} \frac{\pi_{\theta'_\tau}(a_1|s)^2}{\pi_\phi(a_1|s)} \\ \vdots \\ \frac{\pi_{\theta'_\tau}(a_n|s)^2}{\pi_\phi(a_n|s)} \end{bmatrix} \begin{bmatrix} \frac{\pi_{\theta'_\tau}(a_1|s)^2}{\pi_\phi(a_1|s)} & \cdots & \frac{\pi_{\theta'_\tau}(a_n|s)^2}{\pi_\phi(a_n|s)} \end{bmatrix}.$$

Also,

$$\nabla_\phi^\top \nabla_{\pi(\cdot|s)} d_1^2(\pi_\phi, \pi, s)|_{\pi=\pi_{\theta'_\tau}} = \nabla_\phi^\top \begin{bmatrix} -\frac{\pi_\phi(a_1|s)}{\pi_{\theta'_\tau}(a_1|s)} \\ \vdots \\ -\frac{\pi_\phi(a_n|s)}{\pi_{\theta'_\tau}(a_n|s)} \end{bmatrix} = \begin{bmatrix} -\frac{\nabla_\phi^\top \pi_\phi(a_1|s)}{\pi_{\theta'_\tau}(a_1|s)} \\ \vdots \\ -\frac{\nabla_\phi^\top \pi_\phi(a_n|s)}{\pi_{\theta'_\tau}(a_n|s)} \end{bmatrix}. \tag{22}$$

Then, plugging these equations into (20), we have

$$\nabla_\phi^\top J_\tau(\pi_{\theta'_\tau}) = \frac{1}{1-\gamma}\mathbb{E}_{s\sim\nu_\tau^{\pi_{\theta'_\tau}}}\left[A_\tau^{\pi_{\theta'_\tau}}(s,\cdot)^\top \nabla_\phi^\top \pi_{\theta'_\tau}(\cdot|s)\right],$$

$$= \frac{1}{1-\gamma}\mathbb{E}_{s\sim\nu_\tau^{\pi_{\theta'_\tau}}}\left[A_\tau^{\pi_{\theta'_\tau}}(s,\cdot)^\top \left(M(s)^{-1} - \frac{M(s)^{-1}\mathbf{1}\,\mathbf{1}^\top M(s)^{-1}}{\mathbf{1}^\top M(s)^{-1}\mathbf{1}}\right) \begin{bmatrix} \frac{1}{\lambda}\nabla_\phi^\top Q_\tau^{\pi_\phi}(s,a_1) + \frac{\nabla_\phi^\top \pi_\phi(a_1|s)}{\pi_{\theta'_\tau}(a_1|s)} \\ \vdots \\ \frac{1}{\lambda}\nabla_\phi^\top Q_\tau^{\pi_\phi}(s,a_n) + \frac{\nabla_\phi^\top \pi_\phi(a_n|s)}{\pi_{\theta'_\tau}(a_n|s)} \end{bmatrix}\right]$$

$$= \frac{1}{1-\gamma}\mathbb{E}_{s\sim\nu_\tau^{\pi_{\theta'_\tau}}}\left[\left(\begin{bmatrix}(A_\tau^{\pi_{\theta'_\tau}}(s,a_1) - c_\tau(s))\frac{\pi_{\theta'_\tau}(a_1|s)^2}{\pi_\phi(a_1|s)} & \cdots & (A_\tau^{\pi_{\theta'_\tau}}(s,a_n) - c_\tau(s))\frac{\pi_{\theta'_\tau}(a_n|s)^2}{\pi_\phi(a_n|s)}\end{bmatrix}\right)\right.$$
$$\left.\begin{bmatrix} \frac{1}{\lambda}\nabla_\phi^\top Q_\tau^{\pi_\phi}(s,a_1) + \frac{\nabla_\phi^\top \pi_\phi(a_1|s)}{\pi_{\theta'_\tau}(a_1|s)} \\ \vdots \\ \frac{1}{\lambda}\nabla_\phi^\top Q_\tau^{\pi_\phi}(s,a_n) + \frac{\nabla_\phi^\top \pi_\phi(a_n|s)}{\pi_{\theta'_\tau}(a_n|s)} \end{bmatrix}\right],$$

where

$$c_\tau(s) = \frac{\sum_{a\in\mathcal{A}} A_\tau^{\pi_{\theta'_\tau}}(s,a)\frac{\pi_{\theta'_\tau}(a|s)^2}{\pi_\phi(a|s)}}{\sum_{a\in\mathcal{A}}\frac{\pi_{\theta'_\tau}(a|s)^2}{\pi_\phi(a|s)}}. \tag{23}$$

Then, we simplify the computation of $\nabla_\phi^\top J_\tau(\pi_{\theta'_\tau})$, we have $\nabla_\phi^\top J_\tau(\pi_{\theta'_\tau}) =$

$$\frac{1}{1-\gamma}\mathbb{E}_{s\sim\nu_\tau^{\pi_{\theta'_\tau}}}\left[\sum_{a\in\mathcal{A}}(A_\tau^{\pi_{\theta'_\tau}}(s,a) - c_\tau(s))\frac{\pi_{\theta'_\tau}(a|s)^2}{\pi_\phi(a|s)}(\frac{1}{\lambda}\nabla_\phi^\top Q_\tau^{\pi_\phi}(s,a) + \frac{\nabla_\phi^\top \pi_\phi(a|s)}{\pi_{\theta'_\tau}(a|s)})\right]$$

$$= \frac{1}{1-\gamma}\mathbb{E}_{s\sim\nu_\tau^{\pi_{\theta'_\tau}}}\left[\sum_{a\in\mathcal{A}}\pi_{\theta'_\tau}(a|s)(A_\tau^{\pi_{\theta'_\tau}}(s,a) - c_\tau(s))(\frac{\pi_{\theta'_\tau}(a|s)}{\lambda\pi_\phi(a|s)}\nabla_\phi^\top Q_\tau^{\pi_\phi}(s,a) + \frac{\nabla_\phi^\top \pi_\phi(a|s)}{\pi_\phi(a|s)})\right].$$

When the tabular policy is the softmax policy, we have $\hat{\pi}_\phi(a|s) = \frac{\exp(\phi(s,a))}{\sum_{a'\in\mathcal{A}}\exp(\phi(s,a'))}$, then

$$\frac{\nabla_\phi^\top \hat{\pi}_\phi(a|s)}{\hat{\pi}_\phi(a|s)} = \nabla_\phi^\top \ln \hat{\pi}_\phi(a|s) = \nabla_\phi^\top \phi(s,a) - \nabla_\phi^\top \ln \sum_{a'\in\mathcal{A}}\exp(\phi(s,a'))$$
$$= \mathbf{1}(s,a) - \hat{\pi}_\phi(\cdot|s). \tag{24}$$

Here, $\mathbf{1}(s',a')$ denote the column vector where the element is 1 if $s=s'$ and $a=a'$, otherwise is 0, for each pair $(s,a)\in\mathcal{S}\times\mathcal{A}$; $\hat{\pi}_\phi(\cdot|s')$ is the column vector, where the element is $\hat{\pi}_\phi(a|s')$ if $s=s'$, 0 if $s\neq s'$, for each pair $(s.a)\in\mathcal{S}\times\mathcal{A}$.

So, we have

$$
\begin{aligned}
\nabla_\phi^\top J_\tau(\hat{\pi}_{\theta_\tau'}) =& \frac{1}{1-\gamma} \mathbb{E}_{s\sim\nu_\tau^{\hat{\pi}_{\theta_\tau'}}} \left[ \sum_{a\in\mathcal{A}} \hat{\pi}_{\theta_\tau'}(a|s)(A_\tau^{\hat{\pi}_{\theta_\tau'}}(s,a) - c_\tau(s)) \right. \\
& \left. (\frac{\hat{\pi}_{\theta_\tau'}(a|s)}{\lambda\hat{\pi}_\phi(a|s)} \nabla_\phi^\top Q_\tau^{\hat{\pi}_\phi}(s,a) + \frac{\nabla_\phi^\top \hat{\pi}_\phi(a|s)}{\hat{\pi}_\phi(a|s)}) \right] \\
=& \frac{1}{1-\gamma} \mathbb{E}_{s\sim\nu_\tau^{\hat{\pi}_{\theta_\tau'}}} \left[ \sum_{a\in\mathcal{A}} \hat{\pi}_{\theta_\tau'}(a|s)(A_\tau^{\hat{\pi}_{\theta_\tau'}}(s,a) - c_\tau(s)) \right. \\
& \left. (\frac{\hat{\pi}_{\theta_\tau'}(a|s)}{\lambda\hat{\pi}_\phi(a|s)} \nabla_\phi^\top Q_\tau^{\hat{\pi}_\phi}(s,a) + \mathbf{1}^\top(s,a) - \hat{\pi}_\phi(\cdot|s)^\top) \right] \\
=& \frac{1}{1-\gamma} \mathbb{E}_{s\sim\nu_\tau^{\hat{\pi}_{\theta_\tau'}}, a\sim\hat{\pi}_{\theta_\tau'}} \left[ (A_\tau^{\hat{\pi}_{\theta_\tau'}}(s,a) - c_\tau(s)) \right. \\
& \left. (\frac{\hat{\pi}_{\theta_\tau'}(a|s)}{\lambda\hat{\pi}_\phi(a|s)} \nabla_\phi^\top Q_\tau^{\hat{\pi}_\phi}(s,a) + \mathbf{1}^\top(s,a) - \hat{\pi}_\phi(\cdot|s)^\top) \right].
\end{aligned}
\tag{25}
$$

## K.2  Convergence guarantee when $D_\tau = D_{\tau,1}$

### K.2.1  Auxiliary lemmas

**Lemma 5.** *Suppose that Assumption 2 holds. Let $\pi_{\theta_\tau'} = \mathcal{A}lg(\pi_\phi, \lambda, \tau)$ where $D_\tau = D_{\tau,1}$, for any $s \in \mathcal{S}$ and $a \in \mathcal{A}$, we have*

$$
\frac{\lambda}{\lambda + \max_{s,a}|A_\tau^{\pi_\phi}(s,a)|} \le \frac{\pi_{\theta_\tau'}(a|s)}{\pi_\phi(a|s)} \le \frac{\lambda}{\lambda - \max_{s,a}|A_\tau^{\pi_\phi}(s,a)|}.
$$

*Proof.* From (16), when $D_\tau = D_{\tau,1}$, we have $\pi_{\theta_\tau'} = \mathcal{A}lg(\pi_\phi, \lambda, \tau)$ and

$$
\pi_{\theta_\tau'}(\cdot|s) = \operatorname*{argmax}_{(\cdot|s)} \sum_{a\in\mathcal{A}} \pi(a|s) Q_\tau^{\pi_\phi}(s,a) - \lambda d_1^2(\pi_\phi, \pi, s),
$$
$$
\text{subject to } \sum_{a\in\mathcal{A}} \pi(a|s) = 1.
$$

For any $s \in \mathcal{S}$, the Lagrangian of the above maximization problem is

$$
-\sum_{a\in\mathcal{A}} \pi(a|s) Q_\tau^{\pi_\phi}(s,a) + \lambda d^2(\pi_\phi(\cdot|s), \pi(\cdot|s)) + \mu(s)(\sum_{a\in\mathcal{A}} \pi(a|s) - 1),
$$

where $\mu$ is the Lagrangian multiplier. The optimality condition of $\pi(\cdot|s)$ is that,

$$
-Q_\tau^{\pi_\phi}(s,\cdot) + \lambda \nabla_{\pi(\cdot|s)} d_1^2(\pi_\phi, \pi, s) + \mu(s)[1, \cdots, 1]^\top = 0.
$$

Solve the equation,

$$
-Q_\tau^{\pi_\phi}(s,a) - \lambda \frac{\pi_\phi(a|s)}{\pi_{\theta_\tau'}(a|s)} + \mu(s) = 0.
$$

Let $\mu_1(s) = -V_\tau^{\pi_\phi}(s) + \mu(s)$, we have

$$
-A_\tau^{\pi_\phi}(s,a) - \lambda \frac{\pi_\phi(a|s)}{\pi_{\theta_\tau'}(a|s)} + \mu_1(s) = 0.
\tag{26}
$$

Then,

$$
-\pi_{\theta_\tau'}(a|s) A_\tau^{\pi_\phi}(s,a) - \lambda\pi_\phi(a|s) + \pi_{\theta_\tau'}(a|s)\mu_1(s) = 0.
$$

We derive the summation of all $a \in \mathcal{A}$,

$$
\sum_{a\in\mathcal{A}} -\pi_{\theta_\tau'}(a|s) A_\tau^{\pi_\phi}(s,a) - \lambda + \mu_1(s) = 0.
$$

We have
$$\mu_1(s) = \sum_{a \in \mathcal{A}} \pi_{\theta'_\tau}(a|s) A_\tau^{\pi_\phi}(s,a) + \lambda.$$

From (26), we have
$$\frac{\pi_\phi(a|s)}{\pi_{\theta'_\tau}(a|s)} = \frac{\mu_1(s) - A_\tau^{\pi_\phi}(s,a)}{\lambda}$$
$$= \frac{\lambda + \sum_{a' \in \mathcal{A}} \pi_{\theta'_\tau}(a'|s) A_\tau^{\pi_\phi}(s,a') - A_\tau^{\pi_\phi}(s,a)}{\lambda}$$

So, we have
$$\frac{\pi_{\theta'_\tau}(a|s)}{\pi_\phi(a|s)} = \frac{\lambda}{\lambda + \sum_{a' \in \mathcal{A}} \pi_{\theta'_\tau}(a'|s) A_\tau^{\pi_\phi}(s,a') - A_\tau^{\pi_\phi}(s,a)},$$

then
$$\frac{\lambda}{\lambda + \max_{s,a} |A_\tau^{\pi_\phi}(s,a)|} \leq \frac{\pi_{\theta'_\tau}(a|s)}{\pi_\phi(a|s)} \leq \frac{\lambda}{\lambda - \max_{s,a} |A_\tau^{\pi_\phi}(s,a)|}.$$
$\square$

**Lemma 6.** *Suppose that Assumption 2 holds. Let $\pi_{\theta'_\tau} = \mathcal{A}lg(\pi_\phi, \lambda, \tau)$ where $D_\tau = D_{\tau,1}$, we have*

$$\|\nabla_\phi J_\tau(\pi_{\theta'_\tau})\| \leq \frac{\max_{s,a} |A_\tau^{\pi_{\theta'_\tau}}(s,a)|}{1-\gamma} \left( \frac{\max_{s,a} |A_\tau^{\pi_{\theta'_\tau}}(s,a)|}{\lambda - \max_{s,a} |A_\tau^{\pi_\phi}(s,a)|} \frac{\gamma}{1-\gamma} + 2 \right).$$

*Proof.* As shown in (25),

$$\nabla_\phi^\top J_\tau(\hat{\pi}_{\theta'_\tau}) = \frac{1}{1-\gamma} \mathbb{E}_{s \sim \nu_\tau^{\hat{\pi}_{\theta'_\tau}}} \left[ \sum_{a \in \mathcal{A}} \hat{\pi}_{\theta'_\tau}(a|s) (A_\tau^{\hat{\pi}_{\theta'_\tau}}(s,a) - c_\tau(s)) \right.$$
$$\left. \left( \frac{\hat{\pi}_{\theta'_\tau}(a|s)}{\lambda \hat{\pi}_\phi(a|s)} \nabla_\phi^\top Q_\tau^{\hat{\pi}_\phi}(s,a) + \mathbf{1}^\top(s,a) - \hat{\pi}_\phi(\cdot|s)^\top \right) \right]$$
$$= \frac{1}{1-\gamma} \sum_{s \in \mathcal{S}} \sum_{a \in \mathcal{A}} \nu_\tau^{\hat{\pi}_{\theta'_\tau}}(s) \hat{\pi}_{\theta'_\tau}(a|s) (A_\tau^{\hat{\pi}_{\theta'_\tau}}(s,a) - c_\tau(s))$$
$$\left( \frac{\hat{\pi}_{\theta'_\tau}(a|s)}{\lambda \hat{\pi}_\phi(a|s)} \nabla_\phi^\top Q_\tau^{\hat{\pi}_\phi}(s,a) + \mathbf{1}^\top(s,a) - \hat{\pi}_\phi(\cdot|s)^\top \right).$$

Since $\sum_{s \in \mathcal{S}} \nu_\tau^{\hat{\pi}_{\theta'_\tau}}(s) = 1$ and $\sum_{a \in \mathcal{A}} \hat{\pi}_{\theta'_\tau}(a|s) = 1$ for all $s \in \mathcal{S}$, we have $\|\nabla_\phi J_\tau(\hat{\pi}_{\theta'_\tau})\| \leq$

$$\frac{1}{1-\gamma} \max_{a,s} \|(A_\tau^{\hat{\pi}_{\theta'_\tau}}(s,a) - c_\tau(s)) \left( \frac{\hat{\pi}_{\theta'_\tau}(a|s)}{\lambda \hat{\pi}_\phi(a|s)} \nabla_\phi^\top Q_\tau^{\hat{\pi}_\phi}(s,a) + \mathbf{1}^\top(s,a) - \hat{\pi}_\phi(\cdot|s)^\top \right)\|.$$

From (23), for any $s \in \mathcal{S}$ and $a \in \mathcal{A}$, we have
$$|A_\tau^{\hat{\pi}_{\theta'_\tau}}(s,a) - c_\tau(s)| \leq \max_{s,a} |A_\tau^{\hat{\pi}_{\theta'_\tau}}(s,a)|.$$

Also, for any $s \in \mathcal{S}$ and $a \in \mathcal{A}$,
$$\|\mathbf{1}(s,a) - \hat{\pi}_\phi(\cdot|s))\| \leq 1 + \|\hat{\pi}(\cdot|s)\| \leq 2.$$

From Lemma 5,
$$\frac{\hat{\pi}_{\theta'_\tau}(a|s)}{\lambda \hat{\pi}_\phi(a|s)} \leq \frac{1}{\lambda - \max_{s,a} |A_\tau^{\hat{\pi}_\phi}(s,a)|}.$$

From the computation of $\nabla_\phi Q_\tau^{\hat{\pi}_\phi}(s,a)$ shown in (5) of Appendix C,
$$|\nabla_{\phi(s',a')} Q_\tau^{\hat{\pi}_\phi}(s,a)| = \left| \frac{\gamma}{1-\gamma} \cdot \sigma_{\tau,\hat{\pi}_\phi}^{(s,a)}(s') \hat{\pi}_\phi(a'|s') A_\tau^{\hat{\pi}_\phi}(s,a) \right|$$
$$\leq \frac{\gamma}{1-\gamma} \sigma_{\tau,\hat{\pi}_\phi}^{(s,a)}(s') \hat{\pi}_\phi(a'|s') |\max_{a,s} A_\tau^{\hat{\pi}_\phi}(s,a)|.$$

Also, since $\sum_{a \in \mathcal{A}, s \in \mathcal{S}} \sigma_{\tau, \hat{\pi}_\phi}^{(s,a)}(s') \hat{\pi}_\phi(a'|s') = 1$, we have

$$\|\nabla_\phi Q_\tau^{\hat{\pi}_\phi}(s,a)\| \leq \frac{\gamma}{1-\gamma} \max_{a,s} |A_\tau^{\hat{\pi}_\phi}(s,a)|. \tag{27}$$

Therefore, we have

$$\|\nabla_\phi J_\tau(\hat{\pi}_{\theta_\tau'})\| \leq \frac{\max_{s,a}|A_\tau^{\hat{\pi}_{\theta_\tau'}}(s,a)|}{1-\gamma} \left( \frac{\max_{s,a}|A_\tau^{\hat{\pi}_{\theta_\tau'}}(s,a)|}{\lambda - \max_{s,a}|A_\tau^{\hat{\pi}_\phi}(s,a)|} \frac{\gamma}{1-\gamma} + 2 \right).$$

$\square$

**Lemma 7.** *Suppose that Assumption 2 holds. Let $\hat{\pi}_{\theta_\tau'} = \mathcal{Alg}(\hat{\pi}_\phi, \lambda, \tau)$ where $D_\tau = D_{\tau,1}$, for any $s \in \mathcal{S}$ we have*

$$\sum_{a \in \mathcal{A}} \|\nabla_\phi \hat{\pi}_{\theta_\tau'}(a|s)\| \leq \frac{1}{\lambda - \max_{s,a}|A_\tau^{\hat{\pi}_\phi}(s,a)|} \left( \frac{\gamma \max_{a,s}|A_\tau^{\hat{\pi}_\phi}(s,a)|}{1-\gamma} + 2(\lambda + \max_{a,s}|A_\tau^{\hat{\pi}_\phi}(s,a)|) \right)$$

*and*

$$\sum_{a \in \mathcal{A}} \|\nabla_\phi^2 \hat{\pi}_{\theta_\tau'}(a|s)\| \leq \frac{1}{\lambda - \max_{s,a}|A_\tau^{\hat{\pi}_\phi}(s,a)|} \left( \frac{8r_{max}}{(1-\gamma)^3} \right.$$
$$\left. + \frac{\lambda + \max_{s,a}|A_\tau^{\hat{\pi}_\phi}(s,a)|}{\lambda - \max_{s,a}|A_\tau^{\hat{\pi}_\phi}(s,a)|} \left( \frac{(2-\gamma)\max_{a,s}|A_\tau^{\hat{\pi}_\phi}(s,a)|}{1-\gamma} + 2\lambda + 2 \right) \right).$$

*Proof.* From (19), for any $s \in \mathcal{S}$,

$$\nabla_\phi^\top \hat{\pi}_{\theta_\tau'}(\cdot|s) = \left( M(s)^{-1} - \frac{M(s)^{-1} \mathbf{1} \mathbf{1}^\top M(s)^{-1}}{\mathbf{1}^\top M(s)^{-1} \mathbf{1}} \right) \left( \nabla_\phi^\top Q_\tau^{\hat{\pi}_\phi}(s,\cdot) - \lambda \nabla_\phi^\top \nabla_{\hat{\pi}(\cdot|s)} d_1^2(\hat{\pi}_\phi, \hat{\pi}, s) \right) |_{\hat{\pi}=\hat{\pi}_{\theta_\tau'}}.$$

From the computations of $M(s)^{-1}$, $\nabla_\phi^\top \nabla_{\hat{\pi}(\cdot|s)} d_1^2(\hat{\pi}_\phi, \hat{\pi}, s)|_{\hat{\pi}=\hat{\pi}_{\theta_\tau'}}$, and $\nabla_\phi Q_\tau^{\hat{\pi}_\phi}(s,\cdot)$ in (21) (27), we have

$$\left\| \left( M(s)^{-1} - \frac{M(s)^{-1} \mathbf{1} \mathbf{1}^\top M(s)^{-1}}{\mathbf{1}^\top M(s)^{-1} \mathbf{1}} \right)_j \right\| \leq \frac{\hat{\pi}_{\theta_\tau'}(a|s)}{\lambda} \max_{a,s} \frac{\hat{\pi}_{\theta_\tau'}(a|s)}{\hat{\pi}_\phi(a|s)} \leq \frac{\hat{\pi}_{\theta_\tau'}(a|s)}{\lambda - \max_{s,a}|A_\tau^{\hat{\pi}_\phi}(s,a)|},$$

and

$$\|\nabla_\phi Q_\tau^{\hat{\pi}_\phi}(s,a)\| \leq \max_a \|\nabla_\phi Q_\tau^{\hat{\pi}_\phi}(s,a)\| \leq \frac{\gamma}{1-\gamma} \max_{a,s} |A_\tau^{\hat{\pi}_\phi}(s,a)|.$$

From (22)(24)

$$\|\lambda \nabla_\phi^\top \nabla_{\hat{\pi}(a|s)} d_1^2(\hat{\pi}_\phi, \hat{\pi}, s)\| = \|\lambda(\mathbf{1}(s,a) - \hat{\pi}_\phi(\cdot|s)) \frac{\hat{\pi}_\phi(a|s)}{\hat{\pi}_{\theta_\tau'}(a|s)}\| \leq 2(\lambda + \max_{a,s}|A_\tau^{\hat{\pi}_\phi}(s,a)|).$$

The last inequality comes from Lemma 5. So, we have

$$\|\nabla_\phi \hat{\pi}_{\theta_\tau'}(a|s)\| \leq \frac{\hat{\pi}_{\theta_\tau'}(a|s)}{\lambda - \max_{s,a}|A_\tau^{\hat{\pi}_\phi}(s,a)|} \left( \frac{\gamma \max_{a,s}|A_\tau^{\hat{\pi}_\phi}(s,a)|}{1-\gamma} + 2(\lambda + \max_{a,s}|A_\tau^{\hat{\pi}_\phi}(s,a)|) \right).$$

Therefore,

$$\sum_{a \in \mathcal{A}} \|\nabla_\phi \hat{\pi}_{\theta_\tau'}(a|s)\| \leq \frac{1}{\lambda - \max_{s,a}|A_\tau^{\hat{\pi}_\phi}(s,a)|} \left( \frac{\gamma \max_{a,s}|A_\tau^{\hat{\pi}_\phi}(s,a)|}{1-\gamma} + 2(\lambda + \max_{a,s}|A_\tau^{\hat{\pi}_\phi}(s,a)|) \right).$$

Also, we have

$$\|\nabla_\phi^2 \hat{\pi}_{\theta_\tau'}(a|s)\| \leq \frac{\hat{\pi}_{\theta_\tau'}(a|s)}{\lambda - \max_{s,a}|A_\tau^{\hat{\pi}_\phi}(s,a)|} (\|\nabla_\phi^2 Q_\tau^{\hat{\pi}_\phi}(s,a)\| + \lambda\|\nabla_\phi^2 \nabla_{\hat{\pi}(a|s)} d_1^2(\hat{\pi}_\phi, \hat{\pi}, s)\|).$$

From Lemma D.4 in [2], we have

$$\|\nabla_\phi^2 Q_\tau^{\hat{\pi}_\phi}(s,a)\| \leq \frac{8r_{max}}{(1-\gamma)^3}.$$

Moreover, we have

$$\lambda \|\nabla_\phi^2 \nabla_{\hat{\pi}(a|s)} d_1^2(\hat{\pi}_\phi, \hat{\pi}, s)\|$$

$$= \lambda \|\nabla_\phi((\mathbf{1}(s,a) - \hat{\pi}_\phi(\cdot|s)) \frac{\hat{\pi}_\phi(a|s)}{\hat{\pi}_{\theta'_\tau}(a|s)})\|$$

$$\leq \frac{\lambda + \max_{s,a}|A_\tau^{\hat{\pi}_\phi}(s,a)|}{\lambda - \max_{s,a}|A_\tau^{\hat{\pi}_\phi}(s,a)|} (\frac{\gamma \max_{a,s}|A_\tau^{\hat{\pi}_\phi}(s,a)|}{1-\gamma} + 2(\lambda + \max_{a,s}|A_\tau^{\hat{\pi}_\phi}(s,a)|) + 2)$$

$$= \frac{\lambda + \max_{s,a}|A_\tau^{\hat{\pi}_\phi}(s,a)|}{\lambda - \max_{s,a}|A_\tau^{\hat{\pi}_\phi}(s,a)|} (\frac{(2-\gamma)\max_{a,s}|A_\tau^{\hat{\pi}_\phi}(s,a)|}{1-\gamma} + 2\lambda + 2)$$

So,

$$\sum_{a\in\mathcal{A}} \|\nabla_\phi^2 \hat{\pi}_{\theta'_\tau}(a|s)\| \leq \frac{1}{\lambda - \max_{s,a}|A_\tau^{\hat{\pi}_\phi}(s,a)|} (\frac{8r_{max}}{(1-\gamma)^3}$$

$$+ \frac{\lambda + \max_{s,a}|A_\tau^{\hat{\pi}_\phi}(s,a)|}{\lambda - \max_{s,a}|A_\tau^{\hat{\pi}_\phi}(s,a)|} (\frac{(2-\gamma)\max_{a,s}|A_\tau^{\hat{\pi}_\phi}(s,a)|}{1-\gamma} + 2\lambda + 2)).$$

$\square$

**Lemma 8.** *Suppose that Assumptions 1 and 2 hold. Let $\hat{\pi}_{\theta'_\tau} = \mathcal{A}lg(\hat{\pi}_\phi, \lambda, \tau)$ where $D_\tau = D_{\tau,1}$, we have*

$$\|\nabla_\phi^2 J_\tau(\hat{\pi}_{\theta'_\tau})\| \leq \frac{r_{max}B}{(1-\gamma)^2} + \frac{2\gamma r_{max}C^2}{(1-\gamma)^3}, \tag{28}$$

*where* $C = \frac{1}{\lambda - A_{max}}(\frac{\gamma A_{max}}{1-\gamma} + 2\lambda + 2A_{max})$ *and* $B = \frac{1}{\lambda - A_{max}}(\frac{8r_{max}}{(1-\gamma)^3} + \frac{\lambda + A_{max}}{\lambda - A_{max}}(\frac{(2-\gamma)A_{max}}{1-\gamma} + 2\lambda + 2))$.

*Proof.* From Lemma 7, we have bounded $\sum_{a\in\mathcal{A}} \|\nabla_\phi \hat{\pi}_{\theta'_\tau}(a|s)\|$ and $\sum_{a\in\mathcal{A}} \|\nabla_\phi^2 \hat{\pi}_{\theta'_\tau}(a|s)\|$. Borrow the result from Lemma D.2 in [2]. $\square$

### K.2.2 Convergence guarantee

**Theorem 5.** *Consider the tabular softmax policy for the discrete state-action space shown in Section 5.1, and the within-task algorithm $\mathcal{A}lg$ in (1). Suppose that Assumptions 1 and 2 hold. Let $\{\phi_t\}_{t=1}^T$ be the sequence generated by Algorithm 1 with $D_\tau = D_{\tau,1}$, $\lambda > A_{max}$, and the step size selected as*

$$\alpha = \min\left\{\left(\frac{r_{max}B}{(1-\gamma)^2} + \frac{2\gamma r_{max}C^2}{(1-\gamma)^3}\right)^{-1}, \frac{1}{G\sqrt{T}}\right\}.$$

*Then,*

$$\frac{1}{T}\sum_{t=1}^T \mathbb{E}_t \left[\|\nabla_\phi \mathbb{E}_{\tau\sim\mathbb{P}(\Gamma)}[J_\tau(\mathcal{A}lg(\hat{\pi}_{\phi_t}, \lambda, \tau))]\|^2\right]$$

$$\leq \left(\frac{2r_{max}^2 B}{(1-\gamma)^3} + \frac{4\gamma r_{max}^2 C^2}{(1-\gamma)^4}\right)\frac{1}{T} + \left(\frac{2r_{max}}{1-\gamma} + \frac{r_{max}B}{(1-\gamma)^2} + \frac{2\gamma r_{max}C^2}{(1-\gamma)^3}\right)\frac{G}{\sqrt{T}},$$

*where*

$$G = \frac{2A_{max}}{1-\gamma}(\frac{A_{max}}{\lambda - A_{max}}\frac{\gamma}{1-\gamma} + 2),$$

$$C = \frac{1}{\lambda - A_{max}}(\frac{\gamma A_{max}}{1-\gamma} + 2\lambda + 2A_{max}),$$

*and*

$$B = \frac{1}{\lambda - A_{max}}(\frac{8r_{max}}{(1-\gamma)^3} + \frac{\lambda + A_{max}}{\lambda - A_{max}}(\frac{(2-\gamma)A_{max}}{1-\gamma} + 2\lambda + 2)).$$

*Proof.* As the smoothness constant of $J_\tau(\hat{\pi}_{\theta_\tau'})$, i.e., $J_\tau(\mathcal{A}lg(\hat{\pi}_\phi, \lambda, \tau))$ is obtained in (8), the smoothness constant of $\mathbb{E}_{\tau\sim\mathbb{P}(\Gamma)}[J_\tau(\mathcal{A}lg(\hat{\pi}_\phi, \lambda, \tau))]$ is the same, i.e.,

$$\|\nabla^2_\phi \mathbb{E}_{\tau\sim\mathbb{P}(\Gamma)}[J_\tau(\mathcal{A}lg(\hat{\pi}_\phi, \lambda, \tau))]\| \le \frac{Br_{max}}{(1-\gamma)^2} + \frac{2\gamma r_{max}C^2}{(1-\gamma)^3}.$$

Moreover, from Lemma 6, we have

$$\|\nabla_\phi J_\tau(\mathcal{A}lg(\hat{\pi}_\phi, \lambda, \tau))\| \le \frac{A_{max}}{1-\gamma}\left(\frac{A_{max}}{\lambda - A_{max}}\frac{\gamma}{1-\gamma} + 2\right).$$

From the convergence theorem of SDG with smoothness and bounded gradient shown in [19], let the step size

$$\alpha = \min\left\{\left(\frac{r_{max}B}{(1-\gamma)^2} + \frac{2\gamma r_{max}C^2}{(1-\gamma)^3}\right)^{-1}, \frac{1}{G\sqrt{T}}\right\},$$

we have

$$\frac{1}{T}\sum_{t=1}^T \mathbb{E}_t\left[\|\nabla_\phi \mathbb{E}_{\tau\sim\mathbb{P}(\Gamma)}[J_\tau(\mathcal{A}lg(\hat{\pi}_{\phi_t}, \lambda, \tau))]\|^2\right]$$

$$\le\left(\frac{2r_{max}B}{(1-\gamma)^2} + \frac{4\gamma r_{max}C^2}{(1-\gamma)^3}\right)\mathbb{E}_{\tau\sim\mathbb{P}(\Gamma)}[J_\tau(\mathcal{A}lg(\hat{\pi}_{\phi_T}, \lambda, \tau)) - J_\tau(\mathcal{A}lg(\hat{\pi}_{\phi_0}, \lambda, \tau))]\frac{1}{T}$$

$$+ \left(2\mathbb{E}_{\tau\sim\mathbb{P}(\Gamma)}[J_\tau(\mathcal{A}lg(\hat{\pi}_{\phi_T}, \lambda, \tau)) - J_\tau(\mathcal{A}lg(\hat{\pi}_{\phi_0}, \lambda, \tau))] + \frac{r_{max}B}{(1-\gamma)^2} + \frac{2\gamma r_{max}C^2}{(1-\gamma)^3}\right)$$

Since $\mathbb{E}_{\tau\sim\mathbb{P}(\Gamma)}[J_\tau(\mathcal{A}lg(\hat{\pi}_{\phi_T}, \lambda, \tau)) - J_\tau(\mathcal{A}lg(\hat{\pi}_{\phi_0}, \lambda, \tau))] \le \frac{r_{max}}{1-\gamma}$, we have

$$\frac{1}{T}\sum_{t=1}^T \mathbb{E}_t\left[\|\nabla_\phi \mathbb{E}_{\tau\sim\mathbb{P}(\Gamma)}[J_\tau(\mathcal{A}lg(\hat{\pi}_{\phi_t}, \lambda, \tau))]\|^2\right]$$

$$\le\left(\frac{2r^2_{max}B}{(1-\gamma)^3} + \frac{4\gamma r^2_{max}C^2}{(1-\gamma)^4}\right)\frac{1}{T} + \left(\frac{2r_{max}}{1-\gamma} + \frac{r_{max}B}{(1-\gamma)^2} + \frac{2\gamma r_{max}C^2}{(1-\gamma)^3}\right)\frac{G}{\sqrt{T}},$$

where

$$G = \frac{2A_{max}}{1-\gamma}\left(\frac{A_{max}}{\lambda - A_{max}}\frac{\gamma}{1-\gamma} + 2\right),$$

$$C = \frac{1}{\lambda - A_{max}}\left(\frac{\gamma A_{max}}{1-\gamma} + 2\lambda + 2A_{max}\right),$$

and

$$B = \frac{1}{\lambda - A_{max}}\left(\frac{8r_{max}}{(1-\gamma)^3} + \frac{\lambda + A_{max}}{\lambda - A_{max}}\left(\frac{(2-\gamma)A_{max}}{1-\gamma} + 2\lambda + 2\right)\right).$$

$\square$

**Corollary 1.** *Suppose all assumptions and conditions in Theorem 5 hold, and we set* $\lambda \ge 2A_{max}$, *then*

$$\frac{1}{T}\sum_{t=1}^T \mathbb{E}_t\left[\|\nabla_\phi \mathbb{E}_{\tau\sim\mathbb{P}(\Gamma)}[J_\tau(\mathcal{A}lg(\hat{\pi}_{\phi_t}, \lambda, \tau))]\|^2\right] \le \frac{(B + 2C^2)r_{max}}{(1-\gamma)^4}\left(\frac{2r_{max}}{T} + \frac{G}{\sqrt{T}}\right),$$

*where* $B \triangleq \frac{16r_{max}}{\lambda(1-\gamma)^3} + \frac{24}{1-\gamma} + \frac{12}{\lambda}$, $C \triangleq \frac{6}{1-\gamma}$, *and* $G \triangleq \frac{4A_{max}}{(1-\gamma)^2}$.

*Proof.* Since $\lambda \ge 2A_{max}$, we have $\frac{1}{\lambda - A_{max}} \le \frac{1}{A_{max}}$ and $\frac{1}{\lambda - A_{max}} \le \frac{2}{\lambda}$. Then, simplify the inequality in Theorem 5. $\square$

# L  Proofs of convergence when $D_\tau = D_{\tau,2}$

## L.1  Gradients of $\nabla_\phi J_\tau(\hat{\pi}_{\theta'_\tau})$ when $D_\tau = D_{\tau,2}$

From Proposition 1, we have

$$\nabla_\phi J_\tau(\hat{\pi}_{\theta'_\tau}) = \frac{1}{1-\gamma}\mathbb{E}_{s\sim\nu_\tau^{\hat{\pi}_{\theta'_\tau}}}\left[\nabla_\phi\hat{\pi}_{\theta'_\tau}(\cdot|s)\cdot A_\tau^{\hat{\pi}_{\theta'_\tau}}(s,\cdot)\right],\tag{29}$$

where

$$\nabla_\phi^\top\hat{\pi}_{\theta'_\tau}(\cdot|s) = \left(M(s)^{-1} - \frac{M(s)^{-1}\mathbf{1}\,\mathbf{1}^\top M(s)^{-1}}{\mathbf{1}^\top M(s)^{-1}\mathbf{1}}\right)$$
$$\left(\nabla_\phi^\top Q_\tau^{\hat{\pi}_\phi}(s,\cdot) - \lambda\nabla_\phi^\top\nabla_{\hat{\pi}(\cdot|s)}d_2^2(\hat{\pi}_\phi,\hat{\pi},s)\right)|_{\hat{\pi}=\hat{\pi}_{\theta'_\tau}},\tag{30}$$

where

$$M(s) = \lambda\nabla_{\hat{\pi}(\cdot|s)}^2 d_2^2(\hat{\pi}_\phi,\hat{\pi},s) = \lambda\begin{bmatrix}\frac{1}{\hat{\pi}_{\theta'_\tau}(a_1|s)} & & \\ & \ddots & \\ & & \frac{1}{\hat{\pi}_{\theta'_\tau}(a_n|s)}\end{bmatrix}.$$

Then,

$$M(s)^{-1} = \frac{1}{\lambda}\begin{bmatrix}\hat{\pi}_{\theta'_\tau}(a_1|s) & & \\ & \ddots & \\ & & \hat{\pi}_{\theta'_\tau}(a_n|s)\end{bmatrix}.\tag{31}$$

Also,

$$\nabla_\phi^\top\nabla_{\hat{\pi}(\cdot|s)}d_2^2(\hat{\pi}_\phi,\hat{\pi},s)|_{\hat{\pi}=\hat{\pi}_{\theta'_\tau}} = \begin{bmatrix}-\frac{\nabla_\phi^\top\hat{\pi}_\phi(a_1|s)}{\hat{\pi}_\phi(a_1|s)}\\ \vdots \\ -\frac{\nabla_\phi^\top\hat{\pi}_\phi(a_n|s)}{\hat{\pi}_\phi(a_n|s)}\end{bmatrix}.\tag{32}$$

Specially, $A_\tau^{\hat{\pi}_{\theta'_\tau}}(s,\cdot)^\top\frac{M(s)^{-1}\mathbf{1}\,\mathbf{1}^\top M(s)^{-1}}{\mathbf{1}^\top M^{-1}\mathbf{1}} = 0$, because we have

$$A_\tau^{\hat{\pi}_{\theta'_\tau}}(s,\cdot)^\top M(s)^{-1}\mathbf{1} = \sum_{a\in\mathcal{A}}\hat{\pi}_{\theta'_\tau}(a|s)A_\tau^{\hat{\pi}_{\theta'_\tau}}(s,a) = 0.$$

Then,

$$\nabla_\phi^\top J_\tau(\hat{\pi}_{\theta'_\tau}) = \frac{1}{1-\gamma}\mathbb{E}_{s\sim\nu_\tau^{\hat{\pi}_{\theta'_\tau}},a\sim\hat{\pi}_{\theta'_\tau}}\left[A_\tau^{\hat{\pi}_{\theta'_\tau}}(s,a)(\frac{1}{\lambda}\nabla_\phi^\top Q_\tau^{\hat{\pi}_\phi}(s,a) + \mathbf{1}^\top(s,a) - \hat{\pi}_\phi(\cdot|s)^\top)\right].$$

Since $\sum_{a\in\mathcal{A}}\hat{\pi}_{\theta'_\tau}(a|s)A_\tau^{\hat{\pi}_{\theta'_\tau}}(s,a) = 0$, then $\sum_{a\in\mathcal{A}}\hat{\pi}_{\theta'_\tau}(a|s)A_\tau^{\hat{\pi}_{\theta'_\tau}}(s,a)\hat{\pi}_\phi(\cdot|s)^\top = 0$. We have

$$\nabla_\phi^\top J_\tau(\hat{\pi}_{\theta'_\tau}) = \frac{1}{1-\gamma}\mathbb{E}_{s\sim\nu_\tau^{\hat{\pi}_{\theta'_\tau}},a\sim\hat{\pi}_{\theta'_\tau}}\left[A_\tau^{\hat{\pi}_{\theta'_\tau}}(s,a)(\frac{1}{\lambda}\nabla_\phi^\top Q_\tau^{\hat{\pi}_\phi}(s,a) + \mathbf{1}^\top(s,a))\right].\tag{33}$$

Here, $\mathbf{1}(s',a')$ denote the column vector where the element is 1 if $s = s'$ and $a = a'$, otherwise is 0, for each pair $(s,a)\in\mathcal{S}\times\mathcal{A}$.

## L.2  Convergence guarantee when $D_\tau = D_{\tau,2}$

### L.2.1  Auxiliary lemmas

**Lemma 9.** *Suppose that Assumption 2 holds. Let $\hat{\pi}_{\theta'_\tau} = \mathcal{A}lg(\hat{\pi}_\phi,\lambda,\tau)$ where $D_\tau = D_{\tau,2}$, we have*

$$\|\nabla_\phi J_\tau(\hat{\pi}_{\theta'_\tau})\| \le \frac{\max_{s,a}|A_\tau^{\hat{\pi}_{\theta'_\tau}}(s,a)|}{1-\gamma}(\frac{\max_{s,a}|A_\tau^{\hat{\pi}_{\theta'_\tau}}(s,a)|}{\lambda}\frac{\gamma}{1-\gamma} + 1).$$

*Proof.* As shown in (33)

$$\nabla_\phi^\top J_\tau(\hat{\pi}_{\theta'_\tau}) = \frac{1}{1-\gamma} \mathbb{E}_{s \sim \nu_\tau^{\hat{\pi}_{\theta'_\tau}}, a \sim \hat{\pi}_{\theta'_\tau}} \left[ A_\tau^{\hat{\pi}_{\theta'_\tau}}(s,a)(\frac{1}{\lambda}\nabla_\phi^\top Q_\tau^{\hat{\pi}_\phi}(s,a) + \mathbf{1}^\top(s,a)) \right].$$

As shown in proof of Lemma 6 in (27),

$$\|\nabla_\phi Q_\tau^{\hat{\pi}_\phi}(s,a)\| \le \frac{\gamma}{1-\gamma} \max_{a,s} |A_\tau^{\hat{\pi}_\phi}(s,a)|,$$

we have that

$$\|\nabla_\phi J_\tau(\hat{\pi}_{\theta'_\tau})\| \le \frac{\max_{s,a}|A_\tau^{\hat{\pi}_{\theta'_\tau}}(s,a)|}{1-\gamma} (\frac{\max_{s,a}|A_\tau^{\hat{\pi}_{\theta'_\tau}}(s,a)|}{\lambda} \frac{\gamma}{1-\gamma} + 1).$$

$\square$

**Lemma 10.** *Suppose that Assumption 2 holds. Let $\hat{\pi}_{\theta'_\tau} = Alg(\hat{\pi}_\phi, \lambda, \tau)$ where $D_\tau = D_{\tau,2}$, for any $s \in \mathcal{S}$, we have*

$$\sum_{a \in \mathcal{A}} \|\nabla_\phi \hat{\pi}_{\theta'_\tau}(a|s)\| \le \frac{2\gamma}{\lambda(1-\gamma)} \max_{a,s} |A_\tau^{\hat{\pi}_\phi}(s,a)| + 4$$

*and*

$$\sum_{a \in \mathcal{A}} \|\nabla_\phi^2 \hat{\pi}_{\theta'_\tau}(a|s)\| \le (\frac{2\gamma}{\lambda(1-\gamma)} \max_{a,s} |A_\tau^{\hat{\pi}_\phi}(s,a)| + 4)^2 + \frac{16r_{max}}{\lambda(1-\gamma)^3} + 2.$$

*Proof.* As shown in 30, we have $\nabla_\phi^\top \hat{\pi}_{\theta'_\tau}(\cdot|s) =$

$$\left( M(s)^{-1} - \frac{M(s)^{-1}\mathbf{1}\,\mathbf{1}^\top M(s)^{-1}}{\mathbf{1}^\top M(s)^{-1}\mathbf{1}} \right) \left( \nabla_\phi^\top Q_\tau^{\hat{\pi}_\phi}(s,\cdot) - \lambda\nabla_\phi^\top \nabla_{\hat{\pi}(\cdot|s)} d_2^2(\hat{\pi}_\phi, \hat{\pi}, s) \right) |_{\hat{\pi}=\hat{\pi}_{\theta'_\tau}},$$

where the computations of $M(s)^{-1}$ and $\nabla_\phi^\top \nabla_{\hat{\pi}(\cdot|s)} d_2^2(\hat{\pi}_\phi, \hat{\pi}, s)$ are shown in (31) (32) and (24), then

$$\nabla_\phi \hat{\pi}_{\theta'_\tau}(a|s) = \hat{\pi}_{\theta'_\tau}(a|s)(\frac{1}{\lambda}\nabla_\phi Q_\tau^{\hat{\pi}_\phi}(s,a) + \mathbf{1}(s,a) - \hat{\pi}_\phi(\cdot|s))$$

$$- \hat{\pi}_{\theta'_\tau}(a|s) \sum_{a' \in \mathcal{A}} \hat{\pi}_{\theta'_\tau}(a'|s)(\frac{1}{\lambda}\nabla_\phi Q_\tau^{\hat{\pi}_\phi}(s,a') + \mathbf{1}(s,a') - \hat{\pi}_\phi(\cdot|s)).$$

Therefore,

$$\|\nabla_\phi \hat{\pi}_{\theta'_\tau}(a|s)\| \le \left\| \hat{\pi}_{\theta'_\tau}(a|s)(\frac{1}{\lambda}\nabla_\phi Q_\tau^{\hat{\pi}_\phi}(s,a) + \mathbf{1}(s,a) - \hat{\pi}_\phi(\cdot|s)) \right\|$$

$$+ \left\| \hat{\pi}_{\theta'_\tau}(a|s) \sum_{a' \in \mathcal{A}} \hat{\pi}_{\theta'_\tau}(a'|s)(\frac{1}{\lambda}\nabla_\phi Q_\tau^{\hat{\pi}_\phi}(s,a') + \mathbf{1}(s,a') - \hat{\pi}_\phi(\cdot|s)) \right\|.$$

Then,

$$\sum_{a \in \mathcal{A}} \|\nabla_\phi \hat{\pi}_{\theta'_\tau}(a|s)\| \le \sum_{a \in \mathcal{A}} \left\| \hat{\pi}_{\theta'_\tau}(a|s)(\frac{1}{\lambda}\nabla_\phi Q_\tau^{\hat{\pi}_\phi}(s,a) + \mathbf{1}(s,a) - \hat{\pi}_\phi(\cdot|s)) \right\|$$

$$+ \sum_{a \in \mathcal{A}} \left\| \hat{\pi}_{\theta'_\tau}(a|s) \sum_{a' \in \mathcal{A}} \hat{\pi}_{\theta'_\tau}(a'|s)(\frac{1}{\lambda}\nabla_\phi Q_\tau^{\hat{\pi}_\phi}(s,a') + \mathbf{1}(s,a') - \hat{\pi}_\phi(\cdot|s)) \right\|.$$

From (27), we have

$$\|\nabla_\phi Q_\tau^{\hat{\pi}_\phi}(s,a)\| \le \frac{\gamma}{1-\gamma} \max_{a,s} |A_\tau^{\hat{\pi}_\phi}(s,a)|.$$

Then,

$$\sum_{a\in\mathcal{A}}\|\nabla_\phi\hat{\pi}_{\theta'_\tau}(a|s)\| \leq \sum_{a\in\mathcal{A}}\hat{\pi}_{\theta'_\tau}(a|s)\left\|\frac{1}{\lambda}\nabla_\phi Q_\tau^{\hat{\pi}_\phi}(s,a)+\mathbf{1}(s,a)-\hat{\pi}_\phi(\cdot|s)\right\|$$

$$+\sum_{a\in\mathcal{A}}\hat{\pi}_{\theta'_\tau}(a|s)\left\|\sum_{a'\in\mathcal{A}}\hat{\pi}_{\theta'_\tau}(a'|s)(\frac{1}{\lambda}\nabla_\phi Q_\tau^{\hat{\pi}_\phi}(s,a')+\mathbf{1}(s,a')-\hat{\pi}_\phi(\cdot|s))\right\|$$

$$\leq\sum_{a\in\mathcal{A}}\hat{\pi}_{\theta'_\tau}(a|s)(\frac{\gamma}{\lambda(1-\gamma)}\max_{a,s}|A_\tau^{\hat{\pi}_\phi}(s,a)|+2)$$

$$+\sum_{a\in\mathcal{A}}\hat{\pi}_{\theta'_\tau}(a|s)\sum_{a'\in\mathcal{A}}\hat{\pi}_{\theta'_\tau}(a'|s)(\frac{\gamma}{\lambda(1-\gamma)}\max_{a,s}|A_\tau^{\hat{\pi}_\phi}(s,a)|+2)$$

$$\leq\frac{2\gamma}{\lambda(1-\gamma)}\max_{a,s}|A_\tau^{\hat{\pi}_\phi}(s,a)|+4,$$

And

$$\|\nabla_\phi\hat{\pi}_{\theta'_\tau}(a|s)\| \leq \hat{\pi}_{\theta'_\tau}(a|s)(\frac{2\gamma}{\lambda(1-\gamma)}\max_{a,s}|A_\tau^{\hat{\pi}_\phi}(s,a)|+4) \tag{34}$$

Moreover, since

$$\nabla_\phi\hat{\pi}_{\theta'_\tau}(a|s) = \hat{\pi}_{\theta'_\tau}(a|s)(\frac{1}{\lambda}\nabla_\phi Q_\tau^{\hat{\pi}_\phi}(s,a)+\mathbf{1}(s,a)-\hat{\pi}_\phi(\cdot|s))$$

$$-\hat{\pi}_{\theta'_\tau}(a|s)\sum_{a'\in\mathcal{A}}\hat{\pi}_{\theta'_\tau}(a'|s)(\frac{1}{\lambda}\nabla_\phi Q_\tau^{\hat{\pi}_\phi}(s,a')+\mathbf{1}(s,a')-\hat{\pi}_\phi(\cdot|s)).$$

we have

$$\nabla_\phi^2\hat{\pi}_{\theta'_\tau}(a|s) = \nabla_\phi\hat{\pi}_{\theta'_\tau}(a|s)(\frac{1}{\lambda}\nabla_\phi Q_\tau^{\hat{\pi}_\phi}(s,a)+\mathbf{1}(s,a)-\hat{\pi}_\phi(\cdot|s))$$

$$+\hat{\pi}_{\theta'_\tau}(a|s)(\frac{1}{\lambda}\nabla_\phi^2 Q_\tau^{\hat{\pi}_\phi}(s,a)+-\nabla_\phi\hat{\pi}_\phi(\cdot|s))$$

$$-\nabla_\phi\left(\hat{\pi}_{\theta'_\tau}(a|s)\sum_{a'\in\mathcal{A}}\hat{\pi}_{\theta'_\tau}(a'|s)(\frac{1}{\lambda}\nabla_\phi Q_\tau^{\hat{\pi}_\phi}(s,a')+\mathbf{1}(s,a')-\hat{\pi}_\phi(\cdot|s))\right).$$

Then,

$$\|\nabla_\phi^2\hat{\pi}_{\theta'_\tau}(a|s)\| \leq 2\|\nabla_\phi\hat{\pi}_{\theta'_\tau}(a|s)\|\|\frac{1}{\lambda}\nabla_\phi Q_\tau^{\hat{\pi}_\phi}(s,a)+\mathbf{1}(s,a)-\hat{\pi}_\phi(\cdot|s)\|$$

$$+2\hat{\pi}_{\theta'_\tau}(a|s)\|\frac{1}{\lambda}\nabla_\phi^2 Q_\tau^{\hat{\pi}_\phi}(s,a)-\nabla_\phi\hat{\pi}_\phi(\cdot|s)\|.$$

From (34),

$$\|\nabla_\phi\hat{\pi}_{\theta'_\tau}(a|s)\| \leq \hat{\pi}_{\theta'_\tau}(a|s)(\frac{2\gamma}{\lambda(1-\gamma)}\max_{a,s}|A_\tau^{\hat{\pi}_\phi}(s,a)|+4).$$

From (27)

$$\|\frac{1}{\lambda}\nabla_\phi Q_\tau^{\hat{\pi}_\phi}(s,a)+\mathbf{1}(s,a)-\hat{\pi}_\phi(\cdot|s)\| \leq \frac{\gamma}{\lambda(1-\gamma)}\max_{a,s}|A_\tau^{\hat{\pi}_\phi}(s,a)|+2$$

From Lemma D.4 in [2], we have

$$\|\nabla_\phi^2 Q_\tau^{\hat{\pi}_\phi}(s,a)\| \leq \frac{8r_{max}}{(1-\gamma)^3},$$

then

$$\|\frac{1}{\lambda}\nabla_\phi^2 Q_\tau^{\hat{\pi}_\phi}(s,a)-\nabla_\phi\hat{\pi}_\phi(\cdot|s)\| \leq \frac{8r_{max}}{\lambda(1-\gamma)^3}+1.$$

Therefore,

$$\|\nabla_\phi^2\hat{\pi}_{\theta'_\tau}(a|s)\| \leq \hat{\pi}_{\theta'_\tau}(a|s)(\frac{2\gamma}{\lambda(1-\gamma)}\max_{a,s}|A_\tau^{\hat{\pi}_\phi}(s,a)|+4)^2+2\hat{\pi}_{\theta'_\tau}(a|s)(\frac{8r_{max}}{\lambda(1-\gamma)^3}+1).$$

So,

$$\sum_{a\in\mathcal{A}}\|\nabla_\phi^2\hat{\pi}_{\theta'_\tau}(a|s)\| \leq (\frac{2\gamma}{\lambda(1-\gamma)}\max_{a,s}|A_\tau^{\hat{\pi}_\phi}(s,a)|+4)^2+\frac{16r_{max}}{\lambda(1-\gamma)^3}+2.$$

$\square$

**Lemma 11.** *Suppose that Assumptions 1 and 2 hold. Let $\hat{\pi}_{\theta'_\tau} = \mathcal{A}lg(\hat{\pi}_\phi, \lambda, \tau)$ where $D_\tau = D_{\tau,2}$, we have*

$$\|\nabla_\phi^2 J_\tau(\hat{\pi}_{\theta'_\tau})\| \leq \frac{r_{max}B}{(1-\gamma)^2} + \frac{2\gamma r_{max}C^2}{(1-\gamma)^3}, \tag{35}$$

*where $C = \frac{2\gamma}{\lambda(1-\gamma)}A_{max} + 4$ and $B = (\frac{2\gamma}{\lambda(1-\gamma)}A_{max} + 4)^2 + \frac{16r_{max}}{\lambda(1-\gamma)^3} + 2$.*

*Proof.* Similar to the proof of Lemma 8 by using Lemma 10. $\qquad\square$

### L.2.2 Convergence guarantee

**Theorem 6.** *Consider the tabular softmax policy for the discrete state-action space shown in Section 5.1, and the within-task algorithm $\mathcal{A}lg$ in (1). Suppose that Assumptions 1 and 2 hold. Let $\{\phi_t\}_{t=1}^T$ be the sequence generated by Algorithm 1 with $D_\tau = D_{\tau,2}$ and the step size selected as*

$$\alpha = \min\left\{\left(\frac{r_{max}B}{(1-\gamma)^2} + \frac{2\gamma r_{max}C^2}{(1-\gamma)^3}\right)^{-1}, \frac{1}{G\sqrt{T}}\right\}.$$

*Then,*

$$\frac{1}{T}\sum_{t=1}^T \mathbb{E}_t\left[\|\nabla_\phi \mathbb{E}_{\tau\sim\mathbb{P}(\Gamma)}[J_\tau(\mathcal{A}lg(\hat{\pi}_{\phi_t}, \lambda, \tau))]\|^2\right]$$

$$\leq \left(\frac{2r_{max}^2 B}{(1-\gamma)^3} + \frac{4\gamma r_{max}^2 C^2}{(1-\gamma)^4}\right)\frac{1}{T} + \left(\frac{2r_{max}}{1-\gamma} + \frac{r_{max}B}{(1-\gamma)^2} + \frac{2\gamma r_{max}C^2}{(1-\gamma)^3}\right)\frac{G}{\sqrt{T}},$$

*where*

$$G = \frac{2A_{max}}{1-\gamma}\left(\frac{A_{max}}{\lambda}\frac{\gamma}{1-\gamma} + 1\right),$$

$$C = \frac{2\gamma}{\lambda(1-\gamma)}A_{max} + 4,$$

*and*

$$B = \left(\frac{2\gamma A_{max}}{\lambda(1-\gamma)} + 4\right)^2 + \frac{16r_{max}}{\lambda(1-\gamma)^3} + 2.$$

*Proof.* Similar to the proof of Theorem 5, by using the gradient bound in Lemma 9 and the smoothness in Lemma 11. $\qquad\square$

**Corollary 2.** *Suppose all assumptions and conditions in Theorem 6 hold, and we set $\lambda \geq 2A_{max}$, then*

$$\frac{1}{T}\sum_{t=1}^T \mathbb{E}_t\left[\|\nabla_\phi \mathbb{E}_{\tau\sim\mathbb{P}(\Gamma)}[J_\tau(\mathcal{A}lg(\hat{\pi}_{\phi_t}, \lambda, \tau))]\|^2\right] \leq \frac{(B+2C^2)r_{max}}{(1-\gamma)^4}\left(\frac{2r_{max}}{T} + \frac{G}{\sqrt{T}}\right),$$

*where $B \triangleq \frac{16r_{max}}{\lambda(1-\gamma)^3} + \frac{18}{(1-\gamma)^2}$, $C \triangleq \frac{4}{1-\gamma}$, and $G \triangleq \frac{2A_{max}}{(1-\gamma)^2}$).*

*Proof.* Since $\lambda \geq 2A_{max}$, we have $\frac{1}{\lambda} \leq \frac{1}{2A_{max}}$. Then, simplify the inequality in Theorem 5. $\qquad\square$

## M   Proofs of convergence when $D_\tau = D_{\tau,3}$

**Lemma 12.** *Suppose that Assumptions 1, 2, and 3 hold. Let $\hat{\pi}_{\theta'_\tau} = \mathcal{A}lg(\hat{\pi}_\phi, \lambda, \tau)$ where $D_\tau = D_{\tau,3}$. If $\lambda > (6L_1^2 + 2L_2)A_{max}$, then $\nabla_\phi J_\tau(\mathcal{A}lg^{(3)}(\hat{\pi}_\phi, \lambda, \tau))$ exists for any $\phi$, and*

$$\|\nabla_\phi J_\tau(\hat{\pi}_{\theta'_\tau})\| \leq \frac{L_1 A_{max}(\lambda + \frac{2\gamma}{1-\gamma}L_1^2 A_{max})}{(1-\gamma)(\lambda - (6L_1^2 + 2L_2)A_{max})}.$$

*Proof.* From Proposition 2, we have

$$\nabla_\phi J_\tau(\hat{\pi}_{\theta'_\tau}) = \frac{1}{1-\gamma}\nabla_\phi\theta'_\tau \cdot \mathop{\mathbb{E}}_{\substack{s\sim\nu_\tau^{\hat{\pi}_{\theta'_\tau}}\\a\sim\hat{\pi}_{\theta'_\tau}(\cdot|s)}} \left[\frac{\nabla_{\theta'_\tau}\hat{\pi}_{\theta'_\tau}(a|s)}{\hat{\pi}_{\theta'_\tau}(a|s)}A_\tau^{\hat{\pi}_{\theta'_\tau}}(s,a)\right],$$

where

$$\nabla_\phi^\top\theta'_\tau = -\mathop{\mathbb{E}}_{\substack{s\sim\nu_\tau^{\hat{\pi}_\phi}\\a\sim\hat{\pi}_\phi(\cdot|s)}}\left[-\frac{\nabla_\theta^2\hat{\pi}_\theta(a|s)}{\hat{\pi}_\phi(a|s)}Q_\tau^{\hat{\pi}_\phi}(s,a) + \lambda\nabla_\theta^2 d^2(\hat{\pi}_\phi(\cdot|s),\hat{\pi}_\theta(\cdot|s))\right]^{-1}$$

$$\mathop{\mathbb{E}}_{\substack{s\sim\nu_\tau^{\hat{\pi}_\phi}\\a\sim\hat{\pi}_\phi(\cdot|s)}}\left[-\frac{\nabla_\theta\hat{\pi}_\theta(a|s)}{\hat{\pi}_\phi(a|s)}\nabla_\phi^\top Q_\tau^{\hat{\pi}_\phi}(s,a) + \lambda\nabla_\phi^\top\nabla_\theta d^2(\hat{\pi}_\phi(\cdot|s),\hat{\pi}_\theta(\cdot|s))\right]|_{\theta=\theta'_\tau}.$$

When $D_\tau = D_{\tau,3}$, and the policy with function approximation is defined by $\hat{\pi}_\theta(a|s) \triangleq \frac{\exp(f_\theta(s,a))}{\int_\mathcal{A}\exp(f_\theta(s,a'))da'}, \forall(s,a)\in\mathcal{S}\times\mathcal{A}$, from Lemma 4,

$$\nabla_\phi J_\tau(\hat{\pi}_{\theta'_\tau}) = \frac{1}{1-\gamma}\nabla_\phi\theta'_\tau \cdot \mathop{\mathbb{E}}_{\substack{s\sim\nu_\tau^{\hat{\pi}_{\theta'_\tau}}\\a\sim\hat{\pi}_{\theta'_\tau}(\cdot|s)}}\left[\nabla_{\theta'_\tau}f_{\hat{\pi}_{\theta'_\tau}}(s,a)A_\tau^{\hat{\pi}_{\theta'_\tau}}(s,a)\right],$$

where $\nabla_\phi^\top\theta'_\tau =$

$$\mathop{\mathbb{E}}_{\substack{s\sim\nu_\tau^{\hat{\pi}_\phi}\\a\sim\hat{\pi}_\phi(\cdot|s)}}\left[-\frac{\nabla_{\theta'_\tau}^2\hat{\pi}_{\theta'_\tau}(a|s)}{\hat{\pi}_\phi(a|s)}Q_\tau^{\hat{\pi}_\phi}(s,a) + \lambda I\right]^{-1}\mathop{\mathbb{E}}_{\substack{s\sim\nu_\tau^{\hat{\pi}_\phi}\\a\sim\hat{\pi}_\phi(\cdot|s)}}\left[\frac{\nabla_{\theta'_\tau}\hat{\pi}_{\theta'_\tau}(a|s)}{\hat{\pi}_\phi(a|s)}\nabla_\phi^\top Q_\tau^{\hat{\pi}_\phi}(s,a) + \lambda I\right]$$

$$= \mathop{\mathbb{E}}_{s\sim\nu_\tau^{\hat{\pi}_\phi}}\left[-\int_\mathcal{A}\nabla_{\theta'_\tau}^2\hat{\pi}_{\theta'_\tau}(a|s)Q_\tau^{\hat{\pi}_\phi}(s,a)da + \lambda I\right]^{-1}\mathop{\mathbb{E}}_{s\sim\nu_\tau^{\hat{\pi}_\phi}}\left[\int_\mathcal{A}\nabla_{\theta'_\tau}\hat{\pi}_{\theta'_\tau}(a|s)\nabla_\phi^\top Q_\tau^{\hat{\pi}_\phi}(s,a)da + \lambda I\right]$$

$$= \mathop{\mathbb{E}}_{s\sim\nu_\tau^{\hat{\pi}_\phi}}\left[-\int_\mathcal{A}\nabla_{\theta'_\tau}^2\hat{\pi}_{\theta'_\tau}(a|s)A_\tau^{\hat{\pi}_\phi}(s,a)da + \lambda I\right]^{-1}\mathop{\mathbb{E}}_{s\sim\nu_\tau^{\hat{\pi}_\phi}}\left[\int_\mathcal{A}\nabla_{\theta'_\tau}\hat{\pi}_{\theta'_\tau}(a|s)\nabla_\phi^\top Q_\tau^{\hat{\pi}_\phi}(s,a)da + \lambda I\right]$$

First, we have

$$\|\nabla_\phi J_\tau(\hat{\pi}_{\theta'_\tau})\| = \frac{1}{1-\gamma}\|\nabla_\phi\theta'_\tau\|\|\mathop{\mathbb{E}}_{\substack{s\sim\nu_\tau^{\hat{\pi}_{\theta'_\tau}}\\a\sim\hat{\pi}_{\theta'_\tau}(\cdot|s)}}\left[\nabla_\theta f_\theta(s,a)A_\tau^{\hat{\pi}_{\theta'_\tau}}(s,a)\right]\|,$$

and

$$\|\mathop{\mathbb{E}}_{\substack{s\sim\nu_\tau^{\hat{\pi}_{\theta'_\tau}}\\a\sim\hat{\pi}_{\theta'_\tau}(\cdot|s)}}\left[\nabla_\theta f_\theta(s,a)A_\tau^{\hat{\pi}_{\theta'_\tau}}(s,a)\right]\| \leq \|\max_{a,s}\nabla_\theta f_\theta(s,a)\|\max_{a,s}|A_\tau^{\hat{\pi}_{\theta'_\tau}}(s,a)| \leq L_1 A_{max}.$$

For the term $\nabla_\phi\theta'_\tau$, consider $\nabla_{\theta'_\tau}\hat{\pi}_{\theta'_\tau}(a|s)$ and $\nabla_{\theta'_\tau}^2\hat{\pi}_{\theta'_\tau}(a|s)$, we have

$$\nabla_\theta\hat{\pi}_\theta(a|s) = \hat{\pi}_\theta(a|s)\nabla_\theta f_\theta(s,a) - \hat{\pi}_\theta(a|s)\frac{\int_\mathcal{A}\nabla_\theta f_\theta(s,a')\exp(f_\theta(s,a'))da'}{\int_\mathcal{A}\exp(f_\theta(s,a'))da'}. \tag{36}$$

Then,

$$\|\nabla_\theta\hat{\pi}_\theta(a|s)\| \leq \hat{\pi}_\theta(a|s)\|\nabla_\theta f_\theta(s,a)\| + \hat{\pi}_\theta(a|s)\left\|\frac{\int_\mathcal{A}\nabla_\theta f_\theta(s,a')\exp(f_\theta(s,a'))da'}{\int_\mathcal{A}\exp(f_\theta(s,a'))da'}\right\| \tag{37}$$

$$\leq 2\hat{\pi}_\theta(a|s)L_1$$

We also have $\nabla_\theta^2\hat{\pi}_\theta(a|s) =$

$$\nabla_\theta\hat{\pi}_\theta(a|s)\nabla_\theta^\top f_\theta(s,a) + \hat{\pi}_\theta(a|s)\nabla_\theta^2 f_\theta(s,a) - \nabla_\theta\hat{\pi}_\theta(a|s)\frac{\int_\mathcal{A}\nabla_\theta^\top f_\theta(s,a')\exp(f_\theta(s,a'))da'}{\int_\mathcal{A}\exp(f_\theta(s,a'))da'}$$

$$- \hat{\pi}_\theta(a|s)\frac{\int_\mathcal{A}\nabla_\theta^2 f_\theta(s,a')\exp(f_\theta(s,a'))da' + \nabla_\theta f_\theta(s,a')\nabla_\theta^\top f_\theta(s,a')\exp(f_\theta(s,a'))da'}{\int_\mathcal{A}\exp(f_\theta(s,a'))da'}$$

$$+ \hat{\pi}_\theta(a|s)\frac{\int_\mathcal{A}\nabla_\theta f_\theta(s,a')\exp(f_\theta(s,a'))da'\int_\mathcal{A}\nabla_\theta^\top f_\theta(s,a')\exp(f_\theta(s,a'))da'}{(\int_\mathcal{A}\exp(f_\theta(s,a'))da')^2}.$$

Then,

$$\|\nabla_\theta^2 \hat{\pi}_\theta(a|s)\| \le 2\hat{\pi}_\theta(a|s)L_1^2 + \hat{\pi}_\theta(a|s)L_2 + 2\hat{\pi}_\theta(a|s)L_1^2 + \hat{\pi}_\theta(a|s)L_2 + 2\hat{\pi}_\theta(a|s)L_1^2$$
$$= 6\hat{\pi}_\theta(a|s)L_1^2 + 2\hat{\pi}_\theta(a|s)L_2. \tag{38}$$

So,

$$\left\| \mathbb{E}_{s\sim\nu_\tau^{\hat{\pi}_\phi}} \left[ \int_{\mathcal{A}} \nabla_{\theta_\tau'}^2 \hat{\pi}_{\theta_\tau'}(a|s) A_\tau^{\hat{\pi}_\phi}(s,a)da \right] \right\| \le (6L_1^2 + 2L_2)A_{max}.$$

Since $\mathbb{E}_{s\sim\nu_\tau^{\hat{\pi}_\phi}} \left[ \int_{\mathcal{A}} \nabla_{\theta_\tau'}^2 \hat{\pi}_{\theta_\tau'}(a|s) A_\tau^{\hat{\pi}_\phi}(s,a)da \right]$ is a diagonal matrix, the above shown its largest absolute eigenvalue is smaller than $(6L_1^2 + 2L_2)A_{max}$. Then, the smallest eigenvalue of $\mathbb{E}_{s\sim\nu_\tau^{\hat{\pi}_\phi}} \left[ -\int_{\mathcal{A}} \nabla_{\theta_\tau'}^2 \hat{\pi}_{\theta_\tau'}(a|s) A_\tau^{\hat{\pi}_\phi}(s,a)da + \lambda I \right]$ is larger than $\lambda - (6L_1^2 + 2L_2)A_{max}$. Therefore, if $\lambda > (6L_1^2 + 2L_2)A_{max}$,

$$\left\| \mathbb{E}_{s\sim\nu_\tau^{\hat{\pi}_\phi}} \left[ -\int_{\mathcal{A}} \nabla_{\theta_\tau'}^2 \hat{\pi}_{\theta_\tau'}(a|s) A_\tau^{\hat{\pi}_\phi}(s,a)da + \lambda I \right]^{-1} \right\| \le \frac{1}{\lambda - (6L_1^2 + 2L_2)A_{max}}. \tag{39}$$

Moreover, if $\lambda > (6L_1^2 + 2L_2)A_{max}$, the objective function in the optimization problem $\mathcal{A}lg(\hat{\pi}_\phi, \lambda, \tau)$ is strongly concave. Then, from [64], the solution is unique and $\nabla_\phi J_\tau(\mathcal{A}lg^{(3)}(\hat{\pi}_\phi, \lambda, \tau))$ exists.

From (6),

$$\nabla_\phi Q_\tau^{\hat{\pi}_\phi}(s,a) = \frac{\gamma}{1-\gamma} \cdot \mathbb{E}_{(s',a')\sim\sigma_{\tau,\hat{\pi}_\phi}^{(s,a)}} \left[ \nabla_\phi f_\phi(s',a') A_\tau^{\hat{\pi}_\phi}(s',a') \right].$$

Then,

$$\|\nabla_\phi Q_\tau^{\hat{\pi}_\phi}(s,a)\| \le \frac{\gamma}{1-\gamma} L_1 A_{max}.$$

Combine (37), we have

$$\left\| \mathbb{E}_{s\sim\nu_\tau^{\hat{\pi}_\phi}} \left[ \int_{\mathcal{A}} \nabla_{\theta_\tau'} \hat{\pi}_{\theta_\tau'}(a|s) \nabla_\phi^\top Q_\tau^{\hat{\pi}_\phi}(s,a)da + \lambda I \right] \right\| \le \lambda + \frac{2\gamma}{1-\gamma} L_1^2 A_{max}.$$

So we have

$$\|\nabla_\phi \theta_\tau'\| \le \frac{\lambda + \frac{2\gamma}{1-\gamma} L_1^2 A_{max}}{(1-\gamma)(\lambda - (6L_1^2 + 2L_2)A_{max})}. \tag{40}$$

Therefore, we have

$$\|\nabla_\phi J_\tau(\hat{\pi}_{\theta_\tau'})\| \le \frac{L_1 A_{max}(\lambda + \frac{2\gamma}{1-\gamma} L_1^2 A_{max})}{(1-\gamma)(\lambda - (6L_1^2 + 2L_2)A_{max})}.$$

$\square$

**Lemma 13.** *For a softmax policy parameterized by $\phi$,*

$$\|\nabla_\phi^2 J_\tau(\hat{\pi}_\phi)\| \le \frac{(6L_1^2 + 2L_2)r_{max}}{(1-\gamma)^2} + \frac{8\gamma L_1^2 r_{max}}{(1-\gamma)^3}$$

$$\|\nabla_\phi^2 Q_\tau^{\hat{\pi}_\phi}(s,a)\| \le \frac{8\gamma^2 L_1^2 r_{max}}{(1-\gamma)^3} + \frac{\gamma(6L_1^2 + 2L_2)r_{max}}{(1-\gamma)^2}. \tag{41}$$

*Proof.* From 37,

$$\int_{\mathcal{A}} \|\nabla_\phi \hat{\pi}_\phi(a|s)\|da \le 2L_1.$$

From 38,

$$\int_{\mathcal{A}} \|\nabla_\phi \hat{\pi}_\phi(a|s)\|da \le 6L_1^2 + 2L_2.$$

Borrow the result from Lemma D.2 in [2],

$$\|\nabla_\phi^2 J_\tau(\hat{\pi}_\phi)\| \le \frac{(6L_1^2 + 2L_2)r_{max}}{(1-\gamma)^2} + \frac{8\gamma L_1^2 r_{max}}{(1-\gamma)^3}$$

$$\|\nabla_\phi^2 Q_\tau^{\hat{\pi}_\phi}(s,a)\| \leq \frac{8\gamma^2 L_1^2 r_{max}}{(1-\gamma)^3} + \frac{\gamma(6L_1^2 + 2L_2)r_{max}}{(1-\gamma)^2}.$$

$\square$

**Lemma 14.** *Suppose that Assumption 2 holds. Let $\hat{\pi}_{\theta'_\tau} = Alg(\hat{\pi}_\phi, \lambda, \tau)$ where $D_\tau = D_{\tau,3}$, for any $s \in \mathcal{S}$, we have*

$$\int_{\mathcal{A}} \|\nabla_\phi \hat{\pi}_{\theta'_\tau}(a|s)\| da \leq \frac{2L_1(\lambda + \frac{2\gamma}{1-\gamma}L_1^2 A_{max})}{(1-\gamma)(\lambda - (6L_1^2 + 2L_2)A_{max})}$$

*and*

$$\int_{\mathcal{A}} \|\nabla_\phi^2 \hat{\pi}_{\theta'_\tau}(a|s)\| da \leq \frac{(160L_1^3 + 56L_1 L_2 + 4L_3)(\lambda + \frac{2\gamma}{1-\gamma}L_1^2 A_{max})^2}{(1-\gamma)^3(\lambda - (6L_1^2 + 2L_2)A_{max})^2}.$$

*Proof.* First consider $\nabla_\phi \hat{\pi}_{\theta'_\tau}(a|s)$, we have

$$\nabla_\phi \hat{\pi}_{\theta'_\tau}(a|s) = \hat{\pi}_{\theta'_\tau}(a|s)\nabla_\phi \theta'_\tau \nabla_{\theta'_\tau} f_{\theta'_\tau}(s,a) - \hat{\pi}_{\theta'_\tau}(a|s)\nabla_\phi \theta'_\tau \frac{\int_{\mathcal{A}} \nabla_{\theta'_\tau} f_{\theta'_\tau}(s,a')\exp(f_{\theta'_\tau}(s,a'))da'}{\int_{\mathcal{A}} \exp(f_{\theta'_\tau}(s,a'))da'}, \tag{42}$$

Then,

$$\begin{aligned}
\|\nabla_\phi \hat{\pi}_{\theta'_\tau}(a|s)\| \leq &\hat{\pi}_{\theta'_\tau}(a|s)\|\nabla_\phi \theta'_\tau\|\|\nabla_{\theta'_\tau} f_{\theta'_\tau}(s,a)\| + \\
&\hat{\pi}_{\theta'_\tau}(a|s)\|\nabla_\phi \theta'_\tau\| \left\| \frac{\int_{\mathcal{A}} \nabla_{\theta'_\tau} f_{\theta'_\tau}(s,a')\exp(f_{\theta'_\tau}(s,a'))da'}{\int_{\mathcal{A}} \exp(f_{\theta'_\tau}(s,a'))da'} \right\| \\
\leq &2\hat{\pi}_{\theta'_\tau}(a|s)\frac{(\lambda + \frac{2\gamma}{1-\gamma}L_1^2 A_{max})L_1}{(1-\gamma)(\lambda - (6L_1^2 + 2L_2)A_{max})}.
\end{aligned} \tag{43}$$

Then,

$$\int_{\mathcal{A}} \|\nabla_\phi \hat{\pi}_{\theta'_\tau}(a|s)\| da \leq \frac{2L_1(\lambda + \frac{2\gamma}{1-\gamma}L_1^2 A_{max})}{(1-\gamma)(\lambda - (6L_1^2 + 2L_2)A_{max})}.$$

Next, we consider $\nabla_\phi^2 \hat{\pi}_{\theta'_\tau}(a|s)$. From (42), we have

$$\begin{aligned}
\nabla_\phi^2 \hat{\pi}_{\theta'_\tau}(a|s) =& \nabla_\phi \theta'_\tau \nabla_{\theta'_\tau} f_{\theta'_\tau}(s,a)\nabla_\phi^\top \hat{\pi}_{\theta'_\tau}(a|s) + \hat{\pi}_{\theta'_\tau}(a|s)\nabla_\phi^2 \theta'_\tau \nabla_{\theta'_\tau} f_{\theta'_\tau}(s,a) \\
&+ \hat{\pi}_{\theta'_\tau}(a|s)\nabla_\phi \theta'_\tau \nabla_{\theta'_\tau}^2 f_{\theta'_\tau}(s,a)\nabla_\phi^\top \theta'_\tau - \nabla_\phi \theta'_\tau \frac{\int_{\mathcal{A}} \nabla_{\theta'_\tau} f_{\theta'_\tau}(s,a')\exp(f_{\theta'_\tau}(s,a'))da'}{\int_{\mathcal{A}} \exp(f_{\theta'_\tau}(s,a'))da'}\nabla_\phi^\top \hat{\pi}_{\theta'_\tau}(a|s) \\
&- \hat{\pi}_{\theta'_\tau}(a|s)\nabla_\phi^2 \theta'_\tau \frac{\int_{\mathcal{A}} \nabla_{\theta'_\tau} f_{\theta'_\tau}(s,a')\exp(f_{\theta'_\tau}(s,a'))da'}{\int_{\mathcal{A}} \exp(f_{\theta'_\tau}(s,a'))da'} - \hat{\pi}_{\theta'_\tau}(a|s)\nabla_\phi \theta'_\tau \\
&\frac{\int_{\mathcal{A}}(\nabla_{\theta'_\tau}^2 f_{\theta'_\tau}(s,a')\exp(f_{\theta'_\tau}(s,a')) + \nabla_{\theta'_\tau} f_{\theta'_\tau}(s,a')\nabla_{\theta'_\tau}^\top f_{\theta'_\tau}(s,a')\exp(f_{\theta'_\tau}(s,a')))da'}{\int_{\mathcal{A}} \exp(f_{\theta'_\tau}(s,a'))da'}\nabla_\phi^\top \theta'_\tau \\
&+ \hat{\pi}_{\theta'_\tau}(a|s)\nabla_\phi \theta'_\tau \left( \frac{\int_{\mathcal{A}} \nabla_{\theta'_\tau} f_{\theta'_\tau}(s,a')\exp(f_{\theta'_\tau}(s,a'))da'}{\int_{\mathcal{A}} \exp(f_{\theta'_\tau}(s,a'))da'} \right)^2 \nabla_\phi^\top \theta'_\tau.
\end{aligned}$$

Therefore,

$$\begin{aligned}
\|\nabla_\phi^2 \hat{\pi}_{\theta'_\tau}(a|s)\| \leq& \|\nabla_\phi \theta'_\tau\|\|\nabla_{\theta'_\tau} f_{\theta'_\tau}(s,a)\|\|\nabla_\phi \hat{\pi}_{\theta'_\tau}(a|s)\| + \hat{\pi}_{\theta'_\tau}(a|s)\|\nabla_\phi^2 \theta'_\tau\|\|\nabla_{\theta'_\tau} f_{\theta'_\tau}(s,a)\| \\
&+ \hat{\pi}_{\theta'_\tau}(a|s)\|\nabla_\phi \theta'_\tau\|^2\|\nabla_{\theta'_\tau}^2 f_{\theta'_\tau}(s,a)\| + \|\nabla_\phi \theta'_\tau\|\|\nabla_{\theta'_\tau} f_{\theta'_\tau}(s,a)\|\|\nabla_\phi \hat{\pi}_{\theta'_\tau}(a|s)\| \\
&+ \hat{\pi}_{\theta'_\tau}(a|s)\|\nabla_\phi^2 \theta'_\tau\|\|\nabla_{\theta'_\tau} f_{\theta'_\tau}(s,a)\| + \hat{\pi}_{\theta'_\tau}(a|s)\|\nabla_\phi \theta'_\tau\|^2(\|\nabla_{\theta'_\tau}^2 f_{\theta'_\tau}(s,a)\| + \|\nabla_{\theta'_\tau} f_{\theta'_\tau}(s,a)\|^2) \\
&+ \hat{\pi}_{\theta'_\tau}(a|s)\|\nabla_\phi \theta'_\tau\|^2\|\nabla_{\theta'_\tau} f_{\theta'_\tau}(s,a)\|^2 \\
\leq& 2L_1\|\nabla_\phi \theta'_\tau\|\|\nabla_\phi \hat{\pi}_{\theta'_\tau}(a|s)\| + 2\hat{\pi}_{\theta'_\tau}(a|s)L_1\|\nabla_\phi^2 \theta'_\tau\| + 2\hat{\pi}_{\theta'_\tau}(a|s)\|\nabla_\phi \theta'_\tau\|^2(L_2 + L_1^2).
\end{aligned}$$

From (40) and (43)

$$\|\nabla_\phi^2 \hat{\pi}_{\theta'_\tau}(a|s)\| \leq \hat{\pi}_{\theta'_\tau}(a|s)\left( \frac{2L_2 + 6L_1^2(\lambda + \frac{2\gamma}{1-\gamma}L_1^2 A_{max})^2}{(1-\gamma)^2(\lambda - (6L_1^2 + 2L_2)A_{max})^2} + 2L_1\|\nabla_\phi^2 \theta'_\tau\| \right).$$

Then,

$$\int_{\mathcal{A}} \|\nabla_\phi^2 \hat{\pi}_{\theta'_\tau}(a|s)\| da \leq \frac{2L_2 + 6L_1^2(\lambda + \frac{2\gamma}{1-\gamma} L_1^2 A_{max})^2}{(1-\gamma)^2(\lambda - (6L_1^2 + 2L_2)A_{max})^2} + 2L_1 \|\nabla_\phi^2 \theta'_\tau\|.$$

Next, we consider $\nabla_\phi^2 \theta'_\tau$. We have

$$\nabla_\phi^2 \theta'_\tau = \mathbb{E}_{s \sim \nu_\tau^{\hat{\pi}_\phi}} \left[ -\int_{\mathcal{A}} \nabla_{\theta'_\tau}^2 \hat{\pi}_{\theta'_\tau}(a|s) Q_\tau^{\hat{\pi}_\phi}(s,a) da + \lambda I \right]^{-1}$$

$$\mathbb{E}_{s \sim \nu_\tau^{\hat{\pi}_\phi}} \left[ \int_{\mathcal{A}} \left( \nabla_{\theta'_\tau}^2 \hat{\pi}_{\theta'_\tau}(a|s) \nabla_\phi^\top \theta'_\tau \nabla_\phi^\top Q_\tau^{\hat{\pi}_\phi}(s,a) + \nabla_{\theta'_\tau} \hat{\pi}_{\theta'_\tau}(a|s) \nabla_\phi^2 Q_\tau^{\hat{\pi}_\phi}(s,a) \right) da \right] -$$

$$M \mathbb{E}_{s \sim \nu_\tau^{\hat{\pi}_\phi}} \left[ -\int_{\mathcal{A}} \left( \nabla_{\theta'_\tau}^3 \hat{\pi}_{\theta'_\tau}(a|s) \nabla_\phi \theta'_\tau A_\tau^{\hat{\pi}_\phi}(s,a) + \nabla_{\theta'_\tau}^2 \hat{\pi}_{\theta'_\tau}(a|s) \nabla_\phi^\top Q_\tau^{\hat{\pi}_\phi}(s,a) \right) da \right] M^{-1} N$$

where $M = \mathbb{E}_{s \sim \nu_\tau^{\hat{\pi}_\phi}} \left[ -\int_{\mathcal{A}} \nabla_{\theta'_\tau}^2 \hat{\pi}_{\theta'_\tau}(a|s) A_\tau^{\hat{\pi}_\phi}(s,a) da + \lambda I \right]$ and $N = \mathbb{E}_{s \sim \nu_\tau^{\hat{\pi}_\phi}} \left[ \int_{\mathcal{A}} \nabla_{\theta'_\tau} \hat{\pi}_{\theta'_\tau}(a|s) \nabla_\phi^\top Q_\tau^{\hat{\pi}_\phi}(s,a) da + \lambda I \right]$. Also, we have $M^{-1} N = \nabla_\phi \theta'_\tau$.

Similar to (37)(38), we can derive the upper bound of $\|\nabla_\phi^3 \hat{\pi}_\phi\|$, then

$$\|\nabla_{\theta'_\tau}^3 \hat{\pi}_{\theta'_\tau}(a|s)\| \leq \hat{\pi}_{\theta'_\tau}(a|s)(40L_1^3 + 16L_1 L_2 + 2L_3).$$

So, from (38)(39)(40)(41), we have

$$\|\nabla_\phi^2 \theta'_\tau\| \leq \frac{2\gamma L_1^2 A_{max}(6L_1^2 + 2L_2)(\lambda + \frac{2\gamma}{1-\gamma} L_1^2 A_{max})}{(1-\gamma)^2(\lambda - (6L_1^2 + 2L_2)A_{max})^2}$$

$$+ \left( \frac{8\gamma^2 L_1^2 r_{max}}{(1-\gamma)^3} + \frac{\gamma(6L_1^2 + 2L_2)r_{max}}{(1-\gamma)^2} \right) \frac{1}{\lambda - (6L_1^2 + 2L_2)A_{max}}$$

$$+ \frac{\lambda + \frac{2\gamma}{1-\gamma} L_1^2 A_{max}}{(1-\gamma)(\lambda - (6L_1^2 + 2L_2)A_{max})^2} \left( \frac{(40L_1^3 + 16L_1 L_2 + 2L_3)(\lambda + \frac{2\gamma}{1-\gamma} L_1^2 A_{max})A_{max}}{(1-\gamma)(\lambda - (6L_1^2 + 2L_2)A_{max})} \right)$$

$$+ \frac{2\gamma}{1-\gamma} L_1(6L_1^2 + 2L_2)A_{max}).$$

Simplify the inequality by $\gamma < 1$ and $1 - \gamma < 0$,

$$\|\nabla_\phi^2 \theta'_\tau\| \leq \frac{(80L_1^3 + 28L_1 L_2 + 2L_3)(\lambda + \frac{2\gamma}{1-\gamma} L_1^2 A_{max})^2}{(1-\gamma)^3(\lambda - (6L_1^2 + 2L_2)A_{max})^2}$$

Then,

$$\int_{\mathcal{A}} \|\nabla_\phi^2 \hat{\pi}_{\theta'_\tau}(a|s)\| da \leq \frac{(160L_1^3 + 56L_1 L_2 + 4L_3)(\lambda + \frac{2\gamma}{1-\gamma} L_1^2 A_{max})^2}{(1-\gamma)^3(\lambda - (6L_1^2 + 2L_2)A_{max})^2}.$$

$\square$

**Lemma 15.** *Suppose that Assumptions 1, 2, and 3 hold. Let $\hat{\pi}_{\theta'_\tau} = \mathcal{A}lg(\hat{\pi}_\phi, \lambda, \tau)$ where $D_\tau = D_{\tau,3}$, we have*

$$\|\nabla_\phi^2 J_\tau(\hat{\pi}_{\theta'_\tau})\| \leq \frac{r_{max} B}{(1-\gamma)^2} + \frac{2\gamma r_{max} C^2}{(1-\gamma)^3}, \tag{44}$$

*where $C = \frac{2L_1(\lambda + \frac{2\gamma}{1-\gamma} L_1^2 A_{max})}{(1-\gamma)(\lambda - (6L_1^2 + 2L_2)A_{max})}$ and $B = \frac{(160L_1^3 + 56L_1 L_2 + 4L_3)(\lambda + \frac{2\gamma}{1-\gamma} L_1^2 A_{max})^2}{(1-\gamma)^3(\lambda - (6L_1^2 + 2L_2)A_{max})^2}$.*

*Proof.* Similar to the proofs of Lemma 8 and Lemma 11 by using Lemma 14. $\square$

**Theorem 7.** *In both discrete and continuous action space, consider the softmax policy with function approximation shown in Section 5.1, and the within-task algorithm $\mathcal{A}lg$ is defined in (2) with $D_\tau = D_{\tau,3}$. Suppose that Assumptions 1, 2, and 3 hold. If $\lambda > (6L_1^2 + 2L_2)A_{max}$, then $\nabla_\phi J_\tau(\mathcal{A}lg^{(3)}(\hat{\pi}_\phi, \lambda, \tau))$ exists for any $\phi$.*

*Let $\{\phi_t\}_{t=1}^T$ be the sequence generated by Algorithm 1 with $\lambda > (6L_1^2 + 2L_2)A_{max}$ and the step size*

$$\alpha = \min\left\{\left(\frac{r_{max}B}{(1-\gamma)^2} + \frac{2\gamma r_{max}C^2}{(1-\gamma)^3}\right)^{-1}, \frac{1}{G\sqrt{T}}\right\}.$$

*Then, the following bound holds:*

$$\frac{1}{T}\sum_{t=1}^T \mathbb{E}_t\left[\|\nabla_\phi \mathbb{E}_{\tau \sim \mathbb{P}(\Gamma)}[J_\tau(\mathcal{A}lg(\hat{\pi}_{\phi_t}, \lambda, \tau))]\|^2\right]$$

$$\leq \left(\frac{2r_{max}^2 B}{(1-\gamma)^3} + \frac{4\gamma r_{max}^2 C^2}{(1-\gamma)^4}\right)\frac{1}{T} + \left(\frac{2r_{max}}{1-\gamma} + \frac{r_{max}B}{(1-\gamma)^2} + \frac{2\gamma r_{max}C^2}{(1-\gamma)^3}\right)\frac{G}{\sqrt{T}},$$

*where*

$$G = \frac{L_1 A_{max}(\lambda + \frac{2\gamma}{1-\gamma}L_1^2 A_{max})}{(1-\gamma)(\lambda - (6L_1^2 + 2L_2)A_{max})},$$

$$C = \frac{2L_1(\lambda + \frac{2\gamma}{1-\gamma}L_1^2 A_{max})}{(1-\gamma)(\lambda - (6L_1^2 + 2L_2)A_{max})},$$

*and*

$$B = \frac{(160L_1^3 + 56L_1 L_2 + 4L_3)(\lambda + \frac{2\gamma}{1-\gamma}L_1^2 A_{max})^2}{(1-\gamma)^3(\lambda - (6L_1^2 + 2L_2)A_{max})^2}.$$

*Proof.* Similar to the proof of Theorem 5, by using the gradient bound in Lemma 12 and the smoothness in Lemma 15. $\qquad\square$

## N  Optimality of one-time policy adaptation

### N.1  Important Lemmas

**Lemma 16.** *Suppose that Assumptions 1, 2 hold. For any task $\tau$, and any policies $\pi$ and $\pi'$, the following bound holds:*

$$\frac{1}{1-\gamma}\mathop{\mathbb{E}}_{\substack{s \sim \nu_\tau^\pi \\ a \sim \pi'(\cdot|s)}}[A_\tau^\pi(s,a)] - C_\tau^\pi(\pi') \leq J_\tau(\pi') - J_\tau(\pi) \leq \frac{1}{1-\gamma}\mathop{\mathbb{E}}_{\substack{s \sim \nu_\tau^\pi \\ a \sim \pi'(\cdot|s)}}[A_\tau^\pi(s,a)] + C_\tau^\pi(\pi')$$

*where*

$$C_\tau^\pi(\pi') = \frac{4\gamma A_{max}}{(1-\gamma)^2}D_{TV}^{max}(\pi\|\pi')\mathbb{E}_{s \sim \nu_\tau^\pi}[D_{TV}(\pi(\cdot|s)\|\pi'(\cdot|s))].$$

*Here, we define $D_{TV}(\pi(\cdot|s)\|\pi'(\cdot|s)) \triangleq \frac{1}{2}\sum_{a \in \mathcal{A}}|\pi(a|s) - \pi'(a|s)|$ in a discrete action space or $D_{TV}(\pi(\cdot|s)\|\pi'(\cdot|s)) \triangleq \frac{1}{2}\int_{a \in \mathcal{A}}|\pi(a|s) - \pi'(a|s)|da$ in a continuous action space, and $D_{TV}^{max}(\pi\|\pi') \triangleq \max_{s \in \mathcal{S}} D_{TV}(\pi(\cdot|s)\|\pi'(\cdot|s))$.*

*Proof.* Let $P_\tau^\pi$ is a matrix where $P_\tau^\pi(i,j) = \mathbb{E}_{a \sim \pi(\cdot|s_i)}P_\tau(s_j|s_i, a)$ and $P_\tau^{\pi'}$ is a matrix where $P_\tau^{\pi'}(i,j) = \mathbb{E}_{a \sim \pi'(\cdot|s_i)}P_\tau(s_j|s_i, a)$. Let $G = (1 + \gamma P_\tau^\pi + (\gamma P_\tau^\pi)^2 + \ldots) = (1 - \gamma P_\tau^\pi)^{-1}$, and similarly $\tilde{G} = (1 + \gamma P_\tau^{\pi'} + (\gamma P_\tau^{\pi'})^2 + \ldots) = (1 - \gamma P_\tau^{\pi'})^{-1}$. Let $\rho$ be a density vector on state space and $r_\tau$ is a reward function vector on state space, thus $r_\tau^\top \rho$ is a scalar meaning the expected reward under density $\rho$. Note that $J_\tau(\pi) = r_\tau^\top G\rho_\tau$, and $J_\tau(\pi') = r_\tau^\top \tilde{G}\rho_\tau$. Here, $\rho_\tau$ is the initial state distribution for task $\tau$. Let $\Delta = P_\tau^{\pi'} - P_\tau^\pi$.

Follow the proof in Appendix B in [51], we have

$$G^{-1} - \tilde{G}^{-1} = (1 - \gamma P_\pi) - (1 - \gamma P_{\tilde{\pi}}) = \gamma\Delta.$$

Left multiply by $\tilde{G}$ and right multiply by $G$,

$$\tilde{G} = \gamma\tilde{G}\Delta G + G. \tag{45}$$

Left multiply by $G$ and right multiply by $\tilde{G}$,

$$\tilde{G} = \gamma G \Delta \tilde{G} + G. \tag{46}$$

Substituting the right-hand side in (45) into $\tilde{G}$ in (46), then

$$\tilde{G} = G + \gamma G \Delta G + \gamma^2 G \Delta \tilde{G} \Delta G.$$

So we have

$$J_\tau(\pi') - J_\tau(\pi) = r_\tau^\top (\tilde{G} - G)\rho_\tau = \gamma r_\tau^\top G \Delta G \rho_\tau + \gamma^2 r_\tau^\top G \Delta \tilde{G} \Delta G \rho_\tau. \tag{47}$$

Note that $r_\tau^\top G = v_\tau^{\pi\,\top}$, where $v$ is the value function on the state space. We also have $G\rho_\tau = \frac{1}{1-\gamma}\nu_\tau^\pi$, where $\nu_\tau^\pi$ is the state visitation distribution vector. So,

$$J_\tau(\tilde{\pi}) - J_\tau(\pi) = r_\tau^\top (\tilde{G} - G)\rho_\tau = \frac{\gamma}{1-\gamma}v_\tau^{\pi\,\top}\Delta\nu_\tau^\pi + \frac{\gamma^2}{1-\gamma}v_\tau^{\pi\,\top}\Delta\tilde{G}\Delta\nu_\tau^\pi.$$

Consider the first term $\frac{\gamma}{1-\gamma}v_\tau^{\pi\,\top}\Delta\nu_\tau^\pi$, similar to Equation (50) in [51], we have

$$\begin{aligned}
\gamma v_\tau^{\pi\,\top}\Delta\nu_\tau^\pi &= v_\tau^{\pi\,\top}(P_\tau^{\pi'} - P_\tau^\pi)\nu_\tau^\pi \\
&= \sum_s \nu_\tau^\pi(s) \sum_{s'} \sum_a (\pi'(a|s) - \pi(a|s))P_\tau(s'|s,a)\gamma v_\tau^\pi(s') \\
&= \sum_s \nu_\tau^\pi(s) \sum_a (\pi'(a|s) - \pi(a|s))\left[r(s) + \sum_{s'} P_\tau(s'|s,a)\gamma v_\tau^\pi(s') - v(s)\right] \\
&= \sum_s \nu_\tau^\pi(s) \sum_a (\pi'(a|s) - \pi(a|s))A_\tau^\pi(s,a)
\end{aligned} \tag{48}$$

Since we have $\sum_a \pi(a|s)A_\tau^\pi(s,a) = 0$, we have

$$\gamma v_\tau^{\pi\,\top}\Delta\nu_\tau^\pi = \sum_s \nu_\tau^\pi(s) \sum_a \pi'(a|s)A_\tau^\pi(s,a) = \mathop{\mathbb{E}}_{\substack{s\sim\nu_\tau^\pi \\ a\sim\pi'(\cdot|s)}} [A_\tau^\pi(s,a)].$$

Combine (47) and the above equation, we have the following for the second term:

$$\frac{\gamma^2}{1-\gamma}v_\tau^{\pi\,\top}\Delta\tilde{G}\Delta\nu_\tau^\pi = J_\tau(\pi') - J_\tau(\pi) - \frac{1}{1-\gamma}\mathop{\mathbb{E}}_{\substack{s\sim\nu_\tau^\pi \\ a\sim\pi'(\cdot|s)}} [A_\tau^\pi(s,a)].$$

Then we need to show

$$\left|\frac{\gamma^2}{1-\gamma}v_\tau^{\pi\,\top}\Delta\tilde{G}\Delta\nu_\tau^\pi\right| \le C_\tau^\pi(\pi').$$

First, by Hölder's inequality,

$$\left|\frac{\gamma^2}{1-\gamma}v_\tau^{\pi\,\top}\Delta\tilde{G}\Delta\nu_\tau^\pi\right| \le \frac{\gamma}{1-\gamma}\|\gamma v_\tau^{\pi\,\top}\Delta\|_\infty\|\tilde{G}\Delta\nu_\tau^\pi\|_1.$$

Similar to (48), each element in the vector $\gamma v_\tau^{\pi\,\top}\Delta$ is $\sum_a(\pi'(a|s) - \pi(a|s))A_\tau^\pi(s,a)$, then we have

$$\|\gamma v_\tau^{\pi\,\top}\Delta\|_\infty \le \sum_a |\pi'(a|s) - \pi(a|s)|A_\tau^\pi(s,a) \le 2A_{max}D_{TV}^{max}(\pi\|\pi').$$

From the Lemma 3 of [1], we have

$$\|\tilde{G}\Delta\nu_\tau^\pi\|_1 \le \frac{2}{1-\gamma}\mathbb{E}_{s\sim\nu_\tau^\pi}[D_{TV}(\pi(\cdot|s)\|\pi'(\cdot|s))].$$

Therefore, we have

$$\left|\frac{\gamma^2}{1-\gamma}v_\tau^{\pi\,\top}\Delta\tilde{G}\Delta\nu_\tau^\pi\right| \le C_\tau^\pi(\pi') = \frac{4\gamma A_{max}}{(1-\gamma)^2}D_{TV}^{max}(\pi\|\pi')\mathbb{E}_{s\sim\nu_\tau^\pi}[D_{TV}(\pi(\cdot|s)\|\pi'(\cdot|s))].$$

Then the bounds hold.

$\square$

**Lemma 17.** *Suppose that Assumptions 1, 2 hold. For any task $\tau$, any bounded parameters $\theta$ and $\theta'$, and $i = 1$ or $2$, the following bound holds for both $i = 1$ and $2$:*

$$J_\tau(\hat{\pi}_{\theta'}) - J_\tau(\hat{\pi}_\theta) \leq \frac{1}{1-\gamma} \mathop{\mathbb{E}}_{\substack{s \sim \nu_\tau^{\hat{\pi}_\theta} \\ a \sim \hat{\pi}_{\theta'}(\cdot|s)}} \left[ A_\tau^{\hat{\pi}_\theta}(s,a) \right] + \frac{2\gamma A_{max}}{(1-\gamma)^2 \epsilon} D_{\tau,i}^2(\hat{\pi}_\theta, \hat{\pi}_{\theta'})$$

*and*

$$J_\tau(\hat{\pi}_{\theta'}) - J_\tau(\hat{\pi}_\theta) \geq \frac{1}{1-\gamma} \mathop{\mathbb{E}}_{\substack{s \sim \nu_\tau^{\hat{\pi}_\theta} \\ a \sim \hat{\pi}_{\theta'}(\cdot|s)}} \left[ A_\tau^{\hat{\pi}_\theta}(s,a) \right] - \frac{2\gamma A_{max}}{(1-\gamma)^2 \epsilon} D_{\tau,i}^2(\hat{\pi}_\theta, \hat{\pi}_{\theta'}).$$

*Proof.* The proof follows similar lines of Theorem 1 in [51] and Corollary 1 and 2 in [1]. For the sake of self-containedness, we provide the complete proof.

We show the first inequality. The second inequality follows a similar way. From Lemma 16,

$$J_\tau(\hat{\pi}_{\theta'}) - J_\tau(\hat{\pi}_\theta) - \frac{1}{1-\gamma} \mathop{\mathbb{E}}_{\substack{s \sim \nu_\tau^{\hat{\pi}_\theta} \\ a \sim \hat{\pi}_{\theta'}(\cdot|s)}} \left[ A_\tau^{\hat{\pi}_\theta}(s,a) \right] \leq \frac{4\gamma A_{max}}{(1-\gamma)^2} D_{TV}^{max}(\hat{\pi}_\theta || \hat{\pi}_{\theta'}) \mathbb{E}_{s \sim \nu_\tau^{\hat{\pi}_\theta}} \left[ D_{TV}(\hat{\pi}_\theta(\cdot|s) || \hat{\pi}_{\theta'}(\cdot|s)) \right].$$

From Assumption 2, $\nu^{\hat{\pi}_\theta}(s) \geq \epsilon$ for any $s \in \mathcal{A}$. Also, $D_{TV}(\hat{\pi}_\theta(\cdot|s) || \hat{\pi}_{\theta'}(\cdot|s)) \geq 0$ for any $s \in \mathcal{A}$. Then, we have

$$\epsilon D_{TV}^{max}(\hat{\pi}_\theta || \hat{\pi}_{\theta'}) \leq \mathbb{E}_{s \sim \nu_\tau^{\hat{\pi}_\theta}} \left[ D_{TV}(\hat{\pi}_\theta(\cdot|s) || \hat{\pi}_{\theta'}(\cdot|s)) \right].$$

From Jensen's inequality, we have

$$\mathbb{E}_{s \sim \nu_\tau^{\hat{\pi}_\theta}} \left[ D_{TV}(\hat{\pi}_\theta(\cdot|s) || \hat{\pi}_{\theta'}(\cdot|s)) \right]^2 \leq \mathbb{E}_{s \sim \nu_\tau^{\hat{\pi}_\theta}} \left[ D_{TV}^2(\hat{\pi}_\theta(\cdot|s) || \hat{\pi}_{\theta'}(\cdot|s)) \right].$$

From the above three inequalities, we have

$$J_\tau(\hat{\pi}_{\theta'}) - J_\tau(\hat{\pi}_\theta) - \frac{1}{1-\gamma} \mathop{\mathbb{E}}_{\substack{s \sim \nu_\tau^{\hat{\pi}_\theta} \\ a \sim \hat{\pi}_{\theta'}(\cdot|s)}} \left[ A_\tau^{\hat{\pi}_\theta}(s,a) \right] \leq \frac{4\gamma A_{max}}{(1-\gamma)^2 \epsilon} \mathbb{E}_{s \sim \nu_\tau^{\hat{\pi}_\theta}} \left[ D_{TV}^2(\hat{\pi}_\theta(\cdot|s) || \hat{\pi}_{\theta'}(\cdot|s)) \right].$$

(49)

From [8], we have

$$D_{TV}^2(\hat{\pi}_\theta(\cdot|s) || \hat{\pi}_{\theta'}(\cdot|s)) \leq \frac{1}{2} D_{KL}(\hat{\pi}_\theta(\cdot|s) || \hat{\pi}_{\theta'}(\cdot|s)),$$

and

$$D_{TV}^2(\hat{\pi}_\theta(\cdot|s) || \hat{\pi}_{\theta'}(\cdot|s)) \leq \frac{1}{2} D_{KL}(\hat{\pi}_{\theta'}(\cdot|s) || \hat{\pi}_\theta(\cdot|s)).$$

Therefore,

$$J_\tau(\hat{\pi}_{\theta'}) - J_\tau(\hat{\pi}_\theta) \leq \frac{1}{1-\gamma} \mathop{\mathbb{E}}_{\substack{s \sim \nu_\tau^{\hat{\pi}_\theta} \\ a \sim \hat{\pi}_{\theta'}(\cdot|s)}} \left[ A_\tau^{\hat{\pi}_\theta}(s,a) \right] + \frac{2\gamma A_{max}}{(1-\gamma)^2 \epsilon} D_{\tau,1}^2(\hat{\pi}_\theta, \hat{\pi}_{\theta'}),$$

and

$$J_\tau(\hat{\pi}_{\theta'}) - J_\tau(\hat{\pi}_\theta) \leq \frac{1}{1-\gamma} \mathop{\mathbb{E}}_{\substack{s \sim \nu_\tau^{\hat{\pi}_\theta} \\ a \sim \hat{\pi}_{\theta'}(\cdot|s)}} \left[ A_\tau^{\hat{\pi}_\theta}(s,a) \right] + \frac{2\gamma A_{max}}{(1-\gamma)^2 \epsilon} D_{\tau,2}^2(\hat{\pi}_\theta, \hat{\pi}_{\theta'}).$$

$\square$

**Lemma 18.** *Consider the softmax policy with function approximation shown in Section 5.1. Suppose that Assumptions 1, 2, and 3 hold. For any task $\tau$, and any softmax policies parameterized by bounded $\theta$ and $\theta'$, the following bound holds:*

$$J_\tau(\hat{\pi}_{\theta'}) - J_\tau(\hat{\pi}_\theta) \leq \frac{1}{1-\gamma} \mathop{\mathbb{E}}_{\substack{s \sim \nu_\tau^{\hat{\pi}_\theta} \\ a \sim \hat{\pi}_{\theta'}(\cdot|s)}} \left[ A_\tau^{\hat{\pi}_\theta}(s,a) \right] + \frac{4\gamma A_{max} L_1^2}{(1-\gamma)^2 \epsilon} ||\theta - \theta'||^2$$

*and*

$$J_\tau(\hat{\pi}_{\theta'}) - J_\tau(\hat{\pi}_\theta) \geq \frac{1}{1-\gamma} \mathop{\mathbb{E}}_{\substack{s \sim \nu_\tau^{\hat{\pi}_\theta} \\ a \sim \hat{\pi}_{\theta'}(\cdot|s)}} \left[ A_\tau^{\hat{\pi}_\theta}(s,a) \right] - \frac{4\gamma A_{max} L_1^2}{(1-\gamma)^2 \epsilon} ||\theta - \theta'||^2.$$

*Proof.* From (36), for any $\theta \in \mathbb{R}^n$,

$$\nabla_\theta \hat{\pi}_\theta(a|s) = \hat{\pi}_\theta(a|s)\nabla_\theta f_\theta(s,a) - \hat{\pi}_\theta(a|s)\frac{\int_\mathcal{A} \nabla_\theta f_\theta(s,a')\exp\left(f_\theta(s,a')\right)da'}{\int_\mathcal{A}\exp\left(f_\theta(s,a')\right)da'}.$$

Then,

$$\|\nabla_\theta \hat{\pi}_\theta(a|s)\| \le \hat{\pi}_\theta(a|s)\|\nabla_\theta f_\theta(s,a)\| + \hat{\pi}_\theta(a|s)\left\|\frac{\int_\mathcal{A}\nabla_\theta f_\theta(s,a')\exp\left(f_\theta(s,a')\right)da'}{\int_\mathcal{A}\exp\left(f_\theta(s,a')\right)da'}\right\|$$

$$\le 2\hat{\pi}_\theta(a|s)L_1$$

From the mean value theorem, we have

$$|\hat{\pi}_\theta(a|s) - \hat{\pi}_{\theta'}(a|s)| \le 2\hat{\pi}_{\phi(a)}(a|s)L_1\|\theta - \theta'\|,$$

where $\phi(a) = \delta(a)\theta + (1-\delta(a))\theta'$ and $0 \le \delta(a) \le 1$. So,

$$\frac{1}{2}\sum_{a\in\mathcal{A}}|\hat{\pi}_\theta(a|s) - \hat{\pi}_{\theta'}(a|s)| \le L_1\|\theta - \theta'\|.$$

From (49), we have

$$J_\tau(\hat{\pi}_{\theta'}) - J_\tau(\hat{\pi}_\theta) - \frac{1}{1-\gamma}\underset{\substack{s\sim\nu_\tau^{\hat{\pi}_\theta}\\a\sim\hat{\pi}_{\theta'}(\cdot|s)}}{\mathbb{E}}\left[A_\tau^{\hat{\pi}_\theta}(s,a)\right] \le \frac{4\gamma A_{max}L_1^2}{(1-\gamma)^2\epsilon}\|\theta - \theta'\|^2.$$

We use the same way to show another inequality. $\qquad\square$

## N.2 Proof of Theorems 3 and 4

*Proof of Theorem 3.* When the requirement of Theorem 1, $\lambda \ge 2A_{max}$, is satisfied, From Assumption 4 and Theorem 1, for both $i = 1$ and 2,

$$\frac{1}{T}\sum_{t=1}^T \mathbb{E}_t\left[\max_\phi \mathbb{E}_{\tau\sim\mathbb{P}(\Gamma)}[J_\tau(\mathcal{A}lg^{(i)}(\hat{\pi}_\phi,\lambda,\tau)) - \mathbb{E}_{\tau\sim\mathbb{P}(\Gamma)}[J_\tau(\mathcal{A}lg^{(i)}(\hat{\pi}_{\phi_t},\lambda,\tau))]]\right]$$

$$\le\frac{1}{T}\sum_{t=1}^T\mathbb{E}_t\left[h_i\left(\|\nabla_\phi\mathbb{E}_{\tau\sim\mathbb{P}(\Gamma)}[J_\tau(\mathcal{A}lg^{(i)}(\hat{\pi}_{\phi_t},\lambda,\tau))]\|^2\right)\right]$$

$$\le h_i\left(\frac{1}{T}\sum_{t=1}^T\mathbb{E}_t\left[\|\nabla_\phi\mathbb{E}_{\tau\sim\mathbb{P}(\Gamma)}[J_\tau(\mathcal{A}lg^{(i)}(\hat{\pi}_{\phi_t},\lambda,\tau))]\|^2\right]\right)$$

$$\le h_i\left(\frac{K_i}{T} + \frac{M_i}{\sqrt{T}}\right)$$

(50)

where the constants $K_i$ and $M_i$ are shown in Theorem 1. The last inequality sign comes from that $h_i$ is a concave function and Jensen's inequality.

Let $\hat{\pi}_{\theta'_\tau}(\phi) = \mathcal{A}lg^{(i)}(\hat{\pi}_\phi,\lambda,\tau)$ for any meta-parameter $\phi$. From the definition of the within-task algorithm, we have

$$\underset{\substack{s\sim\nu_\tau^{\hat{\pi}_\phi}\\a\sim\hat{\pi}_{\theta'_\tau}(\phi)(\cdot|s)}}{\mathbb{E}}\left[Q_\tau^{\hat{\pi}_\phi}(s,a)\right] - \lambda D_{\tau,i}^2(\hat{\pi}_\phi,\hat{\pi}_{\theta'_\tau}(\phi)) \ge \underset{\substack{s\sim\nu_\tau^{\hat{\pi}_\phi}\\a\sim\hat{\pi}_{\theta^*_\tau}(\cdot|s)}}{\mathbb{E}}\left[Q_\tau^{\hat{\pi}_\phi}(s,a)\right] - \lambda D_{\tau,i}^2(\hat{\pi}_\phi,\hat{\pi}_{\theta^*_\tau}).$$

This is equivalent to

$$\underset{\substack{s\sim\nu_\tau^{\hat{\pi}_\phi}\\a\sim\hat{\pi}_{\theta'_\tau}(\phi)(\cdot|s)}}{\mathbb{E}}\left[A_\tau^{\hat{\pi}_\phi}(s,a)\right] - \lambda D_{\tau,i}^2(\hat{\pi}_\phi,\hat{\pi}_{\theta'_\tau}(\phi)) \ge \underset{\substack{s\sim\nu_\tau^{\hat{\pi}_\phi}\\a\sim\hat{\pi}_{\theta^*_\tau}(\cdot|s)}}{\mathbb{E}}\left[A_\tau^{\hat{\pi}_\phi}(s,a)\right] - \lambda D_{\tau,i}^2(\hat{\pi}_\phi,\hat{\pi}_{\theta^*_\tau}).$$

when $\lambda \geq \frac{2\gamma A_{max}}{(1-\gamma)\epsilon}$, from the second inequality in Lemma 17 and the above inequality,

$$
\begin{aligned}
J_\tau(\hat{\pi}_{\theta'_\tau}(\phi)) - J_\tau(\hat{\pi}_\phi) &\geq \frac{1}{1-\gamma} \mathop{\mathbb{E}}_{\substack{s \sim \nu_\tau^{\hat{\pi}_\phi} \\ a \sim \hat{\pi}_{\theta'_\tau}(\phi)(\cdot|s)}} \left[ A_\tau^{\hat{\pi}_\phi}(s,a) \right] - \frac{2\gamma A_{max}}{(1-\gamma)^2\epsilon} D_{\tau,i}^2(\hat{\pi}_\phi, \hat{\pi}_{\theta'_\tau}(\phi)) \\
&\geq \frac{1}{1-\gamma} \mathop{\mathbb{E}}_{\substack{s \sim \nu_\tau^{\hat{\pi}_\phi} \\ a \sim \hat{\pi}_{\theta'_\tau}(\phi)(\cdot|s)}} \left[ A_\tau^{\hat{\pi}_\phi}(s,a) \right] - \frac{\lambda}{1-\gamma} D_{\tau,i}^2(\hat{\pi}_\phi, \hat{\pi}_{\theta'_\tau}(\phi)) \\
&\geq \frac{1}{1-\gamma} \mathop{\mathbb{E}}_{\substack{s \sim \nu_\tau^{\hat{\pi}_\phi} \\ a \sim \hat{\pi}_{\theta^*_\tau}(\cdot|s)}} \left[ A_\tau^{\hat{\pi}_\phi}(s,a) \right] - \frac{\lambda}{1-\gamma} D_{\tau,i}^2(\hat{\pi}_\phi, \hat{\pi}_{\theta^*_\tau}).
\end{aligned}
$$

From the second inequality in Lemma 17,

$$
J_\tau(\hat{\pi}_{\theta^*_\tau}) - J_\tau(\hat{\pi}_\phi) \leq \frac{1}{1-\gamma} \mathop{\mathbb{E}}_{\substack{s \sim \nu_\tau^{\hat{\pi}_\phi} \\ a \sim \hat{\pi}_{\theta^*_\tau}(\cdot|s)}} \left[ A_\tau^{\hat{\pi}_\phi}(s,a) \right] + \frac{2\gamma A_{max}}{(1-\gamma)^2\epsilon} D_{\tau,i}^2(\hat{\pi}_\phi, \hat{\pi}_{\theta^*_\tau}).
$$

From the last two inequalities,

$$
J_\tau(\hat{\pi}_{\theta'_\tau}(\phi)) - J_\tau(\hat{\pi}_{\theta^*_\tau}) \geq -\left( \frac{2\gamma A_{max}}{(1-\gamma)^2\epsilon} + \frac{\lambda}{1-\gamma} \right) D_{\tau,i}^2(\hat{\pi}_\phi, \hat{\pi}_{\theta^*_\tau}),
$$

i.e.,

$$
J_\tau(\hat{\pi}_{\theta^*_\tau}) - J_\tau(\mathcal{A}lg^{(i)}(\hat{\pi}_\phi, \lambda, \tau)) \leq \left( \frac{2\gamma A_{max}}{(1-\gamma)^2\epsilon} + \frac{\lambda}{1-\gamma} \right) D_{\tau,i}^2(\hat{\pi}_\phi, \hat{\pi}_{\theta^*_\tau}).
$$

Then,

$$
\mathbb{E}_{\tau \sim \mathbb{P}(\Gamma)}[J_\tau(\hat{\pi}_{\theta^*_\tau}) - J_\tau(\mathcal{A}lg^{(i)}(\hat{\pi}_\phi, \lambda, \tau))] \leq \left( \frac{2\gamma A_{max}}{(1-\gamma)^2\epsilon} + \frac{\lambda}{1-\gamma} \right) \mathbb{E}_{\tau \sim \mathbb{P}(\Gamma)}[D_{\tau,i}^2(\hat{\pi}_\phi, \hat{\pi}_{\theta^*_\tau})].
$$

Let $\phi^* = \arg \max_\phi \mathbb{E}_{\tau \sim \mathbb{P}(\Gamma)}[J_\tau(\mathcal{A}lg^{(i)}(\hat{\pi}_\phi, \lambda, \tau))]$, we have

$$
\mathbb{E}_{\tau \sim \mathbb{P}(\Gamma)}[J_\tau(\mathcal{A}lg^{(i)}(\hat{\pi}_{\phi^*}, \lambda, \tau))] \geq \max_\phi \mathbb{E}_{\tau \sim \mathbb{P}(\Gamma)}[J_\tau(\mathcal{A}lg^{(i)}(\hat{\pi}_\phi, \lambda, \tau))].
$$

Therefore,

$$
\begin{aligned}
\mathbb{E}_{\tau \sim \mathbb{P}(\Gamma)}[J_\tau(\hat{\pi}_{\theta^*_\tau}) - J_\tau(\mathcal{A}lg^{(i)}(\hat{\pi}_{\phi^*}, \lambda, \tau))] &\leq \min_\phi \mathbb{E}_{\tau \sim \mathbb{P}(\Gamma)}[J_\tau(\hat{\pi}_{\theta^*_\tau}) - J_\tau(\mathcal{A}lg^{(i)}(\hat{\pi}_\phi, \lambda, \tau))] \\
&\leq \min_\phi \left( \frac{2\gamma A_{max}}{(1-\gamma)^2\epsilon} + \frac{\lambda}{1-\gamma} \right) \mathbb{E}_{\tau \sim \mathbb{P}(\Gamma)}[D_{\tau,i}^2(\hat{\pi}_\phi, \hat{\pi}_{\theta^*_\tau})]
\end{aligned}
$$

Since

$$
\min_\phi \mathbb{E}_{\tau \sim \mathbb{P}(\Gamma)}[D_{\tau,i}^2(\hat{\pi}_\phi, \hat{\pi}_{\theta^*_\tau})] = \mathcal{V}ar_i(\mathbb{P}(\Gamma)),
$$

we have

$$
\mathbb{E}_{\tau \sim \mathbb{P}(\Gamma)}[J_\tau(\hat{\pi}_{\theta^*_\tau}) - J_\tau(\mathcal{A}lg^{(i)}(\hat{\pi}_{\phi^*}, \lambda, \tau))] \leq \left( \frac{2\gamma A_{max}}{(1-\gamma)^2\epsilon} + \frac{\lambda}{1-\gamma} \right) \mathcal{V}ar_i(\mathbb{P}(\Gamma)).
$$

Note that in the above analysis, we need $\lambda \geq 2A_{max}$ and also $\lambda \geq \frac{2\gamma A_{max}}{(1-\gamma)\epsilon}$. So, we select we select $\lambda = \frac{2A_{max}}{(1-\gamma)\epsilon}$ to satisfy the requirement. When $\lambda = \frac{2A_{max}}{(1-\gamma)\epsilon}$, we have

$$
\mathbb{E}_{\tau \sim \mathbb{P}(\Gamma)}[J_\tau(\hat{\pi}_{\theta^*_\tau}) - J_\tau(\mathcal{A}lg^{(i)}(\hat{\pi}_{\phi^*}, \lambda, \tau))] \leq \frac{2(1+\gamma)A_{max}}{(1-\gamma)^2\epsilon} \mathcal{V}ar_i(\mathbb{P}(\Gamma)). \tag{51}
$$

From (50) and (51) we have

$$
\begin{aligned}
\frac{1}{T} \sum_{t=1}^T \mathbb{E}_t &\left[ \mathbb{E}_{\tau \sim \mathbb{P}(\Gamma)}[J_\tau(\hat{\pi}_{\theta^*_\tau}) - J_\tau(\mathcal{A}lg^{(i)}(\hat{\pi}_{\phi_t}, \lambda, \tau))] \right] \\
&\leq h_i \left( \frac{K_i}{T} + \frac{M_i}{\sqrt{T}} \right) + \frac{2(1+\gamma)A_{max}}{(1-\gamma)^2\epsilon} \mathcal{V}ar_i(\mathbb{P}(\Gamma)).
\end{aligned}
$$

$\square$

*Proof of Theorem 4.* Similar to the above proof of Theorem 3. The difference is using two inequalities in Lemma 18 instead of those in Lemma 17 and using Theorem 2 for convergence instead of Theorem 1.

The requirement of Theorem 2 is $\lambda > (6L_1^2 + 2L_2)A_{max}$, and the requirement of Lemma 18 is $\lambda \geq \frac{4\gamma A_{max}L_1^2}{(1-\gamma)\epsilon}$. Therefore, we select $\lambda = \frac{(6L_1^2+2L_2)A_{max}}{(1-\gamma)\epsilon}$. Then, the bound is

$$\frac{1}{T}\sum_{t=1}^{T}\mathbb{E}_t\left[\mathbb{E}_{\tau\sim\mathbb{P}(\Gamma)}[J_\tau(\hat{\pi}_{\theta_\tau^*}) - J_\tau(\mathcal{A}lg^{(3)}(\hat{\pi}_{\phi_t},\lambda,\tau))]\right]$$

$$\leq h_3\left(\frac{K_3}{T} + \frac{M_3}{\sqrt{T}}\right) + \left(\frac{4\gamma L_1^2 A_{max}}{(1-\gamma)^2\epsilon} + \frac{\lambda}{1-\gamma}\right)\mathcal{V}ar_3(\mathbb{P}(\Gamma)),$$

$$\leq h_3\left(\frac{K_3}{T} + \frac{M_3}{\sqrt{T}}\right) + \frac{((6+4\gamma)L_1^2 + 2L_2)A_{max}}{(1-\gamma)^2\epsilon}\mathcal{V}ar_3(\mathbb{P}(\Gamma)),$$

$\square$

### N.3 Clarification of $A_{max}$

In all the proofs in Sections N.1 and N.1, we can replace as $A_{max}$ to $A'_{max}$, where $A'_{max}$ is defined by the maximum advantage function value of policy $\hat{\pi}_{\phi'}$, where $\phi' = \arg\min_\phi \mathbb{E}_{\tau\sim\mathbb{P}(\Gamma)}[D_{\tau,i}^2(\hat{\pi}_\phi, \hat{\pi}_{\theta_\tau^*})]$. It is easy to see $A'_{max} \leq A_{max}$. For simplification of the assumption statements, theorem statements, and convenience of the proofs, we keep $A_{max}$ in the proofs and Theorems 3 and 4. We actually can make the bound in Theorems 3 and 4 tighter by replacing $A_{max}$ to $A'_{max}$. In the verification of the theoretical results of Section 6, we select $\lambda$ based on $A'_{max}$ and verify the tighter bounds by the experiments.

## O Proofs of Remarks

*Proof of part (i) of Remark 1.* If the MDP $\mathcal{M}_\tau$ is ergodic, there exists a policy $\hat{\pi}$ such that $\nu_\tau^{\hat{\pi}}(s) \geq \epsilon_0$. As $\phi$ is bounded, the probability (or probability density) of each action of the softmax policy is larger than 0 and lower bounded by a $\epsilon_1 > 0$. Therefore, the action probability of the policy $\hat{\pi}(a|s)$ can be upper bounded by $\hat{\pi}_\phi(a|s)/\epsilon_1$ for any $a$. Therefore, $\nu_\tau^{\hat{\pi}_\phi}(s) \geq \epsilon_0/\epsilon_1$. $\square$

*Proof of part (ii) of Remark 1.* If the initial state distribution $\rho_\tau$ has $\rho_\tau(s) > 0$ for any $s \in \mathcal{S}$. Since $\mathcal{S}$ is bounded, $\rho_\tau(s) \geq \epsilon_2$ for any $s \in \mathcal{S}$. Then, $\nu_\tau^{\hat{\pi}_\phi}(s) \geq (1-\gamma)\epsilon_2$. $\square$

## P Limitations

In this paper, we provide several theorems, where the hyper-parameter selection, e.g., $\lambda$, is provided by the theorems. The theoretical analysis usually chooses hyper-parameters, which are sometimes conservative. In practice, we can tune them to improve the performance.

