# OpenReview forum: "Meta-Reinforcement Learning with Universal Policy Adaptation: Provable Near-Optimality under All-task Optimum Comparator"
_NeurIPS.cc/2024/Conference — NeurIPS 2024 poster_

### Official Review · Reviewer_8Th6 · 2024-07-10

**Soundness:** 3
**Presentation:** 3
**Contribution:** 3
**Rating:** 6
**Confidence:** 4

**Summary:**

This paper presents a bilevel optimization framework for meta-reinforcement learning (Meta-RL) named BO-MRL, which aims to enhance policy adaptation through a universal policy optimization algorithm. The framework is designed to improve data efficiency by implementing multiple policy optimization steps on a single data collection during task-specific adaptation. The authors provide theoretical guarantees, including upper bounds on the expected optimality gap over a task distribution, which measure the distance between the adapted policy and the task-specific optimal policy. Empirical validation is conducted on several benchmarks, demonstrating the superior effectiveness of the proposed algorithm compared to existing methods.

**Strengths:**

### S1. Theoretical Contributions

The paper provides a solid theoretical foundation for the proposed bilevel optimization framework, including convergence guarantees and upper bounds on the expected optimality gap. This strengthens the understanding of the method's performance and its theoretical underpinnings.

### S2. Empirical Validation

The empirical results presented in the paper are comprehensive and robust, showing significant improvements over state-of-the-art methods on various benchmarks. This demonstrates the practical applicability and effectiveness of the proposed framwork.

### S3. Novel approach

The introduction of a universal policy optimization algorithm within a bilevel optimization framework is a novel and effective approach to tackling the challenges in Meta-RL. This method addresses both the optimality and data-inefficiency issues commonly faced by existing methods.

**Weaknesses:**

### W1. Complexity of Implementation

The proposed framework, while theoretically sound and empirically validated, may be complex to implement in practice. The bi-level optimization and the need for hypergradient computation require careful tuning and expertise, which might limit its accessibility to a broader audience.

### W2. Limited discussion on practical implications

The paper could benefit from a more detailed discussion on the practical implications and potential limitations of the proposed method in real-world applications. This includes considerations such as computational resources, scalability to large-scale environments, and adaptability to varying task complexities.

### W3. Comparison with Broader Range of Methods

While the paper compares the proposed method with several state-of-the-art Meta-RL algorithms, it would be beneficial to include a broader range of methods, especially those outside the immediate Meta-RL context, to provide a more comprehensive evaluation of its performance.

**Questions:**

### Q1. Hyperparameter Sensitivity

How sensitive is the proposed method to the choice of hyperparameters?

### Q2. Scalability

Can the authors provide more insights into the scalability of the proposed method? How does the computational complexity scale with the number of tasks and the size of the state-action space?

### Q3. Practical Deployment

What are the practical considerations for deploying the proposed method in real-world scenarios? Are there any specific challenges or requirements that practitioners should be aware of?

**Limitations:**

The authors have adequately addressed the theoretical limitations and the assumptions underlying their framework. However, a more detailed discussion on the potential negative societal impacts and how to mitigate them would be beneficial.

---

> ### Author Rebuttal · Authors · 2024-08-07
>
> Thanks very much for your time and effort in reviewing our work. Thanks for your suggestions to make our manuscript better. We address your concerns as follows.
>
> >**Weakness 1. Complexity of Implementation The proposed framework, while theoretically sound and empirically validated, may be complex to implement in practice. The bi-level optimization and the need for hypergradient computation require careful tuning and expertise, which might limit its accessibility to a broader audience.**
>
> **Answer:** We propose a practical algorithm (Algorithm 2, lines 628) to simplify the hypergradient computation in Propositions 1 and 2 (lines 199 and 212). In the simplified hypergradient computation, the inverse Hessian matrix is computed by the conjugate gradient, which can be implemented in many standard Python libraries, such as Scipy. The remaining parts of the gradient computation can be handled by the Autograd function in PyTorch.
>
> The experiments in our paper adopt the conjugate gradient. The experiment results validate the computational efficiency of our algorithm. In the experiment of Half-cheetah, the computation time of the hyper-gradient with the inverse of Hessian for a three-layer neural network is about $0.3$ second in each meta-parameter update, where we use only the CPU to compute the hyper-gradient.
> This approach has demonstrated high efficiency across a wide range of applications, including several widely used RL algorithms, such as TRPO [44] and CPO [1]. The details are shown in Appendix C of [44]. In the simplest meta-RL method, MAML [13], the authors use the TRPO to update the meta-parameter, as shown in Section 5.3 of [13], the inverse of the Hessian is computed in a similar way to ours. Therefore, the computational complexity of the hyper-gradient in our proposed method is comparable to many existing RL and meta-RL approaches, which are shown efficient.
>
> >**Weakness 2. Limited discussion on practical implications. The paper could benefit from a more detailed discussion on the practical implications and potential limitations of the proposed method in real-world applications. This includes considerations such as computational resources, scalability to large-scale environments, and adaptability to varying task complexities.**
>
> **Answer:** In the experiment of the proposed algorithm, we only use the CPU to do meta-training and meta-tests. After the meta-model is trained, the policy adaptation only takes a few seconds (2 seconds). Therefore, the requirement of the computational resources for the proposed algorithm is low.
>
> In the experiment section (Section 6, line 362), we conduct the experiments on relatively simple grid-world environments (lines 362, the results are shown in Figure 1,), and also on the complex high-dimensional locomotion tasks (lines 373, the results are shown in Figure 2). The experimental results demonstrate the superior performance of the proposed algorithm, highlighting its good adaptability to varying task complexities.
>
> The proposed algorithm holds the scalability to large-scale environments and tasks. Please refer to **the answer to Question 2** for more details.
>
> A potential limitation in the practice is that the design of the reward functions for multiple tasks could be challenging, and requires lots of efforts of experts.
>
> >**Weakness 3. Comparison with Broader Range of Methods. It would be beneficial to include a broader range of methods, especially those outside the immediate Meta-RL context, to provide a more comprehensive evaluation of its performance.**
>
> **Answer:** The manuscript focuses on meta-RL and compares it with the existing meta-RL algorithms [13,43,48]. It is typical for the meta-RL papers to compare within the context of meta-RL.
>
> >**Question 1. Hyperparameter Sensitivity. How sensitive is the proposed method to the choice of hyperparameters?**
>
> **Answer:** As the performance of RL algorithms can be sensitive to the choice of hyperparameters, as shown in [3.a], the proposed method could hold a similar property. However, the tuning of the hyperparameters is easy. In particular, in the proposed method, the hyperparameters that are required to be tuned often appear in existing widely used meta-RL and RL algorithms, such as TRPO. We can follow the previous standard choice of them.
>
> [3.a] Eimer, Theresa, et al. "Hyperparameters in reinforcement learning and how to tune them." International Conference on Machine Learning, 2023.
>
> >**Question 2. Scalability. Can the authors provide more insights into the scalability of the proposed method? How does the computational complexity scale with the number of tasks and the size of the state-action space?**
>
> **Answer:** Overall, the computational complexity of the proposed method is comparable to many existing meta-RL algorithms, including MAML [13]. More details about the computational complexity analysis are shown in Appendix E and F (lines 633-694).
>
> In terms of the scalability to a large number of tasks, the applied stochastic optimization algorithm, Adam, has shown good performance on a huge amount of data, e.g., millions of data points.
> In terms of the scalability to large state-action space, we apply the neural network as the approximation function of the action policy, which can deal with large-scale problems.
> In the experiment, the dimension of the continuous state-action of the Ant environment is 35, which is sufficiently large for most RL problems.
>
> >**Question 3. Practical Deployment.
> What are the practical considerations for deploying the proposed method in real-world scenarios? Are there any specific challenges or requirements that practitioners should be aware of?**
>
> **Answer:** Meta-RL algorithms have been widely applied to many real-world scenarios, such as robotics [6, 40]. In terms of real-world practice, our algorithm is similar to these meta-RL algorithms and has no extra requirement for practitioners.

---

> > ### Comment · Reviewer_8Th6 · 2024-08-11
> > **Acknowledgement**
> >
> > I appreciate the authors for the detailed responses.
> >
> > I still think this is a good submission worth an acceptance, so I maintain my assessment.

---

### Official Review · Reviewer_DbXf · 2024-07-12

**Soundness:** 3
**Presentation:** 3
**Contribution:** 2
**Rating:** 5
**Confidence:** 3

**Summary:**

This paper proposes an optimization framework to learn the meta-prior for task-specific policy adaptation.

**Strengths:**

1. The proposed method claims to be data-efficient in terms of data collection developed only using one-time data collection. Also, this paper aims to solve the RL problem as a bilevel optimization problem.
2. The proposed method considers both unconstrained and constrained optimization problem cases.
4. the upper bounds of the optimality between the adapted policy and the optimal task-specific policy are provided.

**Weaknesses:**

1. the proposed method heavily depends on the minimization problems of eq. 1 or eq. 2, which minimizes the distance between the policy and the predefined policy and finds the optimal policy for the additional task at the same time. My concern and question is that given the same problem that we want to adapt a pre-defined policy \pi to a new task L while keep the original task J, what is the performance difference if i just train the \pi to minimize L+J ?
2. can you plot the algorithm deployment performance for the example in Fig. 2? since sometimes if you only do \min_\theta ||\pi_\theta - K(x)||, where K(x) is a predefined controller, the \theta converges fast, but the achieved \pi_\theta does not work well in some examples.

**Questions:**

see the weaknesses

**Limitations:**

yes, the author has provided that

---

> ### Author Rebuttal · Authors · 2024-08-07
>
> Thanks very much for your time and effort in reviewing our work. Thanks for your suggestions to make our manuscript better. We address your concerns as follows.
>
> >**Weakness 1. The proposed method heavily depends on the minimization problems of eq. 1 or eq. 2, which minimizes the distance between the policy and the predefined policy and finds the optimal policy for the additional task at the same time. My concern and question is that given the same problem that we want to adapt a pre-defined policy $\pi$ to a new task $L$ while keep the original task $J$, what is the performance difference if I just train the $\pi$ to minimize $L+J$?**
>
> **Answer:** The goal of the policy adaptation problem, i.e., Eq. 1 or Eq. 2 (lines 159 and 161), is **not** adapting a pre-defined policy $\pi$ to a new task $L$ while keeping the original tasks $J$.
>
> Minimizing $L + J$ is essentially multi-task learning. The motivation for multi-task learning is that there exists a common solution for the multiple tasks and training on them together could benefit the training for each task. For example, the feature extraction layers for multiple image classification tasks are shared, and training these layers together could help extract better features for each task.
>
> In the problem of this manuscript, we consider meta-learning, where we aim to learn knowledge for a task distribution. The tasks in the task distribution have different goals and there is no shared component that can applied to all tasks. For example, the task of driving a robot north and the task of driving a robot south are related and follow a task distribution. However, the optimal policies for them have no shared component. If we minimize $L + J$ of two tasks, the solution performs badly for any of the two tasks. Therefore, minimizing $L + J$ is not applicable to our problem.
>
> This manuscript studies meta-learning, which is different from multi-task learning. Meta-learning aims to learn a meta-policy, such that the meta-policy can be adapted to tasks in a task distribution with a small amount of data. It trains a meta-policy during the meta-training. During the meta-test, the meta-policy is adapted to new tasks by the policy adaptation, Eq. 1 or Eq. 2 (lines 159 and 161).
>
> Therefore, the goal of Eq. 1 or Eq. 2 is adapting the meta policy to the new task $L$ in the meta-test, i.e., the adapted policy can minimize the loss of $L$ or is close to its minimum.
> During the meta-test, the original multiple tasks about $J$ are not used anymore, and the learned meta-policy $\pi_\phi$ serves as the prior knowledge to learn the task-specific policy for new tasks.
>
> Next, we would like to discuss why we cannot directly minimize $L$ without learning meta-policy $\pi_\phi$ and why the learned meta-policy $\pi_\phi$ is necessary. As we know, a fundamental difference between RL and supervised learning is that RL can not be solved on one-time collected data, i.e., the loss of $L$ cannot be minimized using the data collected on one policy. The RL algorithm minimizes $L$ by iterative data sampling and policy optimization. However, during the meta-test of meta-RL, we can only collect data on one policy (one-time data collection) on the new task of $L$ and it is impossible to minimize $L$. Therefore, we use the problem in Eq. 1 or Eq. 2 to approximate the minimization of $L$.
> Meta-RL is to learn how to better approximate the minimization of $L$, and the meta-policy $\pi_\phi$ in Eq. 1 or Eq. 2 is the knowledge that is learned by meta-RL to reduce the approximation error.
>
> >**Weakness 2. Can you plot the algorithm deployment performance for the example in Fig. 2? Since sometimes if you only do $\min_\theta ||\pi_\theta - K(x)||$, where $K(x)$ is a predefined controller, the $\theta$ converges fast, but the achieved $\pi_\theta$ does not work well in some examples.**
>
> **Answer:** In Fig. 2 (line 359), we show the performance of adapted policies on the meta-test tasks. The x-axis is the adaptation iteration number that the meta-policy is adapted to the new tasks, and the y-axis is the average accumulated reward of the adapted policies. The meta-test occurs after the new tasks are given, i.e., it represents the deployment of the learned meta-policy to new tasks. Therefore, Fig. 2 can reflect the performance of $\pi_\theta$ beyond the convergence. Figure 5 in Appendix B (line 600) shows the convergence of meta-training.

---

> > ### Comment · Reviewer_DbXf · 2024-08-12
> >
> > thank you, that answers my question.

---

### Official Review · Reviewer_Q7S4 · 2024-07-16

**Soundness:** 3
**Presentation:** 3
**Contribution:** 3
**Rating:** 7
**Confidence:** 4

**Summary:**

The paper proposes a bilevel optimization algorithm for Meta-RL, which unlike MAML, implements multiple-step policy optimization on one-time data collection. In addition, the paper provides an upper bound on the expected optimality gap over the task distribution, that quantifies the model’s generalizability to a new task from the task distribution. Experiments show the advantages of the proposed framework over existing meta-learning approaches.

**Strengths:**

* The paper proposes a practical algorithm, supported by a theoretical upper bound of the optimality of the proposed algorithm.
* The paper is well-written and easy to follow and understand.
* The appendix is well-organized and contains all the proofs and discussions about complexity and its relation to MAML.
* Experiments were performed to verify the theoretical results and show the advantage of the proposed algorithm over MAML.

**Weaknesses:**

* Comparison to state-of-the-art: the paper compares the proposed algorithm to MAML, EMAML, and ProMP, which were proposed 5 years ago. Since meta-RL is a very active research field, and even MAML has newer and more advanced variants, it is hard to be convinced that the paper compares the proposed algorithm to the most relevant approaches. In addition, the authors did not explain why they chose to compare to those specific baselines, and if the contribution of newer MAML variants can also be incorporated into their algorithm.

* The related work section can be improved. Specifically, the section is a single short paragraph that covers only bilevel optimization algorithms for meta-RL. An extended related work section on meta-RL (even in the appendix) that covers the different types of meta-RL algorithms and places the paper with respect to the various approaches could help the reader understand the paper's contribution.

**Questions:**

* Please address the aforementioned weaknesses.

* In Line 178: does NPG mean Natural Policy Gradient? If yes, maybe state it explicitly.

* Is it possible to formulate a similar bound for other meta-RL algorithms, such as RL2 [1] or VariBAD [2]?

* Lines 32-33 explain that even if the learned meta-parameter $\phi$ is close to the best one, the model adapted from the learned $\phi$ might be far from the task-specific optimum for some tasks, since the best meta-parameter is shared for all tasks and learned from the prior distribution that can be with high variance. The proposed algorithm was designed to adapt a stronger optimality metric, where the model is adapted from the task-specific optimal policy for each task. Could you provide a real experiment (even a synthetic experiment) that demonstrates this phenomenon?\
\
The existing experiments show that the proposed approach performs better than MAML, but it is still not entirely clear that this leads to the improved performance. A synthetic example can help fully explain this important failure case of existing meta-learning approaches.
* Perhaps I missed the following point when reading the paper, but during task-specific policy adaptation, the algorithm performs multiple-step policy optimization on one-time data collection, which speeds up the learning process - should it decrease performance (theoretically) compared to one-step policy optimization on one-time data collection?

[1] Yan Duan, John Schulman, Xi Chen, Peter L. Bartlett, Ilya Sutskever, and Pieter Abbeel. RL2: Fast reinforcement learning via slow reinforcement learning. arXiv:1611.02779, 2016.

[2] Luisa Zintgraf, Sebastian Schulze, Cong Lu, Leo Feng, Maximilian Igl, Kyriacos Shiarlis, Yarin Gal, Katja Hofmann, and Shimon Whiteson. VariBAD: Variational Bayes-adaptive deep RL via meta-learning. Journal of Machine Learning Research, 22(289):1–39, 2021.

**Limitations:**

The authors addressed the limitations of the proposed algorithm in section P.

---

> ### Author Rebuttal · Authors · 2024-08-07
>
> Thanks very much for your time and effort in reviewing our work. Thanks for your suggestions to make our manuscript better.
>
> >**Weakness 1. Comparison to state-of-the-art: the paper compares the proposed algorithm to MAML, EMAML, and ProMP, which were proposed 5 years ago. Since meta-RL is a very active research field, and even MAML has newer and more advanced variants, it is hard to be convinced that the paper compares the algorithm to the most relevant approaches. In addition, the authors did not explain why they chose to compare to those specific baselines, and if the contribution of newer MAML variants can also be incorporated into their algorithm.**
>
> **Answer:**  Thanks for the suggestive comments. In **the global rebuttal**, we introduce the categorization of meta-RL and compare the advantages/disadvantages of the categories. Based on it, we justify choosing MAML, EMAML, and ProMP as the baselines.
>
> In this manuscript, we focus on the category of optimization-based meta-RL. It is typical in EMAML, ProMP, and [10, 1.a] that optimization-based methods are only compared with optimization-based methods (due to their worse optimality than black-box methods in the in-distribution meta-test setting). So, in the experiments, we compare the proposed algorithm with the existing optimization-based meta-RL approaches, including MAML, EMAML, and ProMP. The experimental results show that the proposed method can outperform the baselines significantly. Moreover, we also achieve the state-of-art theoretical result over all optimization-based meta-RL papers, as shown in Table 1 (line 35).
>
> Recent optimization-based meta-RL papers, including new variants of MAML [10, 1.a], aim to solve the meta-gradient estimation issues. They usually do not significantly outperform MAML, EMAML and ProMP. In **the attached PDF file of the global rebuttal**, we include a recent baseline from [10] (2022) and compare their performances. The results show that the proposed method can also outperform this baseline.
>
> [1.a] Tang, Yunhao. "Biased gradient estimate with drastic variance reduction for meta-reinforcement learning", 2022.
>
> >**Weakness 2. The related work section can be improved. Specifically, the section is a short paragraph that covers only bilevel optimization algorithms for meta-RL. An extended related work section on meta-RL that covers the different types of meta-RL algorithms and places the paper with respect to the various approaches could help the reader understand the paper's contribution.**
>
> **Answer:** Thanks for the suggestion. In **the global rebuttal** and **the answer to Weakness 1**, we discuss the categorization of existing meta-RL algorithms, compare the two categories, and the advantages of the method in the manuscript over the existing optimization-based meta-RL methods. These discussions make our contribution more clear. We will add them to the revised manuscript.
>
> >**Question 2. Is it possible to formulate a similar bound for other meta-RL algorithms, such as RL2 [1] or VariBAD [2]?**
>
> **Answer:** This manuscript focuses on the theoretical analysis of optimization-based meta-RL. To the best of our knowledge, almost all papers that work on the theoretical analysis of meta-RL algorithms [10, 38, 50, 52] focus on optimization-based meta-RL. From **the answer to Weakness 1**, we can see that the design of optimization-based methods is usually inspired by theoretical analysis, and the design of the black-box method is often more heuristic. As a result, it is challenging to derive optimality bounds for black-box meta-RL, such as RL2 and VariBAD.
>
> >**Question 3. Lines 32-33 explain that even if the learned meta-parameter is close to the best one, the model adapted from the learned might be far from the task-specific optimum for some tasks, since the best meta-parameter is shared for all tasks and learned from the prior distribution that can be with high variance. The proposed algorithm was designed to adapt a stronger optimality metric, where the model is adapted from the task-specific optimal policy for each task. Could you provide a real experiment (even a synthetic experiment) that demonstrates this phenomenon? The existing experiments show that the proposed approach performs better than MAML, but it is still not entirely clear that this leads to improved performance. A synthetic example can help fully explain this important failure case of existing meta-learning approaches.**
>
> **Answer:** We apologize for the confusion regarding the reason for the performance improvement of the proposed algorithm over MAML. The better performance of the proposed algorithm is not due to the use of a stronger optimality metric. The stronger optimality metric is only used to evaluate the proposed algorithm. It is not directly used in the design of the algorithm. Instead, the reason is that we design the policy adaptation problem (Problem (1), line 159) and solve it by multiple optimization steps.
>
> In specific, MAML and its variants apply the policy gradient on the one-time data collection during the meta-test. Problem (1) (line 159) is to maximize a surrogate function, which is an approximate total reward function (as indicated in line 170) using one-time data collection. We solve the optimal solution of Problem (1) by multiple optimization steps. The objective function of Problem (1) is a better approximation for the total reward function than that of the policy gradient in MAML, and therefore can achieve a better performance than MAML.
>
> Moreover, since the objective of Problem (1) is a lower bound of the total reward function (stated in Lemmas 1 and 2, line 304), we can derive the optimality bound of the proposed algorithm under a stronger optimality metric. The theoretical analysis provides more insight into why the design of Problem (1) is good.
>
> To improve clarity, we will swap the third paragraph and the fourth paragraph of the introduction, and add a necessary clarification to the modified manuscript.

---

> ### Author Response · Authors · 2024-08-07
> **Supplementary answer for the reviewer's comment**
>
> We thank the reviewer again for reviewing our work. Here, we have supplementary answers to the reviewer's questions.
>
> >**Question 1. In Line 178: does NPG mean Natural Policy Gradient? If yes, maybe state it explicitly.**
>
> **Answer:** Yes. Thanks for pointing it out. We will clarify it in the revised manuscript.
>
> >**Question 4. During task-specific policy adaptation, the algorithm performs multiple-step policy optimization on one-time data collection, which speeds up the learning process - should it decrease performance (theoretically) compared to one-step policy optimization on one-time data collection**
>
> **Answer:** In this manuscript, our algorithm performs (i) multiple-step policy optimization on one-time data collection, i.e., for each iteration of the meta-test, we collect data on the current policy and adapt the policy using multiple optimization steps. MAML performs (ii) one-step policy optimization on one-time data collection, i.e., for each iteration of the meta-test, it collects data on the current policy and adapts the policy using a one-step policy gradient. As indicated in lines 680-690, the computation times of (i) and (ii) are comparable for one iteration. On the other hand, the optimality of our algorithm is better than MAML for a single iteration, i.e.,  higher data efficiency of the proposed algorithm. So, our algorithm takes fewer iterations and less training time to reach a given optimality requirement, i.e., speed up the adaptation process. The speed-up is due to the higher data efficiency and the better optimality.

---

> > ### Comment · Reviewer_Q7S4 · 2024-08-11
> >
> > I thank the authors for their detailed response.
> >
> > The categorization of existing meta-RL algorithms provided in the global rebuttal answer helps to clarify the differences between the various meta-RL algorithms and place the proposed method relating to the existing approaches. Adding this discussion to the paper would improve the contribution of the manuscript.
> >
> > The discussion explains that although optimization-based meta-RL methods hold worse performance than black-box methods, they are more robust to sub-optimal meta-policies and can handle tasks outside the training distribution, compared with black-box methods (such as RL2 and VariBAD).
> >
> > Can you provide a reference that supports this claim? I understand that the motivation for black-box algorithms such as VariBAD or RL2 was to learn the prior distribution over tasks and theoretically should not work with test tasks outside this prior distribution, but is there anyone who performed any experiment that validates that optimization-based methods work better in such scenarios?
> >
> > I appreciate the added comparison to the recent baseline and the detailed answers to my questions.

---

> ### Author Response · Authors · 2024-08-11
>
> Thanks for your reply.
>
> The categorization of meta-RL approaches and the comparison of the differences between the two categories are justified in [1]. The experiments conducted in [2,3] show that optimization-based meta-RL methods are more robust to sub-optimal meta-policies and can handle tasks outside the training distribution.
>
>
> [1] Beck, Jacob, et al. "A survey of meta-reinforcement learning", 2023.
>
> [2] Xiong, Zheng, et al. "On the Practical Consistency of Meta-Reinforcement Learning Algorithms", 2021.
>
> [3] Finn, Chelsea, et al. "Meta-Learning and Universality: Deep Representations and Gradient Descent Can Approximate any Learning Algorithm", 2018.

---

> > ### Comment · Reviewer_Q7S4 · 2024-08-12
> >
> > Thanks for your reply.
> >
> > These references help to improve the clarity of the discussion above. I didn't find references [2,3] in the manuscript, and I think that it would be beneficial to add them.
> >
> > I raised my score since the authors answered all my concerns.

---

### Author Rebuttal · Authors · 2024-08-07

We are grateful and indebted for the time and effort invested to evaluate our manuscript by all reviewers, and for all the suggestions and reference recommendations to make our manuscript a better and stronger contribution. Please find below our detailed replies to all the reviewers' comments.

In this global rebuttal, we would like to discuss the categorization of existing meta-RL algorithms and compare the advantages and disadvantages of these categories.

>**Categorization of existing meta-RL.** As mentioned in the second paragraph of the introduction (line 19), meta-RL methods can be generally categorized into (i) optimization-based meta-RL, (ii) black-box (also called model-based or context-based) meta-RL. Optimization-based meta-RL approaches, such as MAML and its variants, usually include a policy adaptation algorithm and a meta-algorithm. During the meta-training, the meta-algorithm aims to learn a meta-policy, such that the policy adaptation algorithm can achieve good performance starting from the meta-policy. The learned meta-policy parameter is adapted to the new task using the policy adaptation algorithm during the meta-test.
Black-box meta-RL, such as RL2 and VariBAD, aims to learn an end-to-end neural network model. The model has fixed parameters for the policy adaptation during the meta-test, and generates the task-specific policy using the trajectories of the new task takes. The meta-RL algorithm in the manuscript is an optimization-based method.
>
>
>Optimization-based meta-RL methods are typically less specialized to the training tasks and hold worse performance than black-box methods. However, it is more robust to sub-optimal meta-policies and can handle tasks outside the training distribution, compared with black-box methods.
In optimization-based meta-RL, the task-specific policy is adapted from a shared meta-policy over the task distribution. The learned meta-knowledge is not specialized for any task, and its meta-test performance on a task depends on a general policy optimization algorithm applied to new data from that task.
In contrast, the end-to-end model in black-box meta-RL typically includes specialized knowledge for any task within the task distribution, and uses the new data merely as an indicator to identify the task within the distribution.
As a result, the optimality of optimization-based methods is usually worse than black-box methods, especially when the task distribution is heterogeneous and the data scale for adaptation is extremely small.
On the other hand, the policy adaptation algorithms in the meta-test of optimization-based methods can generally improve the policy and show the convergence starting from any initial policy, not only the learned meta-policy. Therefore, it is robust to sub-optimal meta-policy and can deal with tasks that are out of the training task distribution. In contrast, due to the specialization of the learned model, black-box methods cannot be generalized outside of the training task distribution.

---

### Decision · Program_Chairs · 2024-09-25

**Decision:**

Accept (poster)

**Comment:**

The paper proposes a bilevel optimization algorithm for Meta-RL. Different from the popular approach of MAML, the proposed algorithm performs multiple-step policy optimization on one-time data collection. The paper provides an upper bound on the expected optimality gap over the task distribution, that quantifies the model’s generalizability to a new task from the task distribution. The superior performance of the proposed algorithm over the existing meta-RL method has been shown via simulation experiments on standard benchmarks.

We received three expert reviews, with the scores, 5, 6, 7, and the average score is 6.00. Reviewers are happy about the novelty of the proposed approach, the theoretical guarantees provided. The reviewers are also generally happy about the quality of the presentation.

Reviewer DbXf has a question about the difference between the proposed method and multi-task learning, and the authors have clearly addressed this question in their rebuttal. Reviewer 8Th6 has expressed concerns about the implementation complexity and the comparison with more benchmarks. The authors have agreed to include more details about this in their revision, and the reviewer has accepted that.

I have read the paper, the reviews, and the rebuttal. I believe that the paper proposes an interesting algorithmic approach for meta-RL. While mainly an algorithmic paper, it also provides some theoretical guarantees for the proposed approach. The simulation experiments are convincing.

Please revise your paper based on the reviewers’ comments for the final submission.